

# Lokiceratops rangiformis gen. et sp. nov. (Ceratopsidae: Centrosaurinae) from the Campanian Judith River Formation of Montana reveals rapid regional radiations and extreme endemism within centrosaurine dinosaurs

Mark A. Loewen[1,2,3,*], Joseph J. W. Sertich[3,4,5,*], Scott Sampson[6], Jingmai K. O'Connor[7], Savhannah Carpenter[2], Brock Sisson[8], Anna Øhlenschlæger[3], Andrew A. Farke[9], Peter J. Makovicky[10], Nick Longrich[11] and David C. Evans[12,13]

[1] Natural History Museum of Utah, Salt Lake City, UT, United States of America
[2] Department of Geology and Geophysics, University of Utah, Salt Lake City, Utah, United States of America
[3] Evolutionsmuseet, Knuthenborg, Maribo, Denmark
[4] Smithsonian Tropical Research Institute, Panama City, Panamá
[5] Department of Geosciences, Colorado State University, Fort Collins, Colorado, United States of America
[6] California Academy of Sciences, San Francisco, California, United States of America
[7] The Field Museum, Chicago, Illinois, United States of America
[8] Independent Researcher, Pleasant Grove, Utah, United States of America
[9] Raymond M. Alf Museum of Paleontology, Claremont, California, United States of America
[10] Department of Earth and Environmental Sciences, University of Minnesota, Minneapolis, Minnesota, United States of America
[11] Department of Life Sciences, University of Bath, Bath, United Kingdom
[12] Department of Natural History, Royal Ontario Museum, Toronto, Ontario, Canada
[13] Department of Ecology and Evolution, University of Toronto, Toronto, Ontario, Canada
* These authors contributed equally to this work.

Corresponding author
Mark A. Loewen,
mloewen@nhmu.utah.edu

## ABSTRACT

The Late Cretaceous of western North America supported diverse dinosaur assemblages, though understanding patterns of dinosaur diversity, evolution, and extinction has been historically limited by unequal geographic and temporal sampling. In particular, the existence and extent of faunal endemism along the eastern coastal plain of Laramidia continues to generate debate, and finer scale regional patterns remain elusive. Here, we report a new centrosaurine ceratopsid, *Lokiceratops rangiformis*, from the lower portion of the McClelland Ferry Member of the Judith River Formation in the Kennedy Coulee region along the Canada-USA border. Dinosaurs from the same small geographic region, and from nearby, stratigraphically equivalent horizons of the lower Oldman Formation in Canada, reveal unprecedented ceratopsid richness, with four sympatric centrosaurine taxa and one chasmosaurine taxon. Phylogenetic results show that *Lokiceratops*, together with *Albertaceratops* and *Medusaceratops*, was part of a clade restricted to a small portion of northern Laramidia approximately 78 million years ago. This group, Albertaceratopsini, was one of multiple centrosaurine clades to undergo

geographically restricted radiations, with Nasutuceratopsini restricted to the south and Centrosaurini and Pachyrostra restricted to the north. High regional endemism in centrosaurs is associated with, and may have been driven by, high speciation rates and diversity, with competition between dinosaurs limiting their geographic range. High speciation rates may in turn have been driven in part by sexual selection or latitudinally uneven climatic and floral gradients. The high endemism seen in centrosaurines and other dinosaurs implies that dinosaur diversity is underestimated and contrasts with the large geographic ranges seen in most extant mammalian megafauna.

## INTRODUCTION

Late Cretaceous dinosaur-dominated ecosystems from the Western Interior of North America present an unparalleled opportunity to examine evolution along a latitudinal gradient and within a relatively constrained time interval (~83 to ~70 Ma). Lying along the alluvial and coastal plains of Laramidia, the differences between dinosaur assemblages of the Western Interior were noted several decades ago (*e.g.*, *Russel, 1967*, *1969*), and they were later divided broadly into northern and southern regions (*e.g.*, *Lehman, 1997*, *2001*).

Recent discoveries from underexplored regions of Laramidia, with increased attention to stratigraphic position, geochronology, and regional ecologies, have refined hypotheses regarding dinosaur distribution and evolution in Laramidia (*e.g.*, *Gates et al., 2010b*; *Sampson & Loewen, 2010*; *Sampson et al., 2010*; *Loewen et al., 2013*), though some doubts persist regarding the degree and nature of these differences (*e.g.*, *Lucas et al., 2016*; *Fowler, 2017*). Regardless, increased sampling and stratigraphic resolution reveal local and regional patterns in dinosaur evolution, including rapid turnover of megaherbivores (*Mallon et al., 2012*; *Mallon, 2019*), potential anagenetic evolution (*Horner, Varricchio & Goodwin, 1992*; *Freedman-Fowler & Horner, 2015*; *Carr et al., 2017*; *Fowler & Freedman-Fowler, 2020*; *Wilson, Ryan & Evans, 2020*), and unexpected new forms (*e.g.*, *Brown & Henderson, 2015*; *Wiersma & Irmis, 2018*).

Within the dinosaur ecosystems of Laramidia, the Ceratopsidae were geographically widespread and morphologically diverse, possessing highly variable cranial ornaments including horns and morphologically diverse parietosquamosal frills (*Marsh, 1891a*; *Hatcher, Marsh & Lull, 1907*; *Lull, 1933*; *Dodson, Forster & Sampson, 2004*; *Sampson & Loewen, 2010*). Two distinct clades within Ceratopsidae diverged by at least ~83 Ma. These are the long-nosed, long-frilled Chasmosaurinae, characterized by *Chasmosaurus belli* (*Lambe, 1902*), *Pentaceratops sternbergii* (*Osborn, 1923*), and *Torosaurus latus* (*Marsh, 1891b*), and the round-nosed, relatively short-frilled Centrosaurinae, characterized by *Diabloceratops eatoni* (*Kirkland & DeBlieux, 2010*), *Centrosaurus apertus* (*Lambe, 1904*), *Styracosaurus albertensis* (*Lambe, 1913*), and *Pachyrhinosaurus lakustai* (*Currie, Langston & Tanke, 2008*).

Centrosaurinae represent an ecologically important and diverse radiation of ceratopsids, reaching peak diversity in the Campanian (~83–70 Ma). Historically known from abundant remains in Alberta, Canada and Montana, USA, discoveries over the past two decades have rapidly expanded our understanding of the clade, particularly its geographic (*Xu et al., 2010*; *Loewen et al., 2010*; *Fiorillo & Tykoski, 2012*) and morphologic breadth, with additional insights into centrosaurine ontogeny (*Sampson, Ryan & Tanke, 1997*; *Ryan et al., 2001*; *Tumarkin-Deratzian, 2009*; *Frederickson & Tumarkin-Deratzian, 2014*; *Brown, Russell & Ryan, 2009*; *Brown et al., 2020*). Though locally abundant in some localities in southern Alberta and northern Montana (*e.g.*, *Centrosaurus apertus* (*Lambe, 1904*), *Styracosaurus albertensis* (*Lambe, 1913*), and *Pachyrhinosaurus canadensis* (*Sternberg, 1950*)), centrosaurines were previously rare or poorly known from other regions of Laramidia. Our expanding knowledge about centrosaurines includes new taxa from the southwestern United States and Mexico (*e.g.*, *Diabloceratops eatoni* (*Kirkland & DeBlieux, 2010*), *Nasutoceratops titusi* (*Sampson et al., 2013*; *Lund, Sampson & Loewen, 2016*), *Machairoceratops cornusi* (*Lund et al., 2016*), *Yehuecauhceratops mudei* (*Rivera-Sylva, Hendrick & Dodson, 2016*; *Rivera-Sylva et al., 2017*), *Crittendenceratops krzyzanowskii* (*Dalman et al., 2018*), *Menefeeceratops sealeyi* (*Dalman et al., 2021*)) and new and reinterpreted taxa from Montana and Canada (*e.g.*, *Coronosaurus brinkmani* (*Ryan & Russell, 2005*; *Ryan, Evans & Shepherd, 2012*), *Albertaceratops nesmoi* (*Ryan, 2007*), *Pachyrhinosaurus. lakustai* (*Currie, Langston & Tanke, 2008*), *Styracosaurus ovatus* (*McDonald & Horner, 2010*; *Wilson, Ryan & Evans, 2020*), *Spinops sternbergorum* *Farke et al., 2011*, *Medusaceratops lokii* (*Ryan, Russell & Hartman, 2010*; *Chiba et al., 2017*), *Pachyrhinosaurus perotorum* (*Fiorillo & Tykoski, 2012*), *Xenoceratops foremostensis* (*Ryan, Evans & Shepherd, 2012*), *Wendiceratops pinhornensis* (*Evans & Ryan, 2015*), and *Stellasaurus ancellae* (*Wilson, Ryan & Evans, 2020*)). Many of these new taxa have changed our understanding of morphological disparity within the clade. This proliferation of new taxa and occurrences has enhanced our understanding of the evolution of Centrosaurinae and provides clues regarding the mechanisms driving diversification of large vertebrates in Laramidia (*Sampson & Loewen, 2010*; *Gates et al., 2010a*).

The Campanian deposits of the Judith River Formation of Montana and the Belly River Group of Alberta and Saskatchewan (Foremost, Oldman, and Dinosaur Park formations) preserve a suite of parasynchronous non-marine biotas. Among the most abundant large vertebrates from these deposits are ceratopsid dinosaurs, including both chasmosaurines and centrosaurines. These assemblages represent some of the richest known from the Western Interior (*Weishampel et al., 2004*; *Ryan & Evans, 2005*; *Currie & Russell, 2005*), spanning sediments dated between ~79.4 and ~75.2 million years ago (*Roberts et al., 2013*; *Rogers et al., 2016*; *Ramezani et al., 2022*; *Rogers, Eberth & Ramezani, 2023*).

A new, relatively complete centrosaurine from the lower part of the McClelland Ferry Member of the Judith River Formation, in Kennedy Coulee in northern Montana, USA, is described here as a distinct genus and species, *Lokiceratops rangiformis*. The new taxon is in the same narrow stratigraphic interval and geographic area (Fig. 1) as three other centrosaurines (*Wendiceratops pinhornensis*, *Albertaceratops nesmoi*, and *Medusaceratops lokii*) and one chasmosaurine (*Judiceratops tigris*). Morphologically, *Lokiceratops*

resembles both *Albertaceratops* and *Medusaceratops*, implying rapid, sympatric diversification within a clade, a pattern not previously seen in dinosaurs. Furthermore, the possible sympatric occurrence of five distinct ceratopsids (four centrosaurines, one chasmosaurine) is unparalleled in any other known interval in Laramidia, even in more heavily sampled and documented horizons (*e.g.*, *Mallon et al., 2012*). This discovery supports a novel hypothesis that some dinosaur clades saw rapid regional radiations rather than anagenesis in geographically limited regions along the coastal and alluvial plains of Laramidia.

## Geological context

The Loki Quarry producing the new specimen lies on private land in the badlands of Kennedy Coulee, north of the town of Rudyard in Hill County, Montana, USA (Fig. 1). The upstream end of Kennedy Coulee is also known as Canadian Creek where it originates north of the US/Canada border, west of its confluence with the Milk River. In these badlands, Campanian alluvial deposits of the lower part of the Judith River Formation (*Goodwin & Deino, 1989*; *Rogers, 1998*) crop out extensively along the drainage systems flowing toward the Milk River Valley in the north (Fig. 1).

Following recent stratigraphic revision of the Judith River Formation by *Rogers et al. (2016)*, *Rogers, Eberth & Ramezani (2023)*, the exposed Kennedy Coulee beds correlate in the subsurface to the McClelland Ferry Member to the south, as well as to the upper parts of the Foremost and overlying Oldman formations of southern Alberta to the north, including the Taber Coal Zone and the Herronton Sandstone Zone (*Ogunyomi & Hills, 1977*; *Eberth & Hamblin, 1993*; *Cullen et al., 2016*; *Eberth, 2024*). The Taber Coal Zone, representing the top of the Foremost Formation in Alberta and its correlative coal deposits capped by the Marker A Coal exposed to the south in Montana, represents a datum for calibrating stratigraphic sections and associated fossil taxa (*Eberth & Hamblin, 1993*; *Brinkman et al., 2004*; *Eberth, 2005*; *Ryan, 2007*; *Evans & Ryan, 2015*; *Freedman-Fowler & Horner, 2015*; *Cullen et al., 2016*; *Ryan et al., 2017*; *Rogers, Eberth & Ramezani, 2023*). It should be noted that the Taber Coal Zone is a sequence of coal seams that is much thicker north and west of the Loki Quarry in Canada, near the South Side Ceratopsian Quarry. Physical tracing of this interval along the Milk River into Kennedy Coulee in the area of the Loki Quarry demonstrates that it is laterally continuous with the Marker A Coal seam.

The Loki Quarry lies near two other significant ceratopsian localities in the same Canadian Creek area within Kennedy Coulee (Fig. 1). The Loki Quarry is 4.9 km northwest of the site where the holotype of the putative chasmosaurine ceratopsid *Judiceratops tigris* (YPM VPPU 022404) was collected, and 2.6 km west of the Mansfield Bonebed (*Medusaceratops lokii*). The Mansfield Bonebed that produced *Medusaceratops* occurs ~8 km southwest of the *Probrachylophosaurus bergei* quarry. The Loki Quarry lies 2.8 km west of the *Brachylophosaurus goodwini* (*Horner, 1988*) holotype locality (UCMP Locality No. V83125). Two other important ceratopsian quarries lie just north of the Montana/Alberta border. The South Side Ceratopsian *Wendiceratops* quarry (*Evans & Ryan, 2015*) is 10 km north of the Montana-Alberta border and the *Albertaceratops* quarry (*Ryan, 2007*)

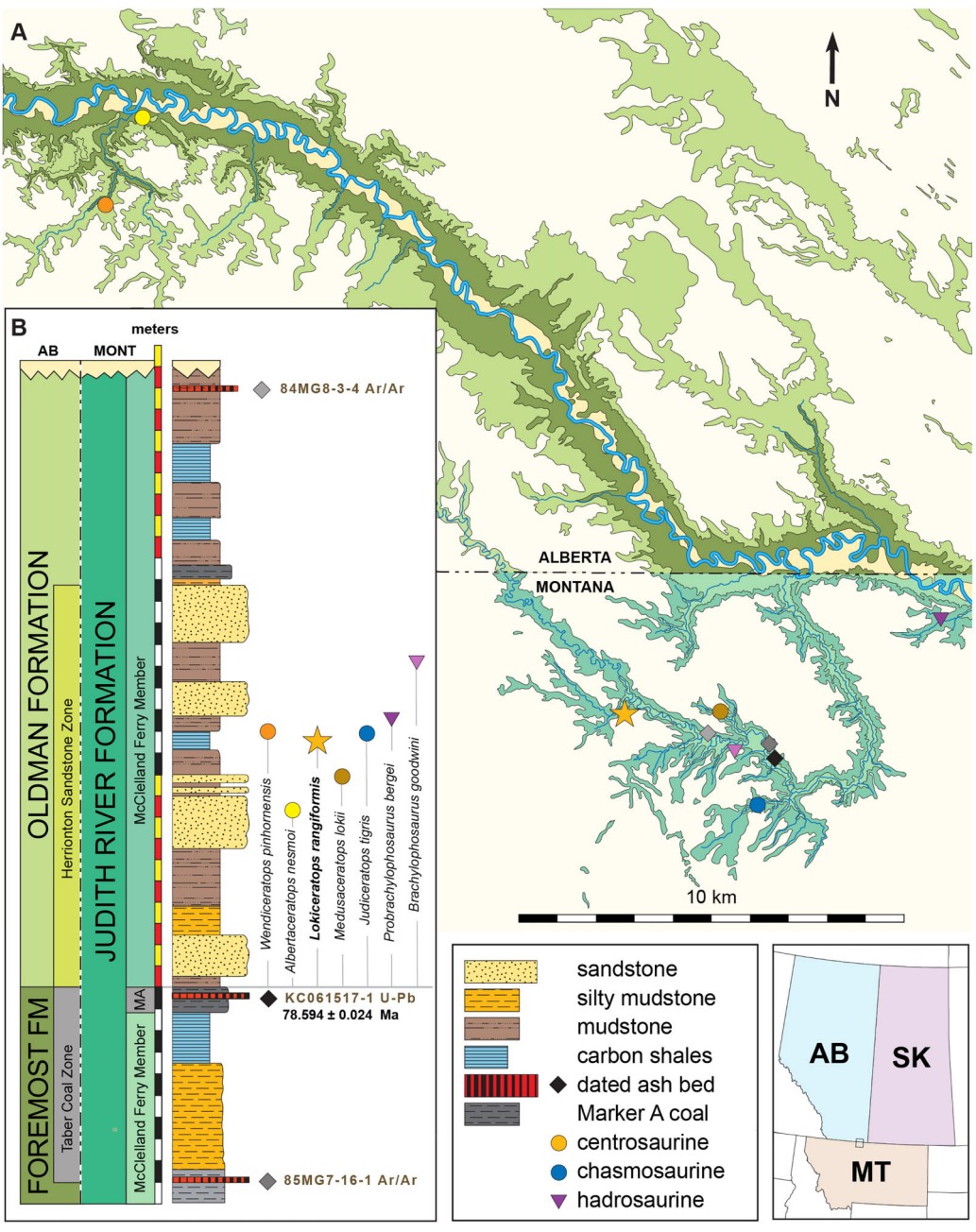

**Figure 1 Geographic and stratigraphic relationships of the holotype EMK 0012 and the Loki Quarry in northern Montana.** (A) Regional relationships between the cross-border paleontological sites in the Oldman and Judith River formations along the Milk River and in Kennedy Coulee in Alberta and Montana. (B) Generalized stratigraphic section in the Kennedy Coulee area modified after *Goodwin & Deino (1989)* and *Rogers, Eberth & Ramezani (2023)* with the relationships between the Foremost and Oldman formations in Canada and the Judith River Formation in Montana. Relative placements of important taxa in this area are indicated. Position of $^{40}$Ar/$^{39}$Ar dates originally obtained by *Goodwin & Deino (1989)* are shown in relation to the new U–Pb CA-ID-TIMS date for KC061517-1 by *Ramezani et al. (2022)*. Bentonite ash beds are only 5 to 7 cm thick so they are exaggerated for clarity. Scale bars delineated in map view are indicated kilometers and in meters stratigraphically.

is 3.5 km north of the South Side Ceratopsian *Wendiceratops* quarry. The Loki Quarry is 22 km southwest of the South Side Ceratopsian quarry (Fig. 1).

The Loki Quarry sits 922 m above sea level and 11.4 m above the top of the Marker A Coal (MAC) seam. The MAC seam is equivalent to the top of the Taber Coal Zone (*sensu Goodwin & Deino, 1989*) based on multiple sections measured in the Kennedy Coulee and at the *Probrachylophosaurus* (*Freedman-Fowler & Horner, 2015*) locality (MOR locality JR-518). The Mansfield Bonebed producing *Medusaceratops* occurs ~10 m above the MAC. All of these quarries occur near the top of a 10–15 m thick interval of interbedded organic-rich mudstones with discontinuous carbonaceous seams, siltstone, and sandstones (Fig. 1).

The stratigraphic occurrence of the Loki Quarry places it above *Medusaceratops* (~10 m above the MAC) and places both taxa within equivalents of the Herronton Sandstone Zone, in the same stratigraphic interval where *Albertaceratops* and *Wendiceratops* were recovered in southern Alberta. Correlation to the top of the Taber Coal Zone (TCZ) places *Albertaceratops* slightly lower in section (~8 m above the TCZ) with respect to *Medusaceratops* (~10 m above the MAC) and places the Loki Quarry at roughly the same level as *Wendiceratops* (~8 and 12 m above the TCZ), making them virtually indistinguishable stratigraphically. The multiple mudstone and sandstone beds and channel deposits recognized as the Herronton Sandstone Zone and its correlative equivalents in the McClelland Ferry Member to the south are laterally discontinuous and variable in nature across the region from north to south and east to west, but do represent a package of similar deposition (*Rogers et al., 2016*; *Eberth, 2024*). These relationships suggest that these centrosaurine quarries are stratigraphically equivalent within the precision that is possible, even though the relative occurrences of these taxa may be slightly uncertain with respect to one another.

Two bentonite ash beds that bracket the Loki Quarry (21 m below and 16 m above) were first radiometrically dated by *Goodwin & Deino (1989)*. The single-crystal, laser-fusion $^{40}$Ar/$^{39}$Ar ages on biotite crystals yielded a weighted mean of 78.5 ± 0.2 Ma for bentonite 85MG7-16-1, approximately 21 m below the quarry, and a weighted mean of 78.2 ± 0.2 Ma for bentonite 84MG8-3-4, approximately 16 m above the quarry (Fig. 1). The ages were recalibrated to the Fish Canyon Tuff (*Kuiper et al., 2008*; *Renne et al., 2011*) by *Roberts et al. (2013)* to 79.02 and 78.71 Ma respectively (using the original legacy decay constant and known fluence monitors) and by *Fowler (2017)* to 79.76 ± 0.2 and 79.46 ± 0.2 Ma using the original mean. For the purposes of this study, and until additional geochronologic work is undertaken in the northern Judith River Fm near the study area, we instead prefer to use recently published high-precision U-Pb dates of *Ramezani et al. (2022)*, summarized below.

High-precision U–Pb analyses of zircons by the CA-ID-TIMS method from a bentonitic ash bed within Marker A Coal (KC061517-1) 11.4 m below the Loki Quarry date to 78.594 ± 0.024 Ma (*Ramezani et al., 2022*). Bayesian model uncertainty constrained by U-Pb dates (*Ramezani et al., 2022*) places the Loki Quarry between 78.4 and 77.2 Ma, with a model median age estimate of roughly 78.1 Ma. The use of Bayesian models to bound ages allows

for better accommodation of changes in sedimentation and more honest (*i.e.*, asymmetric) evaluation of uncertainties.

The lithology of the Loki Quarry is characterized by carbonaceous fine-grained sandstones, siltstones, and mudstones, with depositional features indicating a poorly-drained fluvial system (Figs. 1 and 2). Gar scales and mollusks occur in the quarry. Some of the quarry matrix is in the collections of Evolutionsmuseet, Knuthenborg, Maribo, Denmark. Carbonized plant fragments are common, many attributable to Araucariales, along with beads of amber and indeterminate fragments of carbonized wood.

Many bones recovered from the quarry are broken, but there is no evidence of subaerial or subaqueous weathering of any elements. Some breakage may reflect collection techniques, because most elements were plucked from the quarry sediments and only two plaster jackets (scapulocoracoid and sacrum) were made. Many of the bones were plastically deformed after deposition by compression of the clay-rich, fine-grained sediments. This deformation skews the bones so that the mount does not accurately represent the skull shape. Taphonomic indicators, including a high degree of association of the cranial bones (Fig. 2), indicate little to no fluvial transport after death and disarticulation.

### Discovery and excavational history

EMK 0012 is an associated skeleton of a mature ceratopsid. The specimen was discovered by Mark Eatman on private land of the Wolery Ranch in Kennedy Coulee in late spring of 2019 and excavated under lease later that fall. The skull was associated, but partially disarticulated. The right jugal and squamosal were found together, dorsal side up. Portions of the parietosquamosal frill were found in close association. Both orbits and postorbital horns were found on either side of the braincase, with both maxillae directly in front of them followed by the nasal, premaxillae, and rostral. The synsacrum and ilia were found ventral side facing up, with the right ischium in articulation; the left ischium lay one meter away (Fig. 2). The left parietal with fused epiparietals ep1–ep7 was found dorsal side up along with the left ischium. The right scapulocoracoid was found medial side up just posterior to the pelvis. The free anterior caudal vertebra and chevron were found next to the pelvis. Legal ownership of EMK 0012 was permanently transferred to Evolutionsmuseet, Knuthenborg in 2021, where the specimen is available to researchers.

### Preparation and reconstruction

EMK 0012 was delivered to Fossilogic LLC in Pleasant Grove, Utah for preparation, restoration, mounting, and reconstruction. The skull was received in multiple fragments wrapped in aluminum foil along with two blocks protected with plaster and burlap field jackets. Preparation began with removal of jackets, foil, matrix, and any stabilizing cyanoacrylate applied in the field. Hairline cracks were stabilized using a low-viscosity (2–3 centipose, roughly equivalent to the viscosity of milk) cyanoacrylate (Starbond EM-02). Larger pieces were glued together using a gel-like high-viscosity (2000 3 centipose, roughly equivalent to the viscosity of honey) cyanoacrylate (Starbond EM-2000). Some larger cracks were filled with a polyester resin (Key-Lite) that was not painted to make gap fills obvious to researchers. Finally, all bones were sealed and stabilized with a matte clear

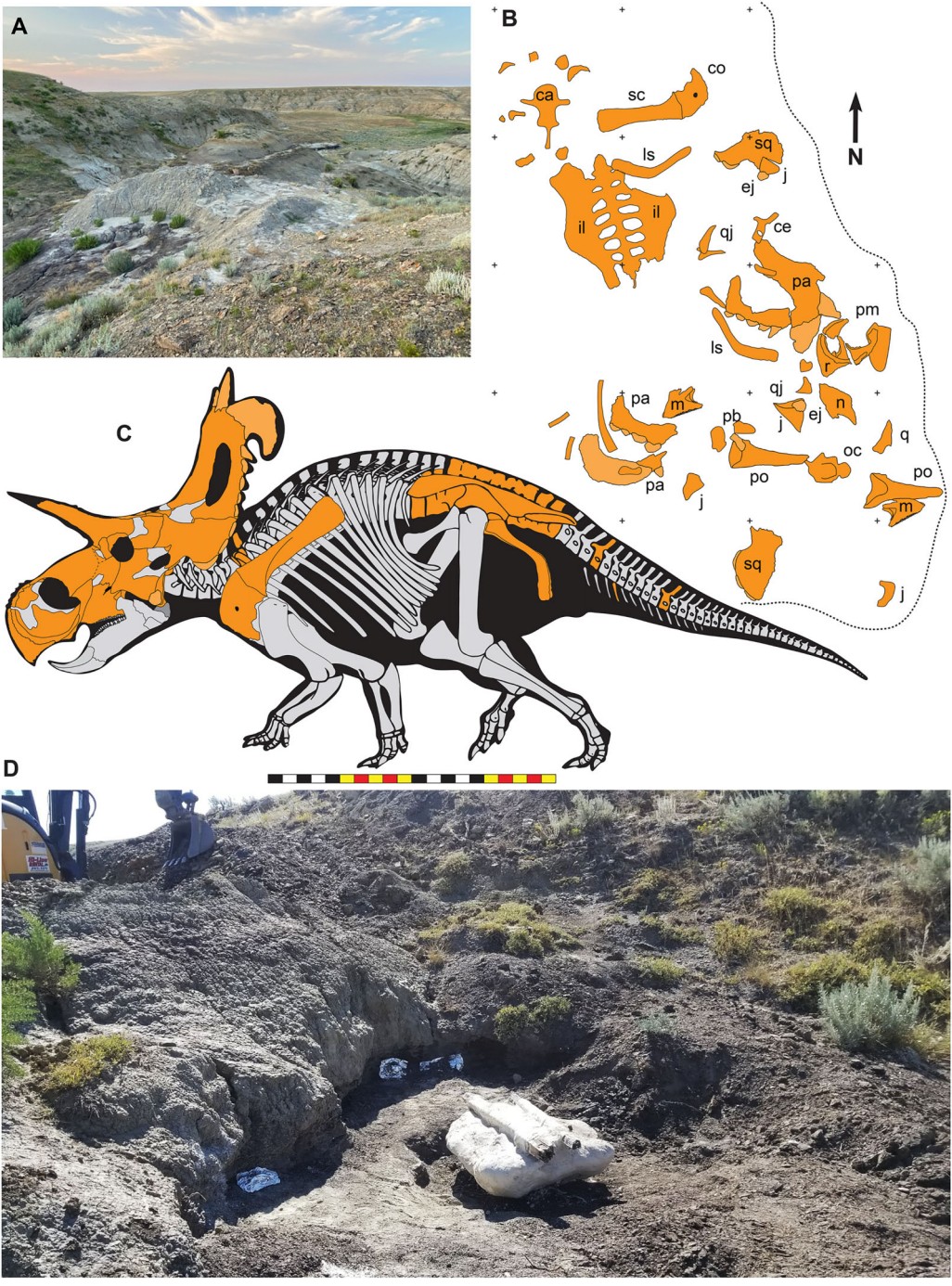

**Figure 2 The Loki Quarry where EMK 0012 was excavated in Hill County, northcentral Montana USA.** (A) View of the Loki Quarry facing north. (B) Quarry map with 1-m grids marked by corner ticks. (C) Osetograph of the skeletal completeness of EMK 0012, cranial elements represent presence on either right or left side. (D) Jacketed pelvic block awaiting removal from the Loki Quarry. *Osteological abbreviations*: ca, caudal vertebrae; co, coracoid; ej, epijugal, il, ilium; is, ischium; j, jugal; l, lacrimal; m, maxilla; oc, occipital condyle; pa, parietal; pb, palpebral; pm, premaxilla; po, postorbital; q, quadrate; qj, quadratojugal; r, rostral; sc, scapula sq, squamosal. Skeletal reconstruction by Mark Loewen. Photo A by David Evans and Photo D provided courtesy of Evolutionsmuseet, Knuthenborg, Maribo, Denmark. Scale bar in C equals 2 m.

paraloid ethyl methacrylate co-polymer B-72 (Rust-Oleum). Preparation was largely performed by Jen Sellers and Estrella Gallegos over the period of several weeks during the fall of 2021.

Following preparation, each element was surrounded by silicone rubber molds prior to any restoration to preserve scientifically valuable data as research casts in a polyurethane casting plastic. These casts are available at the Natural History Museum of Utah as NHMU VP C-991. Mark Loewen, Joseph Sertich, Savhannah Carpenter, and Brock Sisson determined the identity of all recovered elements and articulated and assembled them into their proper locations in a 3D skull reconstruction. Missing elements were sculpted as mirror images of existing material from blocks of polyester resin (Key-Lite). Where plastic deformation had deformed bones, the casts were heated to allow retrodeformation and restored, or cut and restored to original shapes.

Upon assembly, the restored 3D cast skull was surrounded by a silicon rubber mold enabling multiple replicas to be cast. This process included sectioning the restored skull into several major sections: the right and left face, the frill, braincase and quadrates. These sections of the skull were surrounded in clay along a parting line with corresponding keys, vents, and sprues as needed with a hard mother mold of fiberglass and polyester resin to support the flexible silicone and retain its shape. Each section was then flipped and the clay removed, excepting the vents and spues, and the process was repeated. The finished two-part molds (the braincase was a three-part mold) were then opened and the master-cast removed. The molds were then filled with a polyurethane casting plastic that is lightweight, durable, and easily painted to match the original bone. The results are accurate 3D skull replicas for research and display.

One replica was used as a base into which each original bone was mounted in a manner that would allow for its removal for examination by researchers. Custom steel brackets were bent to cradle each individual piece, holding them in their correct anatomical positions without using adhesives or drilling holes into the bones. The replica areas of the "real bone" mount were painted to a similar brown color, making the finished piece aesthetic overall but clearly highlighting the original material compared to sections of reconstruction (Fig. 3).

Mounting and restoration was performed by Ben Meredith, Ethan Storrer, Jose Muñoz, and Seth Bourgeous during the spring of 2022. Upon completion of the mount, two large solid wooden crates were constructed. One held the steel and replica material, and the other was for packing of all of the original material. The packing was done using a custom spray-in-place foam system that allowed for a perfectly form fitting, reusable padding that protects the specimen during transport. Upon completion, the specimen was transported to Evolutionsmuseet, Knuthenborg, Maribo, Denmark *via* airfreight, where it was received by museum staff.

## MATERIALS AND METHODS

### Paleontological ethics statement

The specimen described here (EMK 0012) is in the publicly accessible, permanent repository of Evolutionsmuseet, Knuthenborg, Maribo, Denmark. Ownership title to EMK
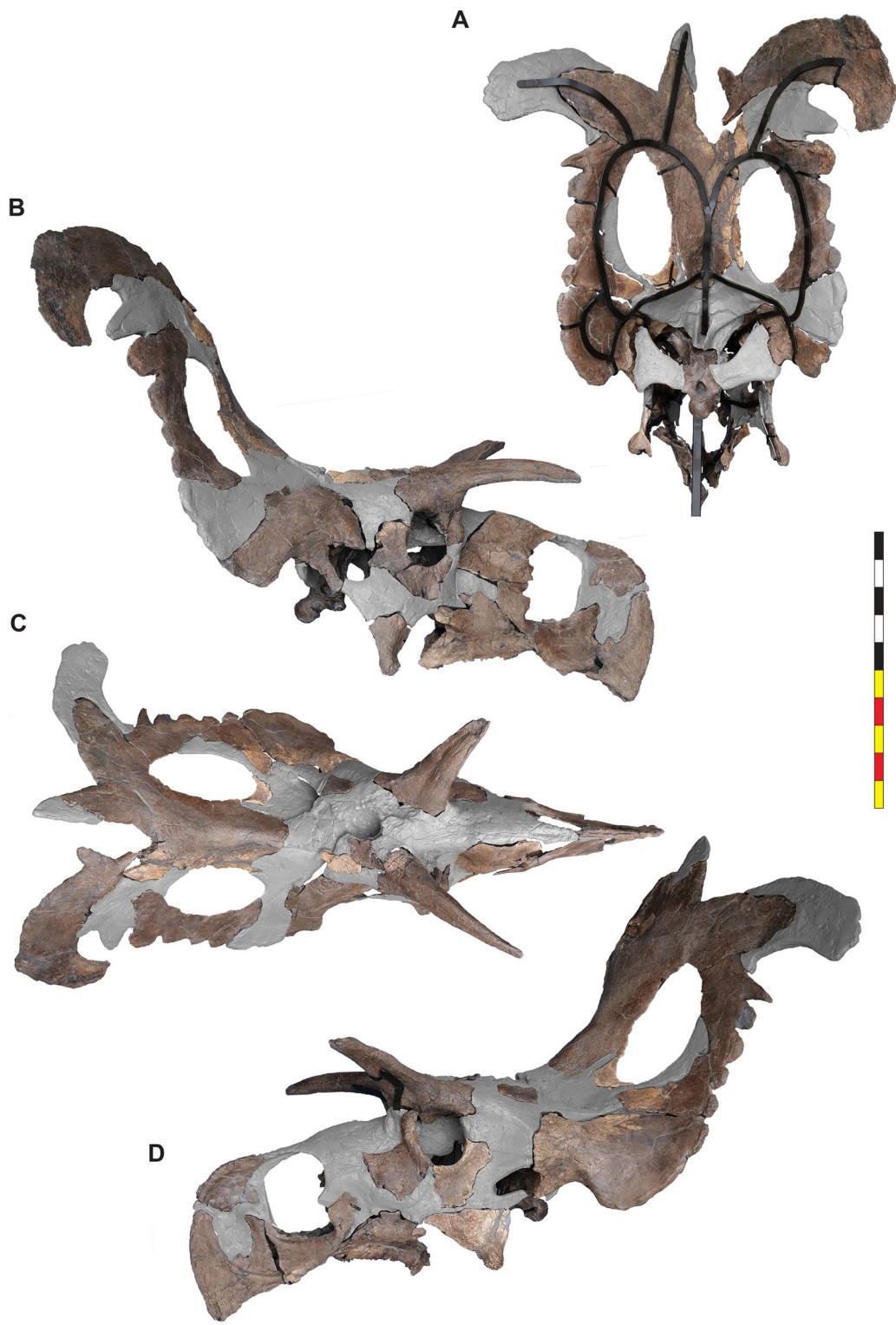

**Figure 3 Mounted skull of EMK 0012.** (A) Mounted skull in posterior view. (B) Mounted skull in right lateral view. (C) Mounted skull in dorsal view. (D) Mounted skull in left lateral view. Areas in gray are reconstructed. Minor changes from side to side and in the orbits are the result of post depositional deformation. Photos by Marcus Donivan and contain parallax. Scale bar equals 1 m.

0012 was transferred from the private landowner to Evolutionsmuseet, Knuthenborg. Casts of EMK 0012 are reposited as UMNH VP C-991 at the Natural History Museum of Utah, Salt Lake City, Utah, USA and as ROM 88670 at the Royal Ontario Museum, Toronto, Canada. Locality coordinates, notes, and diagrams associated with the specimen are available from the specific repository institutions as per institutional policy. All research was conducted subsequent to the specimen's acquisition by Evolutionsmuseet.

## Terminology

We employ traditional, or "Romerian," anatomical and directional terms over veterinary alternatives (*Wilson, 2006*) in order to be consistent with the vast majority of ceratopsid literature. For example, "anterior" and "posterior" are used as directional terms in lieu of the veterinary alternatives "rostral", "cranial", and "caudal", and human anatomical terms "inferior" and "superior". These terms are especially unsuited to descriptions of ceratopsians that possess a rostral bone and caudal vertebrae. English equivalents of standard Latin terms are used, and directional terms follow *Clark (1993)*.

While the terms ectonaris and endonaris have been previously used to describe ceratopsian anatomy to refer to the outer and inner openings of the nasal (*Witmer, 1997*, *2001*; *Sampson et al., 2010*, *2013*) and the terms "external" and "internal naris" could also be used, we to refer to these areas as the narial fossa (=ectonaris) and naris (=endonaris). We consider the opening of the nasal passage in anterior view to be the internal naris.

Major openings posterior to the orbit are referred to as dorsotemporal and laterotemporal fenestrae. Anatomical nomenclature for the sinuses at the roof of the skull are modified from *Farke (2006*, *2010)* to reflect the dorsocranial sinus complex. Anatomical nomenclature for marginal ossifications of the parietosquamosal frill follows the system first proposed by *Hatcher, Marsh & Lull (1907)* and more recently advocated by *Goodwin & Horner (2008)* and modified by *Loewen et al. (2010)*. Marginal ossifications on the squamosal and parietal of ceratopsids are referred to as "episquamosals" (es) and "epiparietals" (ep), respectively. As a group, we refer to these epiossifications as "marginal ossifications of the frill" in place of the anatomically erroneous nomenclature "epoccipitals." Where an epiossification crosses the squamosal-parietal contact, we refer to it as an "epiparietosquamosal marginal ossification" (eps). Epiossifications of the frill are numbered sequentially from the midline of the parietal; with a possible midline epiparietal (ep0) and epiparietals then sequentially numbered lateral from the midline (ep1–ep8); an epiparietosquamosal (eps) if present at the parietosquamosal suture, and episquamosals sequentially from posterior to anterior (es1 to es4 or es5). Raised bumps on the dorsal surface of the marginal parietal frill are termed dorsoparietal processes (dpp).

## Phylogenetic analysis

To assess the systematic position of EMK 0012, the specimen was coded in a matrix initiated by Scott Sampson and Catherine Forster in the 1990's and expanded by Mark Loewen and Andrew Farke during the 2000's and 2010's (*Forster & Sampson, 2002*; *Loewen et al., 2010*; *Sampson et al., 2010*; *Farke et al., 2011*; *Knapp et al., 2018*). Character scorings were based on firsthand observations of specimens. The character-taxon matrix

was assembled in Mesquite v.3.70 (*Maddison & Maddison, 2011*), and the matrix was analyzed using TNT v. 1.5 (*Goloboff, Farris & Nixon, 2008*; *Goloboff & Catalano, 2016*). Tree search using the parsimony criterion were implemented under the heuristic search option using tree bisection and reconnection (TBR) with 10,000 random addition sequence replicates. Zero length branches were collapsed if they lacked support under any of the most parsimonious reconstructions. *Hypsilophodon foxii* was designated the outgroup, and characters were run equally weighted, except for multistate Characters 1, 51, 70, 126, 130, 144, 170, 261, 262, 279, 336, and 339 which were considered ordered (additive). Character 90 regarding postorbital ornamentation in juveniles can be (but was not) excluded, as most taxa do not include immature specimens. The dataset consists of 377 characters (263 cranial, 61 postcranial, and 53 concerning frill-based ornamentation) and 86 taxa.

## Comparative material

We compared EMK 0012 with an exhaustive selection of ceratopsian taxa and accessed the ever-expanding literature focused specifically on ceratopsid dinosaurs. The authors have had the opportunity over the past 20 years to study firsthand and photograph nearly the complete range of marginocephalian materials collected globally. Where published illustrations and descriptions were used to supplement data obtained through direct observation, appropriate references are cited below.

Comparative material included the non-marginocephalian taxa *Hypsilophodon foxii* (NHM 28707; NHM 9560-1; and NHM R 2477) and *Lesothosaurus diagnosticus* (BMNH R8501; BMNH R11956; BMNH RU B17; and BMNH RU B23). Pachycephalosaurians included: *Stegoceras validum* (TMP 99.62.1; CMN 8816; TMP84.5.1; and UALVP 2), *Homalocephale calathocercos* (IGM 100/51), and *Prenocephale prenes* (Zpal MgD-I/104). Basalmost ceratopsians included: *Yinlong downsi* (IVPP V14530), *Hualianceratops wucaiwanensis* (IVPP V12722), *Xuanhuaceratops niei* (IVPP V18642), and *Chaoyangsaurus youngi* (IGCAGS V 371). Psittacosaurs included: *Psittacosaurus lujiatunensis* (IVPP V14341; IVPP V12617; LH PV1; JZMP-V-11; CAGS-IG-VD-004), *Psittacosaurus mongoliensis* (AMNH 6254), *Psittacosaurus sinensis* (IVPP V738; BNHM BPV149), *Psittacosaurus meileyingensis* (IVPP V7705), and *Psittacosaurus sibiricus* (PM TGU 16/4-20). Other basal ceratopsians included: *Mosaiceratops azumai* (ZMNH M8856), *Beg tsi* (IGM 100/3652), *Liaoceratops yanzigouensis* (CAGS-IG-VD-002; NMNH 58749; PMOL-AD00058; PMOL-AD00078; IVPP V12738; and IVPP V12633), *Aquilops americanus* (OMNH 34557), *Archaeoceratops yujingziensis* (CAGS-IG-VD-003), *Yamaceratops dorngobiensis* (IGM 100/1315), *Auroraceratops rugosus* (CAGS-IG-VD-001), and *Archaeoceratops oshimai* (IVPP V11114). Leptoceratopsids included: *Cerasinops hodgskissi* (MOR 300; USNM 13863), *Montanoceratops cerorhynchus* (AMNH 5464; AMNH 5244; MOR 542), *Udanoceratops tschizhovi* (PIN 3907/11), *Prenoceratops pieganensis* (MNHCM material; TCM material), *Zhuchengceratops inexpectus* (ZCDM V0015), and *Leptoceratops gracilis* (CMN 8887; CMN 8889). Derived non-ceratopsid taxa included: *Protoceratops hellenikorhinus* (IMM 95BM1/1; IMM 96BM1/4), *Protoceratops andrewsi* (AMNH 6251, 6408, 6414, 6418, 6425, 6429, 6430, 6438, 6441, 6443, 6444, 6447,

6449, 6451, 6466, 6473, 6477, 6480, 6483, 6485, 6486, 6487 6637, 6638; BMNH R6640; R10060; IGM 100-500, 100-502, 100-522, 100-581), *Protoceratops* sp. (IGM 100-1246), *Breviceratops kozlowskii* (Zpal MgD-I/116; Zpal MgD-I/117), *Bagaceratops rozhdestvenskyi* (Zpal MgD-I-126; ZPAL MgD-I/123; ZPAL MgD-I/124; ZPAL MgD-I/ 125; ZPAL MgD-I/127; ZPAL MgD-I/128; ZPAL MgD-I/129 (Czepinski, 2019)), *Ajkaceratops kozmai* (MTM V2009.192.1; MTM V2009.193.1; MTM V2009.194.1; MTM V2009. 195.1; MTM V2009.196.1), *Graciliceratops mongoliensis* (ZPal MgD-I/156), *Turanoceratops tardabilis* (CCMGE 251/12457), and *Zuniceratops christopheri* (MSM P2101; MSM P2107; MSM P 2110). Centrosaurine taxa included: *Diabloceratops eatoni* (UMNH VP 16699), *Machairoceratops cronusi* (UMNH VP 20550), *Crittendenceratops krzyzanowskii* (NMMNH P-34906), *Menefeeceratops sealeyi* (NMMNH P-25052), *Yehuecauhceratops mudei* (CPC 274), *Avaceratops lammersi type* (ANSP 15800), *Avaceratops sp.* (MOR 692 (*Ryan et al., 2017*)), *Avaceratops sp.* (CMN 8804 (*Ryan et al., 2017*)), *Nasutoceratops titusi* (UMNH VP 16800; UMNH VP 19466), *Xenoceratops foremostensis* (CMN 53282), *Lokiceratops rangiformis* (EMK 0012), *Albertaceratops nesmoi* (TMP 2001.26.01), *Medusaceratops lokii* (TMP 2002.69.1–10; TMP 2002.28–38; WDCB-MC-001; FDMJ-V-10; WDCB unnumbered specimens), *Wendiceratops pinhornensis* (TMP 2011.051.0009 and ~240 other TMP specimens from the South Side Ceratopsian bonebed), *Sinoceratops zhuchengensis* (ZCDM V0010; ZCDM V0011; ZCDM V0012), *Coronosaurus brinkmani* (TMP 2002.68.1), *Spinops sternbergorum* (NHMUKR16307; NHMUKR16308; NHMUKR16309), *Centrosaurus apertus* (CMN 348; CMN 8795; CMN 8798; UAL VP 11735), *Styracosaurus albertensis* (CMN 344), *Styracosaurus ovatus* (USNM 11869), *Stellasaurus ancellae* (MOR 492), *Einiosaurus procurvicornis* (MOR collection), Iddesleigh pachyrhinosaur TMP 2002.76.1 (*Ryan et al., 2010*), *Achelousaurus horneri* (MOR 485), *Pachyrhinosaurus lakustai* (TMP 86.55.285; TMP 87.55.156; TMP 89.55.1234), *Pachyrhinosaurus perotorum* (DMNH 21200; DMNH 22558), *Pachyrhinosaurus canadensis* (CMN 8860, CMN 8866, CMN 8867, CMN 9485, CMN 10645, CMN 10663, CMN 21863, CMN 21864, TMP 82.52.1). Chasmosaurine taxa included: *Mercuriceratops gemini* (UALVP 54559), *Regaliceratops peterhewsi* (RTMP 2005.55.1), *Kosmoceratops richardsoni* (UMNH VP 17000; UMNH VP 16878), *Vagaceratops irvinensis* (NMC 41357; TMP 87.45.1; TMP 98.102.8), *Spiclypeus shipporum* (CMN 58071), *Chasmosaurus belli* (AMNH 5402; BMNH R4948; CMN 2245; ROM 839; ROM 843; YPM 002016), *Mojoceratops kaiseni* (AMNH 5401; AMNH 5656; TMP 79.11.147; TMP 81.19.175; TMP 83.25.1), *Agujaceratops mavericus* (TMM 43098-1), *Agujaceratops mariscalensis* (TMM 46500-1; UTEP P37.7.065; UTEP P.3737.046), *Chasmosaurus russelli* (CMN 8800; CMN 8801; TMP 2013.19.38), *Utahceratops gettyi* (UMNH VP 12198; UMNH VP 16671; UMNH VP 16784), *Pentaceratops sternbergii* (AMNH 1625; AMNH 6325; KUVP 16100; MNA P1.1747; PMU R200), *Anchiceratops ornatus* (AMNH 5251; CMN 8535; TMP 83.01.01), *Arrhinoceratops brachyops* (ROM 796), *Eotriceratops xerinsularis* (TMP 2002.57.7), *Torosaurus latus* (AMNH 5116; ANSP 15192; EM P16.1; MOR 981; MOR 1122; MPM VP 6841; YPM 001830), *Torosaurus utahensis* (USNM 15583), *Triceratops prorsus* (LACM 27428; YPM 001822), *Triceratops horridus* (AMNH 5116; YPM 001820). Some comparative taxa that were considered but

not included in the phylogenetic analysis due to poorly diagnosable holotype material, and/ or a very large proportion of missing data: *Helioceratops brachygnathus* (JLUM L0204-Y-3), *Koreaceratops hwaseongensis* (KIGAM VP 200801), *Gryphoceratops morrisoni* (ROM 56635), *Unescoceratops koppelhusi* (TMP 95.12.6), *Furcatoceratops elucidans* (NSM PV 24660), the *Agujaceratops* sp. Terlingua exemplar (TMM 45922), *Terminocavus sealeyi* (NMMNH VP 27468), *Bisticeratops froeseorum* (NMMNH P-500000); *Navajoceratops sullivani* (SMP VP 1500), *Titanoceratops ouranos* (OMNH 10165); *Coahuilaceratops magnaquerna* (CPC 276; CPC 277), *Bravoceratops polyphemus* (TMM 46015-1), *Judiceratops tigris*, *Sierraceratops turneri* (NMNNH P-76870), *Ojoceratops fowleri* (SMP VP-1865), and *Nedoceratops hatcheri* (USNM 2412).

## Nomenclatural acts

The electronic version of this article in Portable Document Format (PDF) will represent a published work according to the International Commission on Zoological Nomenclature (ICZN), and hence the new names contained in the electronic version are effectively published under that Code from the electronic edition alone. This published work and the nomenclatural acts it contains have been registered in ZooBank. The ZooBank Life Science Identifiers (LSIDs) can be resolved and the associated information viewed through any standard web browser by appending the LSID to the prefix "http://zoobank.org/". The LSID for this publication is: urn:lsid:zoobank.org:pub:77A46B79-9BA1-4764-9AF6-14C69C2B8C8F. The LSID for *Lokiceratops* is: urn:lsid:zoobank.org:act:4640DFB2-63D2-483A-93ED-4EF405285CAC. The LSID for *Lokiceratops rangiformis* is: urn:lsid:zoobank.org:act:548AA668-EE62-49DA-8CA2-939A00223B92. The online version of this work is archived and available from the following digital repositories: CLOCKSS, Zenodo and PubMed Central.

## RESULTS

### Systematic Paleontology

Dinosauria *Owen, 1842*; *sensu* *Padian & May, 1993*
Ornithischia *Seeley, 1887*; *sensu* *Sereno, 1998*
Ceratopsia *Marsh, 1890*; *sensu* *Dodson, 1997*
Ceratopsidae *Marsh, 1888*; *sensu* *Sereno, 1998*
Centrosaurinae *Lambe, 1915*; *sensu* *Dodson, Forster & Sampson, 2004*

**Albertaceratopsini** clade nov.
urn:lsid:zoobank.org:act:4640DFB2-63D2-483A-93ED-4EF405285CAC

**Diagnosis**—Albertaceratopsini is defined as a stem-based clade that consists of all taxa more closely related to *Albertaceratops nesmoi* than to *Centrosaurus apertus*.

***Lokiceratops*** gen. nov.
urn:lsid:zoobank.org:act:4640DFB2-63D2-483A-93ED-4EF405285CAC

**Diagnosis**—Monotypic, same as for species.

*Lokiceratops rangiformis* gen. et sp. nov.
urn:lsid:zoobank.org:act:548AA668-EE62-49DA-8CA2-939A00223B92

**Etymology**—The generic name refers to the god Loki from Norse mythology, and *ceratops*, (Greek) meaning "horned face." The species name refers to the bilateral asymmetry of frill ornamentations, similar to the asymmetry in antlers of the reindeer/caribou genus *Rangifer*.

**Holotype**—EMK 0012 is an associated, disarticulated skull and partial skeleton (Figs. 2–4). The skull is represented by the rostral, premaxillae, maxillae, nasals, lacrimals, jugals, frontals, palpebrals, postorbitals, squamosals, and parietals. It includes the left pterygoid and a partial braincase. Postcranial elements include a cervical vertebra; the right scapula and coracoid; both ischia and the sacrum with attached sacrodorsals and sacrocaudals; an anterior free caudal vertebra; and a chevron from the proximal tail. EMK 0012 is reposited at the Evolutionsmuseet, Knuthenborg, Maribo, Denmark.

**Holotype Locality**—EMK 0012 was recovered from the Loki Quarry in Kennedy Coulee, south of the Milk River in Hill County, northern Montana (Fig. 1). The quarry is 3.6 km from the Montana-Alberta border and 922 m above sea level. Exact coordinates are available at the Evolutionsmuseet, Knuthenborg, Maribo, Denmark, and the Natural History Museum of Utah, United States of America.

**Holotype Horizon**—EMK 0012 was recovered from lower Judith River Formation horizons that correlate to the McClelland Ferry Member to the south, and to the lower part of the Oldman Formation of southern Alberta, 3.6 km to the north. EMK 0012 was located 11.4 m above the Marker A Coal equivalent to the Taber Coal Zone (at the top of the Foremost Formation) and laterally equivalent to the Herronton Sandstone Zone near the base of the Oldman Formation 3.6 km to the north in Alberta.

**Age**—High-precision U–Pb analyses of zircons by the CA-ID-TIMS method in a bentonite within the Marker A Coal (KC061517-1; 11.4 m below the Loki Quarry) date to 78.549 ± 0.024 Ma (*Ramezani et al., 2022*). Using the median Bayesian age model developed by *Ramezani et al. (2022)* for the stratigraphic position of the Loki Quarry recovers a date of roughly 78.1 Ma, with a lower modeled bound of 78.38 Ma and an upper modeled bound of 77.18 Ma.

**Diagnosis**—*Lokiceratops rangiformis* is an albertaceratopsin centrosaurine ceratopsid distinguished from other centrosaurines by the following autapomorphies: presence of unadorned nasal; elongate, uncurved ep1 epiossification directed in plane of frill along posterior margin of parietosquamosal frill; hypertrophied, lateral curving epiparietal ep2 directed in plane of frill. The hypertrophied ep2 is relatively larger than any other parietal epiossification within Centrosaurinae. Both ischia are distinctly kinked distally at about two-thirds of the length of the shaft, at the point where the two ischia contact medially. Postorbital horncore bases are deeply excavated by sinuses penetrating a distance

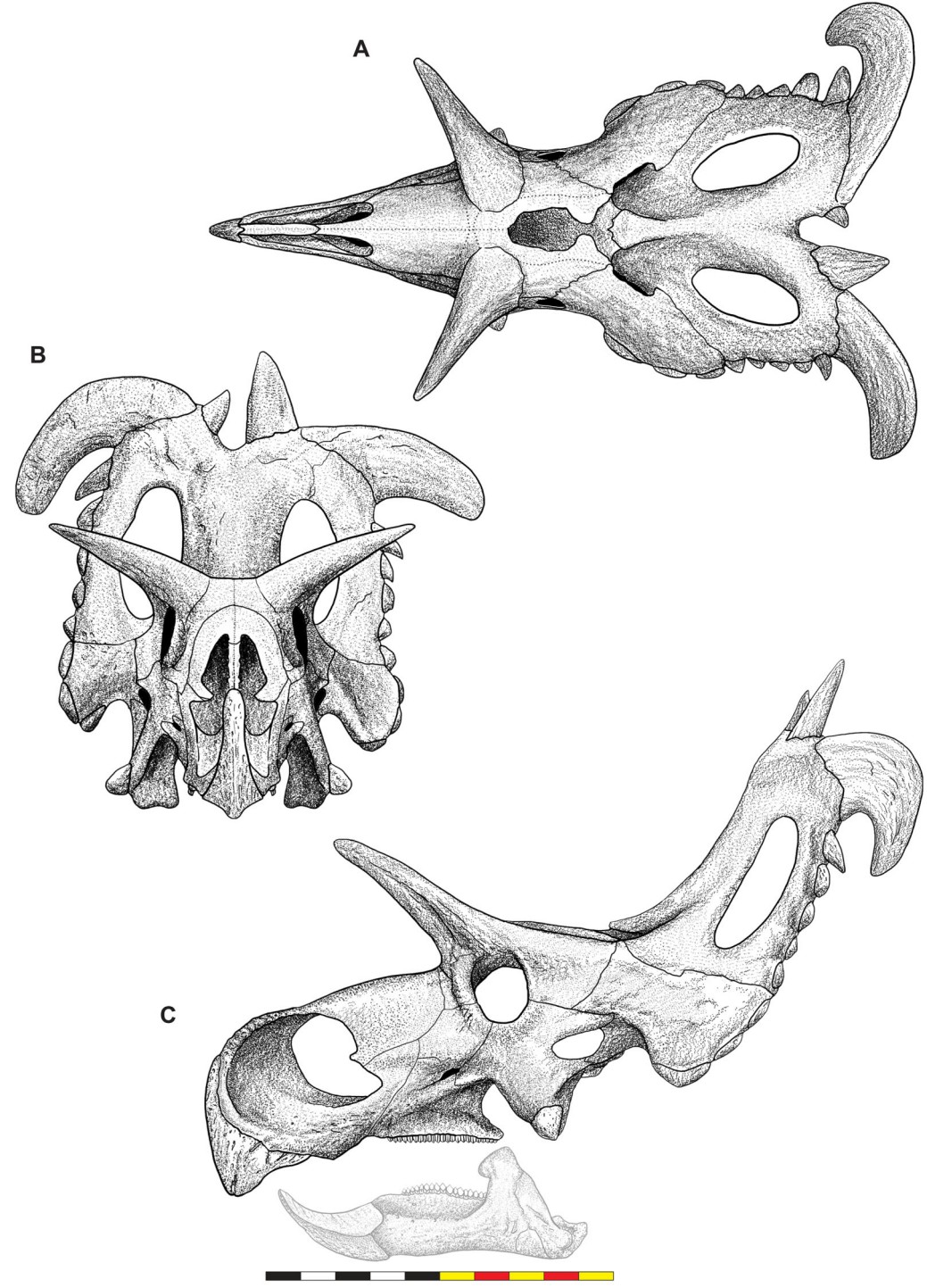

**Figure 4 Skull of *Lokiceratops rangiformis* n. gen et n. sp. (EMK 0012).** (A) Skull of *Lokiceratops rangiformis* in dorsal view. (B) Skull reconstruction in anterior view. (C) Skull reconstruction in lateral view interpreted from both sides. Reconstructions are based on 3D surface scans with deformation and parallax removed. The mandible was not found with EMK 0012. Stippled artwork by Sergey Krasovskiy. Scale bar equals 1 m.

equivalent to the orbital diameter of each horncore, to an extent unobserved in other long horned centrosaurs.

**Differentia**—*Lokiceratops rangiformis* differs from *Zuniceratops* and all known chasmosaurines in possessing an abbreviated, fan-shaped squamosal typical of most centrosaurines. Differs from *Zuniceratops* and all known centrosaurines in the distinct kink in the ischium.

*Medusaceratops lokii* differs from the stratigraphically proximate *Lokiceratops rangiformis* in a number of key features including: presence of distinct nasal ornamentation; limited extent of postorbital pneumaticity; presence of multiple raised undulations on midline ramus of parietal between parietal fenestrae; lack of a narrow, medially restricted embayment on the midline of the posterior edge of parietal; reduced, rather than elongate, posteriorly directed ep1 epiossifications along posterior margin of parietosquamosal frill; relative length of largest laterally curving epiparietal ep2; and presence of five bilateral epiparietals (six or seven in *Lokiceratops*).

*Albertaceratops nesmoi* differs from the stratigraphically proximate, but possibly slightly younger *Lokiceratops rangiformis*, in key features including: presence of nasal ornamentation; presence of four episquamosals (three in *Lokiceratops*); presence of multiple raised undulations on midline ramus of parietal between parietal fenestrae; posterolaterally curving blade-like ep1s, rather than elongate posteriorly directed, ep1 epiossifications along posterior margin of parietosquamosal frill; length of large laterally curving epiparietal ep1; presence of dorsally curled spikes at epiparietal positions ep2–ep5; and presence of five epiparietal positions (six or seven in *Lokiceratops*).

*Wendiceratops pinhornensis* differs from the likely stratigraphically equivalent *Lokiceratops rangiformis* in key features including: presence of nasal ornamentation; lack of medially restricted embayment on the midline of the posterior edge of the parietal; presence of five dorsally recurved epiparietals; lack of hypertrophied laterally curving epiparietal; and presence of five bilateral epiparietals (six or seven in *Lokiceratops*).

*Judiceratops tigris*, a fragmentary chasmosaurine, differs from the stratigraphically similar *Lokiceratops rangiformis* in its elongated, sickle-shaped squamosal; lack of medially restricted midline embayment of posterior parietal bar; and lack of elongated epiparietals on its parietal (*Campbell, 2015*).

## DESCRIPTION AND COMPARATIVE ANATOMY

### Present condition of the skull

The skull of *Lokiceratops rangiformis* (EMK 0012) is exhibited at Evolutionsmuseet, Knuthenborg, Maribo, Denmark. The skull is presently reconstructed into a steel supported mount (Fig. 3) with each individual bone articulated into a 3D cast skull reconstruction. Each bone is removable from the mount for study. A cranial osteograph illustrates the missing parts of each element of the skull (Fig. 5). As a result of elements being removable, there are slight gaps in the real bone mount to accommodate removal. For this reason, measurements were taken from the reconstructed cast skull, which more accurately reflects the actual dimensions of the specimen. These data are presented in

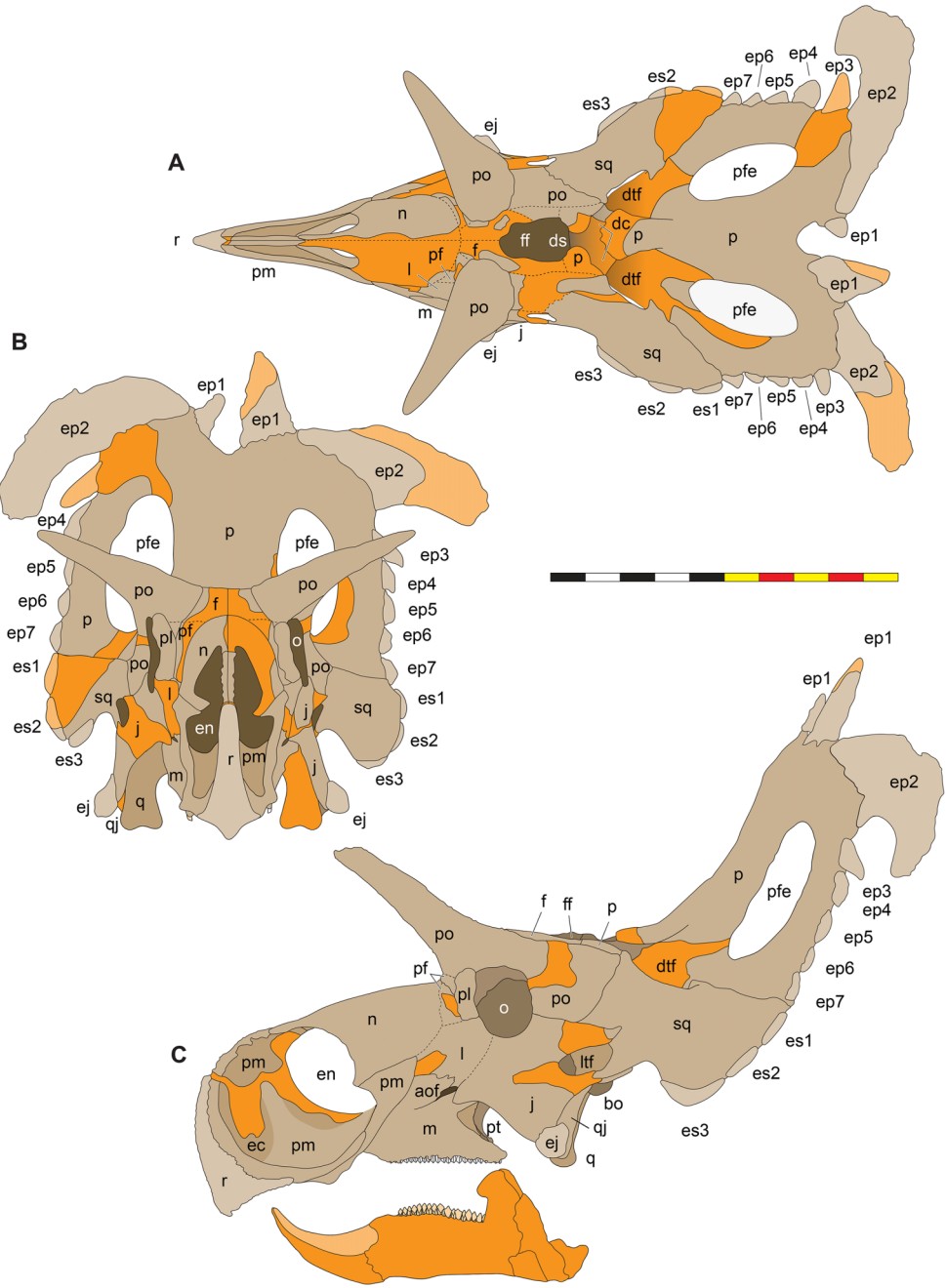

**Figure 5 Sutural interpretation and completeness of *Lokiceratops rangiformis* n. gen et n. sp. (EMK 0012).** (A) Sutural relationships between the cranial elements of *Lokiceratops rangiformis* in dorsal view. (B) Skull reconstruction and sutures in anterior view. (C) Skull reconstruction and sutures in lateral view interpreted from both sides. Reconstructions are based on 3D surface scans with deformation and parallax removed. Orange shades represent missing elements or portions of elements. Stippled sutures are inferred in places where sutures are obliterated by fusion upon maturity. *Osteological abbreviations*: aof, antorbital fenestra; bo, basioccipital; dc, dorsotemporal channels; ds, dorsocranial sinus; dtf, dorso-temporal fenestra; ej, epijugal; en, external naris; ep1–ep7, epiparietals 1–7; es1–es3, episquamosals 1–3; f, frontal; ff, frontal fontanelle; j, jugal; l, lacrimal; ltf, laterotemporal fenestra; m, maxilla; o, orbit; pa, parietal; pf, prefrontal; pfe, parietal fenestra; pl, palpebral; pm, premaxilla; po, postorbital; q, quadrate; qj, quadratojugal; r, rostral; sq, squamosal. Scale bar equals 1 m.

Table 1, and cranial measurements are explained in Fig. 6. The measurements in Table 1 and Fig. 6 are inspired by the measurement table of *Mallon et al. (2016)*. The skull outlines in anterior, dorsal, and ventral views were reconstructed using 3D surface scans and lack parallax.

## General cranial morphology

The narial region of *Lokiceratops rangiformis* (EMK 0012) closely resembles other centrosaurines in being roughly subcircular, with a well-developed premaxillary septum, a ventrally projecting ventral angle, and a narial spine on the posterior margin of the external naris (Figs. 3–6). There is no evidence of nasal ornamentation either as a change in texture from the ventral surface of the nasal to the dorsal surface, or in the shape of the dorsal surface of the nasals. The anterior process of the nasal lacks the rugosity present on the rostral and dorsal premaxilla. The orbits bear dorsally elongated, anterolaterally oriented horncores and a well-developed antorbital buttress formed by the prefrontal, palpebral, and lacrimal as in most basal centrosaurines, and unlike the reduced horncores of eucentrosaurines. The suborbital region is similar to that of all centrosaurines. The parietosquamosal frill is elongated compared to *Centrosaurus*, with typical fan-shaped, stepped squamosals, and elongate, fenestrated parietals. Epiossifications include short epijugal horns, three episquamosals, and six or seven epiparietals on each side.

## Major cranial fenestrae, foramina, fossae, and passageways

**Nasal Vestibule**—The nasal vestibule is the outermost anterior expression of the nasal cavity and is made up of the narial fossa and external naris. This area is small in most dinosaurs including the basal ceratopsian *Yinlong downsi* (*Xu et al., 2006*; *Han et al., 2016*), but is larger in many ceratopsians, including psittacosaurs (*Sereno, 2010*; *You et al., 2005*); leptoceratopsids (*Brown & Schlaikjer, 1940*; *Chinnery, 2004*), and protoceratopsids (*Czepiński, 2020*); the nasal vestibule is hypertrophied in *Zuniceratops* and all ceratopsids.

**Internal Naris and Nasal Cavity**—The nasal cavity proper is the main chamber of the nasal passage, likely containing both olfactory and respiratory epithelia. It extends between the nasal vestibule anteriorly and the nasopharynx posteriorly, which in turn opens into the pharynx *via* the choanae.

**External Narial Fossa**—The external narial fossa (Fig. 7) represents the maximum inferred extent of soft-tissue associated with the narial region, expressed on the lateral surface of the premaxilla and the anterior surface of the nasal. The overall shape of the external narial fossa in lateral view is hemicircular, as in all centrosaurine ceratopsids. The anterodorsal, anterior, and ventral portions of the external narial fossa extend over the lateral surface of the premaxilla. The posteroventral portion of the external narial fossa lies on the dorsal surface of the posteroventral process of the premaxilla, transitioning posteriorly from the lateral surface of the premaxilla to the anterior edge of the premaxillary contribution to the narial spine. The posterior part of the external narial fossa is formed by the nasal contribution to the narial spine, and the dorsal portion of the external narial fossa is formed by the anteroventral surface of the nasal. EMK 0012 is

**Table 1 Measurements of the holotype of *Lokiceratops rangiformis* n. gen et n. sp. (EMK 0012).** Numbers in the left column correspond to those in Fig. 6. Measurements build on those of *Mallon et al. (2016)*. All measurements excluding angles for *Lokiceratops rangiformis* are expressed in mm. An asterisk (*) denotes reconstructed/estimated value.

| | | Left | Right |
|---|---|---|---|
| 1 | Postorbital horncore length (rectilinear) from dorsal rim of orbit to apex | 374 | 400 |
| 2 | Postorbital horncore length (curvilinear) from dorsal rim of orbit to apex | 382 | 432 |
| 3 | Postorbital horncore anteroposterior length at base | 170 | 152* |
| 4 | Postorbital horncore mediolateral width at base | 178* | 137 |
| 5 | Postorbital horncore circumference about base | 397* | 451 |
| 6 | Postorbital horncore angle lateral to saggital plane in dorsal view | NA | NA |
| 7 | Nasal horncore height from base to apex (excluding nasal bridge) | NA | NA |
| 8 | Nasal horncore transverse width at base | NA | NA |
| 9 | Nasal horncore anteroposterior length at base | NA | NA |
| 10 | Rostral-orbit length | 735 | 731 |
| 11 | Posterior margin of external naris-orbit length | 252 | 280* |
| 12 | Rostral-posterior margin of nasal horncore | NA | NA |
| 13 | Epijugal-orbit length (along surface) | 297 | 289 |
| 14 | Lateral temporal fenestra-orbit length | 116 | 118 |
| 15 | Orbit anteroposterior length | 120 | 132 |
| 16 | Orbit dorsoventral height | 150 | 155 |
| 17 | Maximum orbit diameter | 160 | 159 |
| 18 | Minimum distance from otic notch to medial margin of squamosal | 202 | 215 |
| 19 | Minimum distance from lateral margin of anteriormost episquamosal to medial margin of squamosal | 268 | 281 |
| 20 | Rostral-epijugal (rectilinear) | 792 | 789 |
| 21 | Rostral-posterior edge of maxillary tooth row | 631* | 649 |
| 22 | Basal skull length (rostral-occipital condyle) | 948 | 948 |
| 23 | Maximum anteroposterior length of skull (rostral-posterior parietal, excluding epiossifications) | 1,730 | 1,765 |
| 24 | Distance between orbits | 340 | 340 |
| 25 | Length of squamosal from otic notch to its distal end | 353 | 370* |
| 25a | Maximum length of squamosal from apex of anteriormost episquamosal to distal end | 331 | 329* |
| 26 | Jugal notch-parietal fenestra (curvilinear) | 270 | 229 |
| 27 | Epijugal-squamosal (across jugal notch) | 290 | 410* |
| 28 | Posterior margin of postorbital horncores (approximate location of posterior margin of frontoparietal fossa) to posterior edge of medial parietal bar (curvilinear) | 854 | 854 |
| 29 | Maximum transverse width of frill (at mid-length, between episquamosals) | 757 | 757 |
| 30 | Maximum length of parietal fenestra | 348 | 340 |
| 31 | Maximum width of parietal fenestra | 160 | 185 |
| 32 | Maximum skull length, rostral-epiparietosquamosal | 1,920 | 1,990 |
| 33 | Otic notch (from squamosal lateral corner) to apex of epiparietal | 830 | 845 |
| 33a | Otic notch (from squamosal lateral corner) to posterior margin of parietal (not including epiparietals) | 930* | 985 |
| 34 | Occipital condyle- absolute posterior part of frill | 1,060* | 1,210 |
| 35 | Number of episquamosals | 3 | 3* |
| 36 | Number of epiparietals | 7 | 7* |
| 37 | Dentary length, from anterior margin to posterior edge of coronoid base | NA | NA |

| Table 1 (continued) | | Left | Right |
|---|---|---|---|
| 38 | Dentary height at mid-tooth row length | NA | NA |
| 39 | Dentary height at coronoid process | NA | NA |
| 40 | Ectonaris length | 360 | 374 |
| 41 | Ectonaris height | 360* | 350 |
| 42 | Snout height at ventral angle | 330 | 418 |
| 43 | Width between postorbital horn tips | 650 | 680* |
| 44 | Width between epijugals | 534 | 534 |
| 45 | Otic notch to posterior margin of frill (excluding epiossifications) | 820 | 850* |
| 46 | Width between parietal fenestra (absolute width at midpoint of fenestra) | 220 | 220 |
| 47 | Preorbital face height from tooth row to top of face just in front of palpebral | 390 | 435 |
| 48 | Curvalinear distance across frill between parietosquamosal contacts across midline of frill | 940 | 940 |

missing a few millimeters of the contact between the dorsal portion of the premaxilla and the anterodorsal process of the nasal, but as in other centrosaurines, the external narial fossa likely transitioned from the premaxilla directly onto the lateral edge of the anterior process of the nasal. The round overall shape of the external narial fossa in *Lokiceratops* is similar to the condition in all centrosaurines and contrasts with the elongated, oval external narial fossa in *Zuniceratops* and all chasmosaurines.

**External Naris**—The external narial opening (Fig. 7) is formed by the open space between the incipient narial flange along the posterior edge of the narial septum, the posteroventral process of the premaxilla, and the narial spine and anterodorsal process of the nasal. The external naris forms an elliptical "B" shape in lateral view, with the long axis oriented roughly at 15° from vertical. The dorsal end of the external narial fossa extends almost to the dorsal surface of the external narial fossa similar to the condition in *Diabloceratops*. The external naris forms about 35% of the area of the entire external narial fossa. Among basal centrosaurines, the external naris of *Diabloceratops* comprises around 45% of the external narial fossa, compared to the condition in *Avaceratops lammersi*, *Nasutoceratops*, *Centrosaurus*, *Styracosaurus albertensis*, *Einiosaurus*, *Achaeolosaurus*, and the Iddesleigh pachyrhinosaur in which the external naris only makes up 25% or less of the external narial fossa (mainly because the external naris is ventrally displaced from the dorsal premaxilla contact with the nasal in these specimens). In *Pachyrhinosaurus canadensis*, *P. lakustai*, and *P. perotorum*, the external naris makes up less than 20% of the external narial fossa.

**Oral Vault**—The anterior portion of the oral vault is formed in the ventral space between the ventral rami of the rostral and the cutting surface of the ventral premaxillae and is dorsally bound by the palatal shelf of the premaxilla (Fig. 7). The posterior portion of the oral vault is formed between the tooth bearing portions of the maxillae and dorsally bounded by the vomers and palatines.

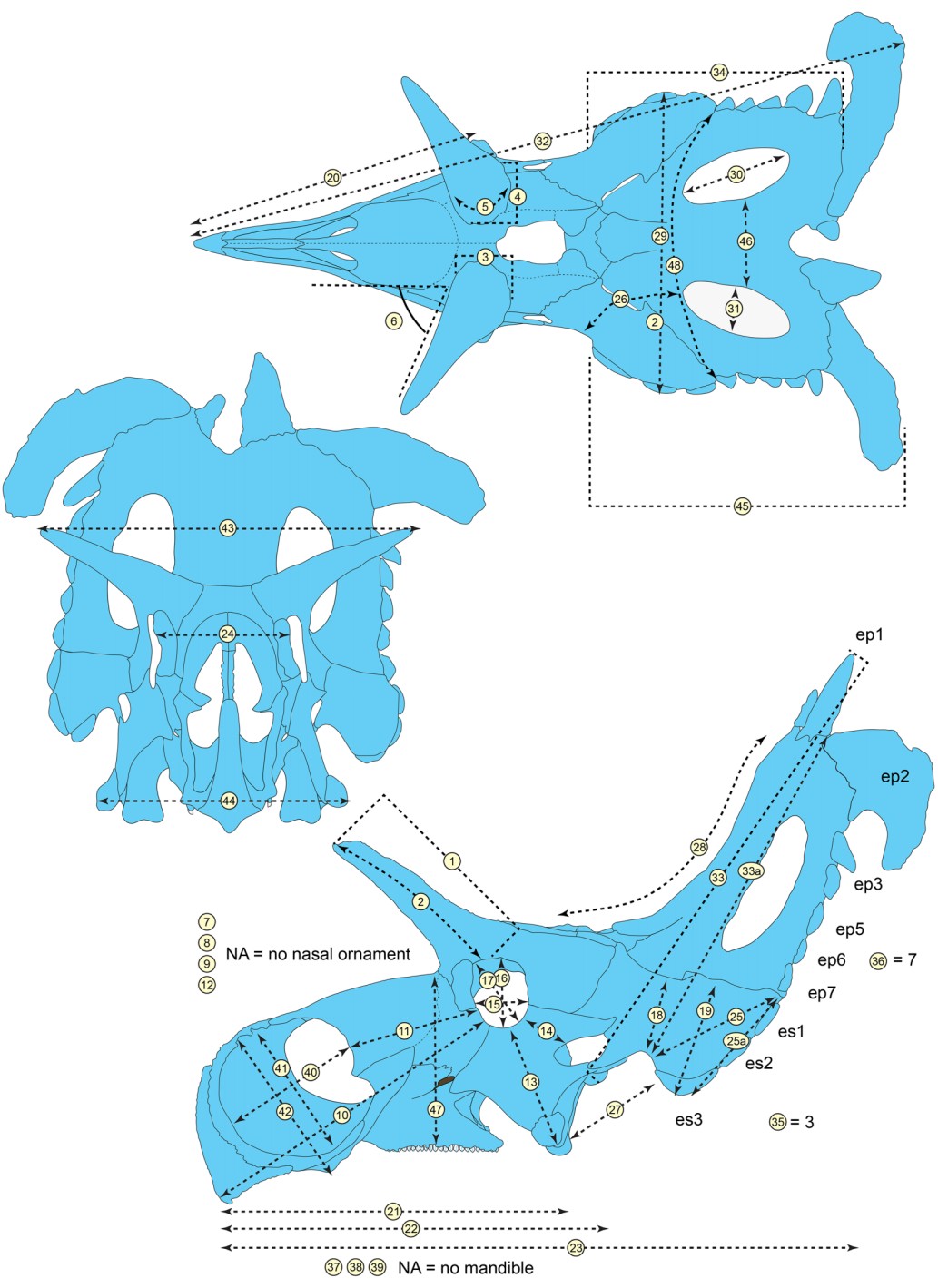

**Figure 6 Skull measurements of *Lokiceratops rangiformis* n. gen et n. sp. (EMK 0012).** See Table 1 for corresponding measurement descriptions and values.

**Buccal Vault**—The medially inset tooth bearing portions of the maxillae presumably formed pouches for cheeks between the anterior maxillary diastema, the dorsal ridge confluent with the anterior process of the jugal, the lateral ridge of the dentary ventrally, and the coronoid process of the dentary and its' adductor musculature posteriorly (Fig. 7).

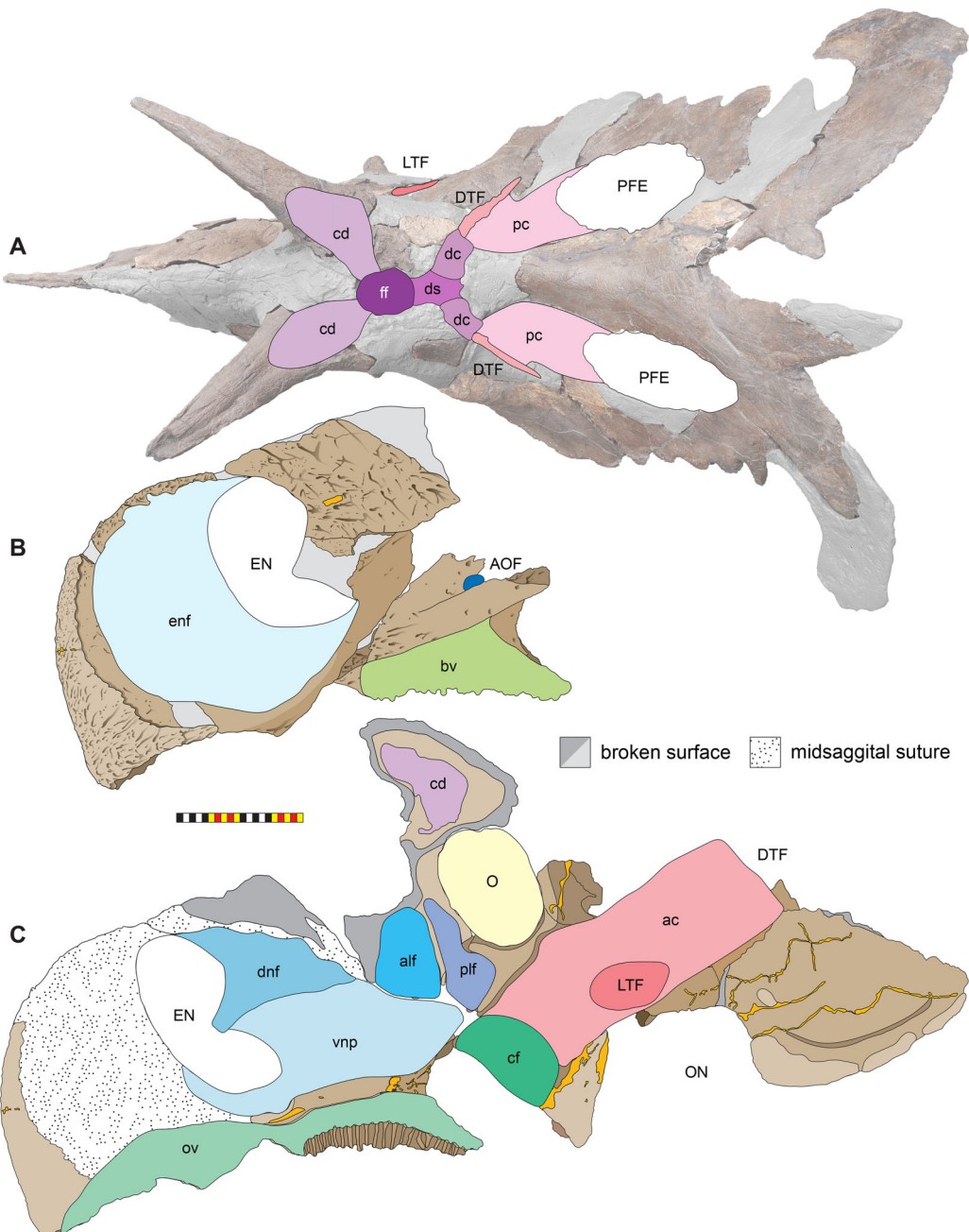

**Figure 7 Major openings, pneumaticity and internal passageways in the skull of *Lokiceratops rangiformis* n. gen et n. sp. (EMK 0012).** (A) Dorsocranial sinuses, channels, and openings on the dorsal surface of the skull. (B) Interpretative line drawings of major external features of the narial region in lateral view. (C) Interpretative line drawings of internal passageways and openings in medial view. Dark gray indicates broken surfaces. Lighter grays indicate outlines interpreted from the preserved material. Orange denotes putty filling cracks. *Osteological abbreviations*: ac, adductor chamber; alf, anterior lacrimal fossa; AOF, antorbital fenestra; bv, buccal vault; cd, cornual diverticulum; cf, coronoid fossa; dc, dorsotemporal channels; dns, dorsal nasal fossa; ds, dorsocranial sinus; DTF, dorsotemporal fenestra; EN, external naris; enf, external narial fossa; ff, frontal fontanelle; LTF, laterotemporal fenestra; O, orbit; ON, otic notch; ov, oral vault; plf, posterior lacrimal fossa; pc, parietal channel; vnp, ventral nasal passage. Photo A by Marcus Donivan. Scale bar equals 20 cm.

It is likely that a "cheek" muscle *M. adductor mandibulae externus* originated on the maxillary ridge and inserted on the lateral ridge of the dentary (*Nabavizadeh, 2020*). The buccal region is similar in all centrosaurines.

**Coronoid Fossa**—As in all other ceratopsids, the jugal is ventrolaterally expanded over the coronoid process of the mandible, creating a slot-like adductor chamber between the posterodorsal margin of the posterior process of the maxilla and the jugal. Here, the jugal bears a smooth fossa on its medial surface, extending dorsally to the ventral margin of the orbit and anterodorsally to a ridge separating it from the posterior lacrimal fossa (Fig. 7). This fossa was presumably for accommodation of the coronoid process during occlusion and passage of adductor musculature inserting on the coronoid process of the mandible.

**Antorbital Fenestra**—As in other ceratopsids, the antorbital fenestra consists of a small, slot-like opening in the posterior rostrum, bordered anteriorly, anterodorsally, and ventrally by the maxilla, posteroventrally by the jugal, and posterodorsally by the lacrimal (Fig. 7). A groove on the bifurcated ascending process of the maxilla extends anteroventrally from the contacts for the anteroventral process of the jugal and the anteroventral margin of the lacrimal. The jugal may contribute to a small part of the posteroventral margin of the antorbital fenestra and then extends posterodorsally to form the base of the orbit. The small slit-shaped nature of the antorbital fenestra is similar to the condition in all centrosaurines including *Diabloceratops*. *Lokiceratops* lacks the accessory antorbial fenestra between the premaxilla, nasal, and maxilla present in *Bagaceratops*, *Zuniceratops*, and *Diabloceratops*.

**Orbit**—The external margins of the orbit are formed by the lacrimal and palpebral anteriorly, the jugal ventrally, and the postorbital dorsally and posteriorly (Figs. 3–7). The lacrimal and palpebral form the antorbital buttress of the anterior portion of the orbit, elevated substantially from the surface of the rostrum. The jugal and postorbital form the ventral and posterior portions of the orbit, with its rim being moderately expressed laterally. Dorsally, the postorbital ornamentation is confluent with the margin of the orbit. The overall shape of the orbit is round as in most centrosaurines, but in contrast with the ovoid orbit of some chasmosaurines. The orbits are parallel to each other and laterally directed, implying no anteriorly overlapping field of vision. The parasphenoid would have been visible in the posterior part of the orbit. The orbit is similar to most centrosaurines, but differs from *Sinoceratops*, *Coronosaurus*, *Einiosaurus*, *Achelousaurus*, the Iddesleigh pachyrhinosaur, and *Pachyrhinosaurus* in the presence of a well-developed antorbital buttress.

**Adductor Chamber**—The adductor chamber housed the jaw closing muscles that originate on and the around the dorsotemporal fenestra and pass deep to the laterotemporal fenestra and the ventral bar of the laterotemporal fenestra and the medial surface of the jugal to insert on the coronoid process of the dentary (Fig. 7). This chamber, medial to the paroccipital groove on the squamosal housed the adductor muscles *M. adductor mandibulae externus profundus* and *M. adductor mandibulae externus*

*medialis* (*Holliday et al., 2019*) which inserted on the coronoid process of the dentary. The adductor chamber is similar in all centrosaurines where it is visible (*i.e.*, *Diabloceratops*, *Centrosaurus*, the Iddesleigh pachyrhinosaur, *Pachyrhinosaurus canadensis*).

**Laterotemporal Fenestra**—The laterotemporal fenestra (Fig. 7) is ovoid, with its long axis oriented anteroventrally. The laterotemporal fenestra is bordered by the jugal anteriorly, dorsally, and anteroventrally, and by the squamosal posteriorly and posteroventrally. While the anterior portion of the fenestra is not preserved in EMK 0012, its shape can be inferred from the shape of the jugals and the presence of the posterior jugal suture on the squamosal. Both squamosals preserve the articular facet at the posterodorsal corner of the fenestra for articulation to the posterodorsal process of the jugal. The left squamosal preserves the articulation for the posteroventral process of the jugal. The right lower bar of the laterotemporal fenestra preserves the tip of the posteroventral process of the jugal. The postorbital and quadratojugal are excluded from the laterotemporal fenestra as in all centrosaurines. The laterotemporal fenestra differs in shape across Centrosaurinae from subround in *Diabloceratops* to an anteroposteriorly elongate oval *in Lokiceratops*, *Albertaceratops*, *Centrosaurus*, *Styracosaurus albertensis*, to the tiny round opening in *Einiosaurus*, the Iddesleigh pachyrhinosaur, and *Pachyrhinosaurus lakustai*.

**Dorsotemporal Fenestra**—The dorsotemporal fenestra (Fig. 7) is the dorsal opening in the skull posterior to the orbit, bordered by the parietal anteromedially and posteriorly, and by the squamosal laterally and anteriorly. In dorsal view, the dorsotemporal fenestra forms an elongated, ovoid slot bordered by the parietal medially and the squamosal laterally. Medially, a channel in the dorsal surface of the anterior parietal leads into the posterior chamber of the dorsocranial sinus, posterior to the frontal fontanelle. The dorsotemporal fenestrae of *Lokiceratops* are typical for centrosaurines, but are most similar in the shape of the stepped lateral margin to *Centrosaurus*, *Styracosaurus*, *Einiosaurus*, *Achelousaurus*, and *Pachyrhinosaurus*. The step is more pronounced than the low-step present in *Diabloceratops*, *Machairoceratops*, *Avaceratops*, and JRF 63 from the Judith River Formation of Malta, Montana.

**Otic Notch**—The otic notch is a restricted region bounded by the jugal/quadratojugal/ quadrate complex anteriorly, by the jugal and squamosal portions of the ventral laterotemporal bar dorsally, and the expanded wing of the squamosal posteriorly (Fig. 7). This space contained the external expression of the auditory meatus. The otic notch is unrestricted and triangular in protoceratopsids, *Diabloceratops*, and *Machairoceratops*. The otic notch is twice as anteroposteriorly long as dorsoventrally tall in *Lokiceratops* (best preserved on the left side) and rectangular, similar to *Styracosaurus albertensis*. The otic notch is subcircular and restricted in *Albertaceratops*, *Centrosaurus*, *Einiosaurus*, *Achelousaurus*, the Iddesleigh pachyrhinosaur, *Pachyrhinosaurus canadensis* and *Pachyrhinosaurus lakustai*.

**Internal Choanae**—The internal choanae, or internal nares, are located on the posterodorsal region of the oral cavity, bounded by the maxilla and palatine laterally, the pterygoid posteriorly, and the premaxilla anteriorly. The chamber would have been partially divided by the vomers, though the vomers and palatines are not preserved in EMK 0012. Air entering from the external nares would have passed into the nasal vestibule, passing posteriorly into the nasal antrum, then entering the pharynx at the posterior end of the nasals along the pterygoids. This area is difficult to assess in many centrosaurine specimens, but *Lokiceratops* seems to have had a similar configuration of the internal choanae to *Centrosaurus*.

**Foramen Magnum**—The foramen magnum in *Lokiceratops* is formed by the exoccipitals laterally and dorsally, and by the basioccipital ventrally. The supraoccipital is excluded from the dorsal margin of the foramen magnum. The basioccipital makes up the entire ventral margin of the foramen magnum. This differs significantly from *Diabloceratops* (UMNH VP 16699), in which the exoccipitals exclude both the basioccipital and supraoccipital from the foramen magnum, but is similar to all other centrosaurines in which this region is preserved.

## Cranial pneumaticity

**Dorsocranial Sinus**—The postorbitals, frontals, and parietals are excavated by the dorsocranial sinus (supracranial sinus of *Farke, 2010*), a presumably pneumatic system extending between the orbits and the base of the parietosquamosal frill (Figs. 3–7). Here, there is evidence of an anterior frontal fontanelle between the frontals and a posterior chamber formed between the frontals and the anterior parietal. This complex, preserved dorsal to the braincase, includes the cornual diverticulae that excavate the bases of the postorbital horncores, connected to the dorsotemporal fenestra by the dorsotemporal channels in the anteriodorsal portion of the parietal. The complex is more pronounced than the condition present in *Centrosaurus apertus* (ROM 767) and *Styracosaurus albertensis* (ROM 1436).

**Cornual Diverticulae**—The cornual diverticulae (*Farke, 2006*) are part of the dorsocranial sinus that extend into the base of the postorbital horncores to a length twice that of the radius of the orbit, extending more than 120 mm distally into the horns. Part of the ventral surfaces of the cornual diverticulae are preserved on the braincase and extended from the frontal fontanelle into the postorbital horncores as presumably pneumatic excavations. The pneumatic excavations extend into and occupy the entire base of the postorbital horncore in *Lokiceratops rangiformis* and differ from *Diabloceratops eatoni* (UMNH VP 16699), *Machiroceratops cronusi* (UMNH VP 20550), and *Medusaceratops lokii* (WDCB 12 1CA 2), in which the diverticulum only shallowly excavates the base of the horncore. The cornual diverticula excavation present in the postorbital horns is more extreme than any other centrosaurine or contemporary chasmosaurine including *Machairoceratops* (UMNH VP 20550), *Medusaceratops* (WDCB 12 1CA 2), *Agujaceratops mariscalensis*, *Chasmosaurus belli* (BMNH R4948), *Chasmosaurus kaiseni* (TMP 85.25.1), *Kosmoceratops* (UMNH VP 12198), *Judiceratops* (YPM VPPU 022404), *Ceratops montanus* (USNM

2411), or ceratopsid horncore from the Foremost Formation (TMP 1989.068.0001 (*Brown, 2018*)).

**Frontal Fontanelle**—The frontal fontanelle is a distinct midline opening between the frontals and lies just posterior to the base of the postorbital horncores (Figs. 3–7). The frontal fontanelle opens ventrally into the cornual diverticulae at the base of the horncores. In EMK 0012, the medial region of the frontal fontanelle and cornual diverticulae is crushed anteroposteriorly on the right horncore and dorsoventrally on the left horncore. Based on the edges of the crushed frontals, the frontal fontanelle in *Lokiceratops* is reconstructed as large and subcircular. Most of the lateral walls and ventral floor of the frontal fontanelle are preserved on the roof of the braincase.

**Dorsotemporal Channels**—The dorsotemporal channels (*Farke, 2010*) are smooth-floored grooves connecting the dorsotemporal fenestrae anteriorly to the posterior chamber of the dorsocranial sinus complex (Figs. 3–7). The right channel is partially preserved in EMK 0012. The smooth, wide channel floor is similar to the condition present in *Centrosaurus* (ROM 767).

**Parietal Channels**—The parietal channels are a smooth, relative untextured area between the posteroventral edge of the laterotemporal fenestra that extend posteriorly to the anterior portion of the parietal fenestrae. The dorsotemporal channel exits laterally into this area and the parietal channel is bounded medially by the anterior portion of the midline parietal bar posterior to the dorsocranial sinus and laterally by the "step" at the lateral edge of the dorsotemporal fenestra (Figs. 3–7). The parietal channels are similar to those in all other centrosaurines.

**Dorsal Narial Sinus**—The internal airway from the external naris passes into two chambers posteriorly inside the snout, demarked by the narial ridge, a distinct horizontal line on the medial surface of each nasal (Fig. 7). Multiple smaller ridges extend posteroventrally from the narial ridge, suggesting an attachment surface for soft tissues. This narial ridge is confluent with the nasal contribution to the narial spine and the dorsal narial sinus occurs dorsal to this feature. The dorsal narial sinus is triangular in *Lokiceratops* and more similar in shape to *Medusaceratops*, *Wendiceratops*, *Avaceratops* sp. MOR 692, and *Nasutoceratops*, than to the elongate rectangular chamber in *Sinoceratops* (ZCDM V0010), *Coronosaurus* (TMP 2002.68.07), and *Centrosaurus* (TMP 93.36.117).

**Ventral Narial Sinus**—The ventral narial sinus extends below the narial ridge on the medial surface of the nasal onto the medial surfaces of the posterior process of the premaxilla, lacrimal, and dorsal surface of the maxilla and is floored by the vomers and palatines (Fig. 7). The two narial sinuses may have been a single chamber with an "hourglass" or "8" shaped cross-section in anterior view. The shape of the ventral narial sinus in *Lokiceratops* resembles the shape in *Avaceratops* sp. MOR 692, *Nasutoceratops*, and *Centrosaurus*.

**Anterior Lacrimal Fossa**—Two chambers are associated with the posterior end of the ventral narial sinus on the medial surface of the lacrimal. The anterior lacrimal sinus is restricted to the medial surface of the lacrimal and is excluded from the posteromedial surface of the nasal (Fig. 7). No distinct demarcation separates the anterior portion of this fossa and the posterior end of the ventral narial sinus. The anterior lacrimal fossa may be analogous to the pneumatic sinus in *Nasutoceratops titusi* (UMNH VP 19466) that invaginates the posterior portion of the nasal (*Lund, Sampson & Loewen, 2016*), but there is no evidence for nasal pneumaticity in EMK 0012. The anterior lacrimal fossa in *Lokiceratops* is similar to the condition in *Avaceratops* sp. (MOR 692) in which it is subequal in size to the posterior lacrimal fossa. The anterior lacrimal fossa in EMK 0012 is much smaller than in *Sinoceratops zhuchengensis* (ZCDM V0010, ZCDM V0010), *Centrosaurus apertus* (ROM 43214), or the Iddesleigh pachyrhinosaur (TMP 2002.78.1).

**Posterior Lacrimal Fossa**—The posterior lacrimal fossa is located just ventral to the anterior border of the orbit and is separated from the anterior lacrimal fossa by a thin posteroventrally oriented ridge (Fig. 7). The posteroventral edge of the posterior lacrimal fossa extends ventrally onto the dorsal portion of the medial surface of the jugal. This fossa is separated from the adductor chamber by a medially directed fin of bone on the medial surface of the jugal. The posterior lacrimal fossa is oriented in line with the ascending ramus of the maxilla and resembles the posterior lacrimal fossa in *Avaceratops* sp. (MOR 692), *Sinoceratops zhuchengensis* (ZCDM V0010), *Centrosaurus apertus* (ROM 43214), the Iddesleigh pachyrhinosaur (TMP 2002.78.1), and *Pachyrhinosaurus lakustai* (TMP 89.55.1).

## Circumnarial region

The narial region of *Lokiceratops rangiformis* (EMK 0012) closely resembles those of other centrosaurines in having a subcircular external narial fossa with a well-developed premaxillary septum and a posteriorly positioned narial spine projecting into the external naris produced by the premaxilla and nasal. The subtriangular rostral, the ventral angle of the ventral premaxilla, and the anterior edentulous section of the maxilla form the buccal cutting surface anterior to the maxillary teeth (Figs. 3–5, 7–10).

**Rostral**—The rostral is single median element that caps the paired premaxillae anteriorly. The overall surface texture of the anterior and lateral surfaces of the rostral is rugose with multiple elongate, two to three mm wide channels and pits all roughly trending toward the anteroventral tip of the element. It is a tri-partite element in lateral view, with a squared anteroventral tip, two short posteroventral processes, and a tall dorsal process bordering the anteroventral margins of the premaxillae (Figs. 3–5, 7–9). The posteroventral process is short, approximately one third the length of the dorsal process. In lateral view, the overall arcuate, concave posterior margin of the rostral is interrupted by a modest convexity positioned just ventral to the midpoint, dividing the posterior margin into two concave segments. In lateral view, the anterior margin of the rostral is nearly straight for approximately half of its ventral length before arcing dorsally along the premaxilla. The ventral margins of the rostral form sharp cutting edges, which converge anteriorly in a

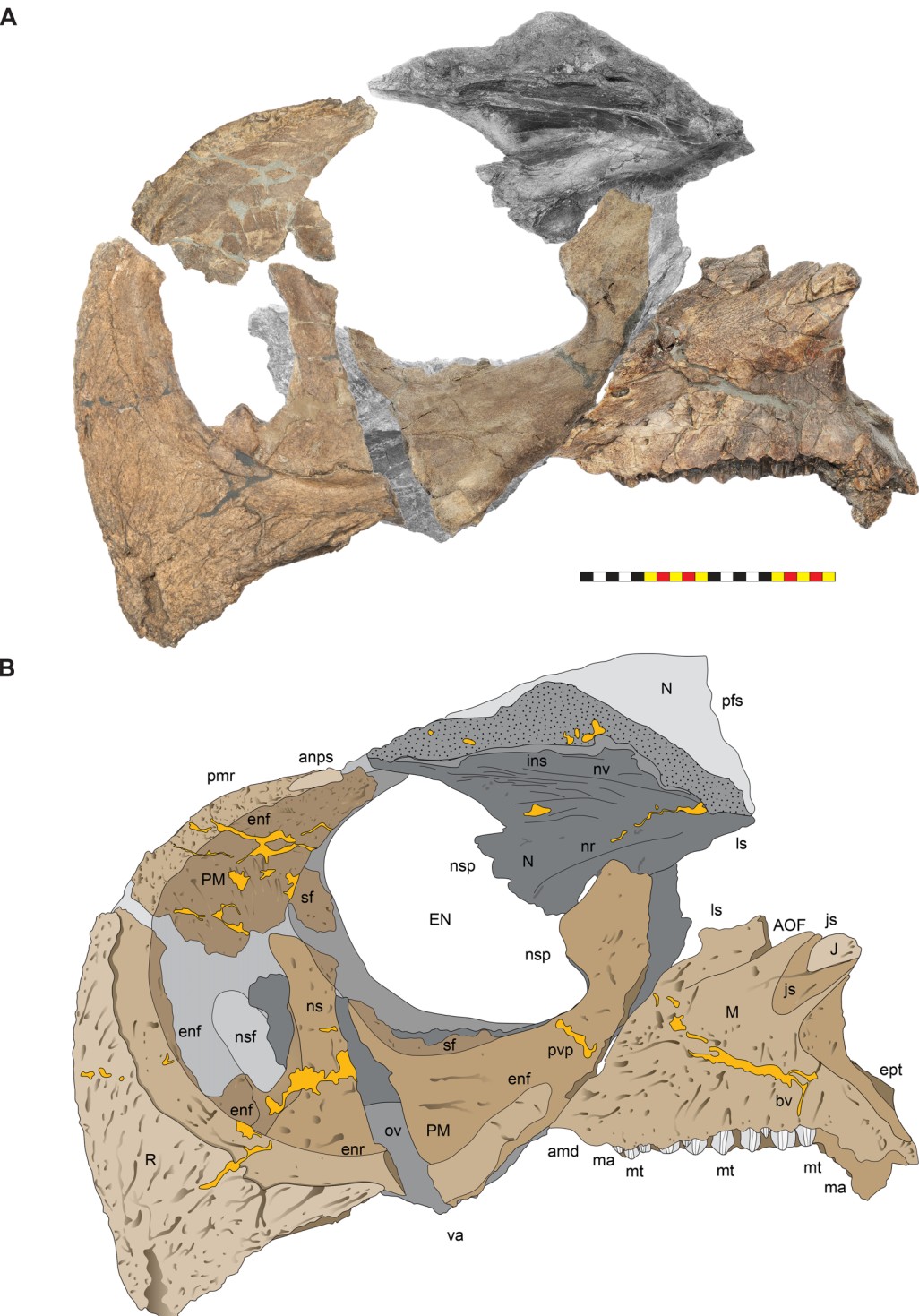

**Figure 8 Left circumnarial region of *Lokiceratops rangiformis* n. gen et n. sp. (EMK 0012).** (A) Photomosaic of preserved bones of the circumnarial region of EMK 0012 in left lateral view. (B) Interpretative line drawings in lateral view. Dark gray indicates medial surfaces of elements of the right side of the face. Lighter grays indicate outlines interpreted from the preserved material. Orange denotes putty filling cracks. Note that while the reconstructed nasal appears to have a peaked ornament dorsally, there is no evidence of a narial ornament in this specimen. This is an artifact of incomplete preservation of the internasal suture. *Osteological abbreviations*: amd, anterior maxillary diastema; anps, anterior nasal

**Figure 8** (continued)
process suture; AOF, antorbital fenestra; bv, buccal vault; enf, external narial fossa; EN, external naris; enr, endonarial recess; ept, ectopterygoid contact; ins, internarial suture; js, jugal suture; ls, lacrimal suture; M, maxilla; ma, maxillary alveoli; N, nasal; nsf, nascent narial fossa; ns, narial strut; nr, narial ridge; sp, narial spine; ov, oral vault; pfs, prefrontal suture; PM, premaxilla; pmr, premaxillary rugosity; pvp, premaxilla ventral process; R, rostral; sf, septal flange; va, ventral angle. Photos by Marcus Donivan. Scale bar equals 20 cm.

narrow arc. The rostral is triangular with a ventral process that is much shorter than the element height, morphologically similar to all centrosaurines including *Diabloceratops* (UMNH VP 16699), *Centrosaurus* (AMNH 5259), and *Pachyrhinosaurus* (CMN 9845), and differing from the sub-equal dorsal and ventral processes of the rostrals of *Zuniceratops* (MSM P2101) and all known Chasmosaurinae.

**Premaxillae**—The majority of both premaxillae are preserved in *Lokiceratops* (EMK 0012) (Figs. 3–5, 7–9), recovered in contact with each other, although slightly displaced, and fused to the rostral. As in other centrosaurines, the premaxilla is a crescentic element, consisting of a primary ventral body from which emanates a laminar anterior portion circumscribed by a thickened anterior and ventral margin, and a dorsally expanded posteroventral process. The laminar central portion, or premaxillary septum, is broad and smooth, forming the medial wall of the anterior narial fossa. The posterior half of the septum is marked with a moderately thickened and anterodorsally inclined nascent narial strut, itself bordered posteriorly by a laminar septal flange along the anterior border of the external naris.

Anterior to the nascent narial strut, smooth margins indicate the presence of an ovoid depression, or incipient narial fossa. Anteriorly and anteroventrally, the thickened rim of the premaxilla is sharply offset laterally from the smooth endonarial recess in the anteroventral portion of the narial fossa. Externally, the ridge contacts, and is tightly sutured to, the rostral. Anterodorsally, the thickened rim of the premaxilla forms the margin of the rostrum between the rostral and nasal. Here, it is relatively rugose, ornamented with deep fissures and grooves, possibly related to keratinous or adherent tissues between the keratinous coverings on the rostral and tissues on the smoother nasal.

As in all ceratopsids, the premaxillae are edentulous, and ventral surfaces of the premaxillae contribute to a posteroventral cutting surface. These surfaces form an inclined, beveled ventral edge, the caudal continuation of the cutting edges of the rostral, which terminates in a robust 'ventral angle' (Fig. 9). This ventral angle occludes with the angled lateral cutting surface of the predentary which is characteristic of centrosaurines. In lateral view, the ventral angle drops well below the ventral margin of maxillary tooth row, in contrast to the slightly developed ventral angle in *Diabloceratops* (UMNH VP 16699), but as in all other centrosaurines.

The posterior processes of the premaxillae diverge in ventral view (Fig. 10). Posteriorly, the posteroventral process of the premaxilla meets the maxilla along a posterodorsally inclined suture, ascending dorsally to a laminar, externally exposed contribution to the narial spine. Dorsally, the nasal contacts the premaxilla to form the ventral portion of the

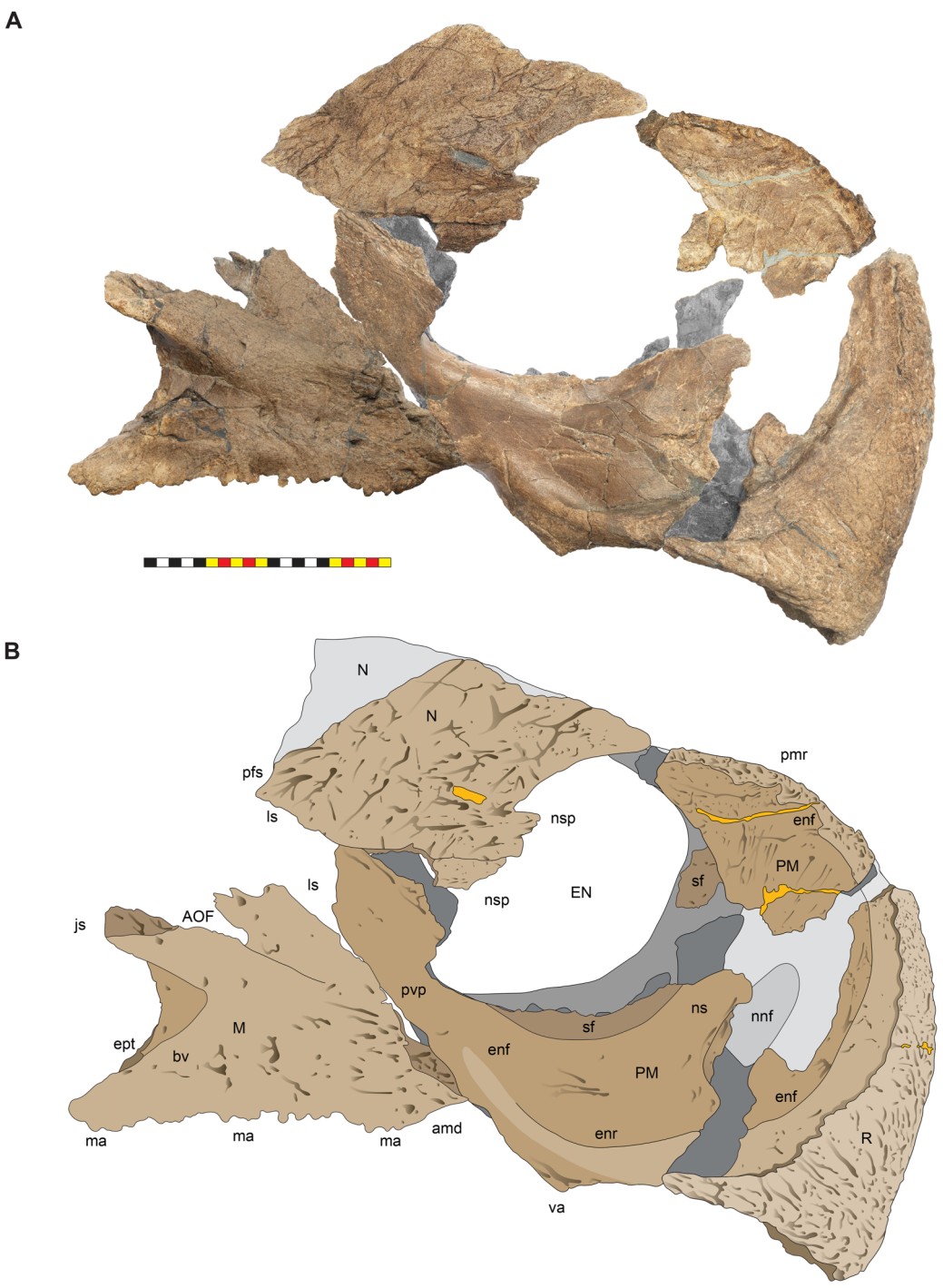

**Figure 9 Right circumnarial region of *Lokiceratops rangiformis* n. gen et n. sp. (EMK 0012).** (A) Photomosaic of preserved bones of the circumnarial region of EMK 0012 in right lateral view. (B) Interpretative line drawings in lateral view. Dark gray indicates medial surfaces of elements of the left side of the face. Lighter grays indicate outlines interpreted from the preserved material. Orange denotes putty filling cracks. Note that while the reconstructed nasal appears to have a peaked ornament dorsally, there is no evidence of a narial ornament in this specimen. This is an artifact of breakage. *Osteological abbreviations*: amd, anterior maxillary diastema; AOF, antorbital fenestra; bv, buccal vault; enf, external narial fossa; EN, external naris; enr, endonarial recess; ept, ectopterygoid contact; ins, internarial suture; js, jugal suture; ls, lacrimal suture; M, maxilla; ma, maxillary alveoli; N, nasal; ns, narial strut; nsf, nascent

**Figure 9** (continued)

septal fossa; nsp, narial spine; nv, narial vault; pfs, prefrontal suture; PM, premaxilla; pmr, premaxillary rugosity; pvp, premaxilla ventral process; R, rostral; sf, septal flange; va, ventral angle. Photos by Marcus Donivan. Scale bar equals 20 cm.                 

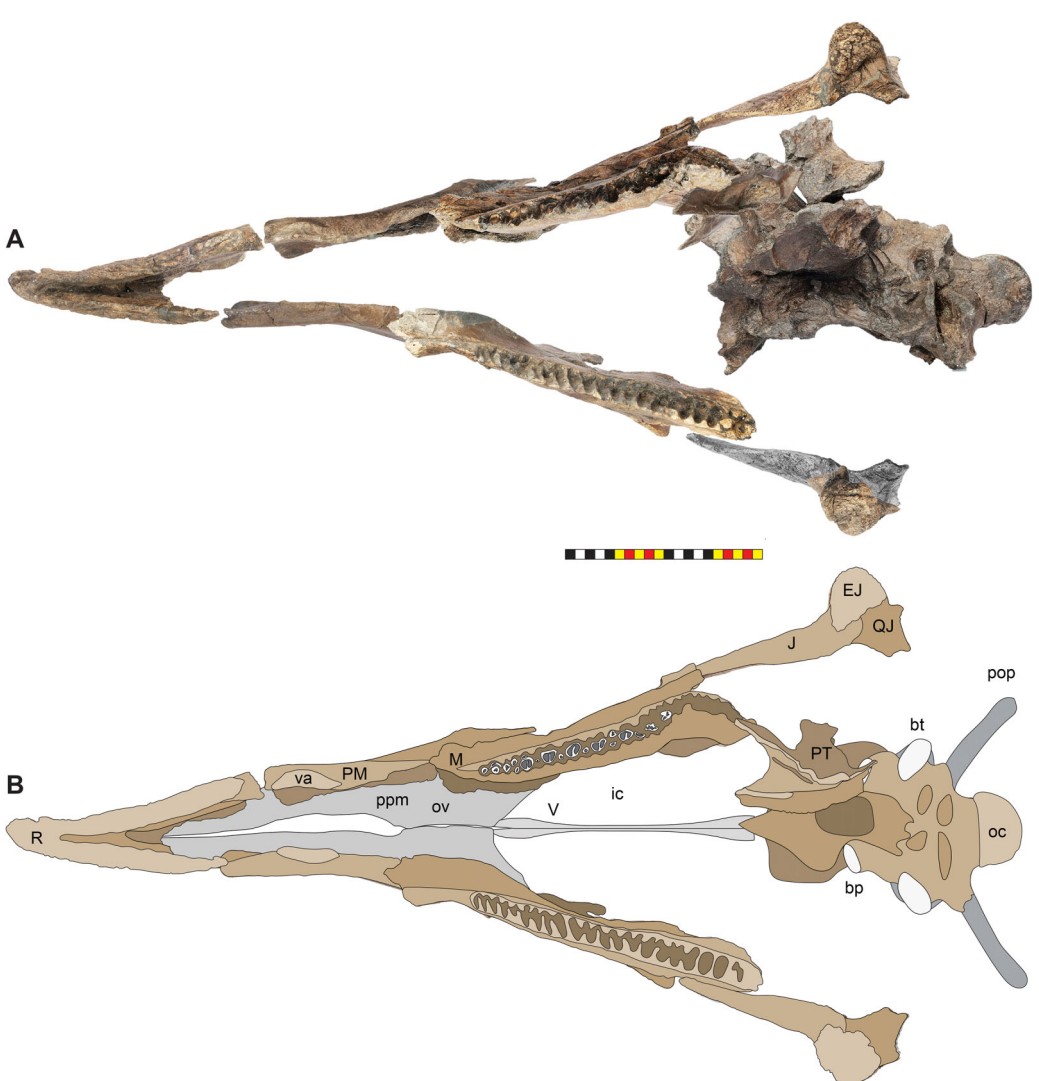

**Figure 10 Ventral occlusal view of the facial skeleton of *Lokiceratops rangiformis* n. gen et n. sp. (EMK 0012).** (A) Photomosaic of preserved bones of the facial skeleton of EMK 0012 in ventral occlusal view. The grayscale element is the ventral surface of the left jugal mirrored to the right side underneath the right jugal and epijugal for clarity. (B) Interpretative line drawings in lateral view. Lighter grays indicate outlines interpreted from the preserved material. *Osteological abbreviations*: bt, basal tubera; bp, basipterygoid; bv, buccal vault; EJ, epijugal; ic, internal choanae; J, jugal; M, maxilla; ma, maxillary alveoli; N, nasal; ov, oral vault; PM, premaxilla; ppm, premaxillary palatal process; QJ, quadratojugal; pop, paroccipital process; PT, pterygoid; oc, occipital condyle; ov, oral vault; R, rostral; V, vomer; va, ventral angle. Photos by Marcus Donivan. Scale bar equals 20 cm.

narial spine, projecting anteriorly and lateral to the narial opening. Posteriorly, the dorsal portion of the posteroventral process meets the lacrimal along a dorsoventrally directed suture, excluding contact between the nasal and maxilla as in *Wendiceratops* (TMP 2014.029.0074), *Styracosaurus* (CMN 344), and *Achelousaurus* (MOR 591), and contrasting with the condition in *Diabloceratops* (UMNH VP 16699), *Centrosaurus* (AMNH 5259), and *Nasutoceratops* (UMNH VP 16800 and UMNH VP 19466.1). Both the nasal and the premaxilla contribute to a narial spine at the posterior edge of the external narial fossa. Medially, the palatal processes of the premaxillae are not preserved but the broken edges of these platforms are preserved and form the demarcation between the buccal vault and the external naris.

**Maxillae**—Both maxillae are preserved in *Lokiceratops* (EMK 0012), though the right maxilla is missing its teeth, and the left maxilla exhibits post-depositional plastic deformation (Figs. 3–5, 7–10). The maxilla contacts the posteroventral surface of the premaxilla anteriorly, the nasal dorsally, the lacrimal posterodorsally, and the jugal posteriorly. The palatine and ectopterygoid contact the maxilla medially. The maxilla is generally similar to other centrosaurines, being trapezoidal in shape, and nearly as tall dorsoventrally as it is long anteroposteriorly. It consists of a dentigerous horizontal ramus from which a bifurcating dorsal process emanates. The external surface of the horizontal ramus is moderately rugose, pierced obliquely by several foramina associated with anteroventrally directed neurovascular grooves. A distinct lateral ridge is inclined moderately anterodorsally in lateral view and curves laterally from the toothrow to the premaxillary shelf, for contact with the posterior occlusal surface of the premaxilla posterior to the ventral angle. A similar shelf occurs in *Diabloceratops*, *Nasutoceratops*, and *Wendiceratops*, but not in *Centrosaurus*. The "cheek" muscle, the *M. adductor mandibulae externus*, likely originated on this shelf on the maxilla and inserted on the lateral ridge of the dentary lateral to the dentary tooth row (*Nabavizadeh, 2020*).

Medially, the maxillary cavity is a well-developed internal chamber formed largely within the medial surface of the ventral portion of the ascending maxillary ramus. The medial wall of this space is formed by the maxilla, palatine, and pterygoid, and the roof is formed by the lacrimal, palatine, and maxilla. The maxillary cavity, posited here to be of pneumatic origin and interconnected with the antorbital sinus, is distinct from the intramaxillary sinus of some basal neoceratopsians (*e.g.*, *Bagaceratops*, *Protoceratops*). The maxillary cavity forms within the body of the maxilla on the medial side and was likely connected dorsally to the antorbital pneumatic system. The internal maxillary fossa is a broad, shallow trough that extends along the internal surface of the ascending maxillary ramus from the lacrimal posteriorly to the premaxilla anteriorly and is also bounded medially by the palatine. The buccal wall of the right maxilla is missing at the line of the dental foramina, the teeth are missing, and the toothrow is inset from the lateral surface of the maxilla, bearing 22 alveolar grooves and extending anteriorly to a short edentulous ridge.

Posteriorly, the toothrow continues along the maxillary body onto the posterior process, which bears the last five alveoli ventral to the anterior extension of the postmaxillary

fenestra. The dorsal process is divided into a laminar anterodorsal portion and a robust posterodorsal portion by the antorbital fenestra and anteriorly extending, slot-like antorbital fossa. The anterior body and anterior edge of the anterodorsal process meet the premaxilla along a continuous, posterodorsally directed suture. The dorsal margin of the anterodorsal process forms the anteroposteriorly directed suture with the lacrimal. The posterodorsal process originates from the lateral body of the maxilla as a thickened, laterally expanded, posterodorsally inclined jugal ridge, passing below the antorbital fenestra to meet the anterior process of the jugal along a broad sutural surface. Laterally, the main body of the maxilla is medially inset from the ridge confluent with the anterior process of the jugal forming the buccal vault or "cheek space". Internally, a vaulted oral cavity is outlined by a thin palatal process that extends anteriorly to form the medial portion of the premaxillary shelf (Fig. 11). Posteriorly, the palatal process blends into the surface at the level of the suture for the palatine. This suture continues posteriorly onto the medial surface of the posterior ramus for contact with the pterygoid.

**Nasal**—The anterior portion of the right nasal is preserved in *Lokiceratops* (EMK 0012), broken just lateral to the midline suture with its counterpart (Figs. 3, 4, 7–11). As in all known centrosaurines the nasals are paired elements lacking a discrete epinasal ossification (seen in chasmosaurines). Generally, the nasals form much of the proximodorsal rostrum anterior to the orbits and forms the dorsal and posterodorsal borders of the external nares. Anteriorly, the nasal forms the smooth, arcuate posterodorsal margin of the external nares, interrupted about halfway down its extent by the acute anterodorsal extension of the contribution of the nasal to the narial spine. The narial spine is a distinct bony process that extends from the posterior ectonarial margin anteromedially into the nasal vestibule. This process, a derived feature of Centrosaurinae, arises from the nasal but often includes a ventral contribution from the premaxilla. The narial spine occurs in conjunction with a bilateral narrowing of the nasal cavity, effectively forming an "hourglass" or "8" shaped opening (as viewed anteriorly) into the nasal cavity proper. Chasmosaurines exhibit a similar "pinching" of this portion the nasal cavity, associated with a medial thickening of the nasal that is likely homologous to the narial spine of centrosaurines. In centrosaurines, however, this process is distinct and projects anteriorly into the nasal vestibule.

In *Lokiceratops*, the nasal meets the dorsal lamina of the posteroventral process of the premaxilla ventrally along an anteroventrally directed suture. The ventral margin of the nasal posterior to the premaxilla meets the anterior extent of the lacrimal, excluding contact with the anterodorsal portion of the dorsal process of the maxilla. Internally, the nasal is domed forming an internal nasal vault similar to the configuration in *Nasutoceratops* (UMNH VP 19466.1) but different from *Medusaceratops lokii* (WDCB-MC-001), in which the entire dorsal surface of the internal narial vault is flat from anterior to posterior. The narial ridge is an internal horizontal ridge that extends from the nasal contribution of the narial spine posteriorly along the medial surface of the preserved right nasal (Fig. 11). Multiple smaller ridges extend posteroventrally from the narial ridge, suggestive of an attachment surface for soft tissues. Even though the dorsocranial sinus complex is more extensive in *Lokiceratops* than any other centrosaurine, there are no

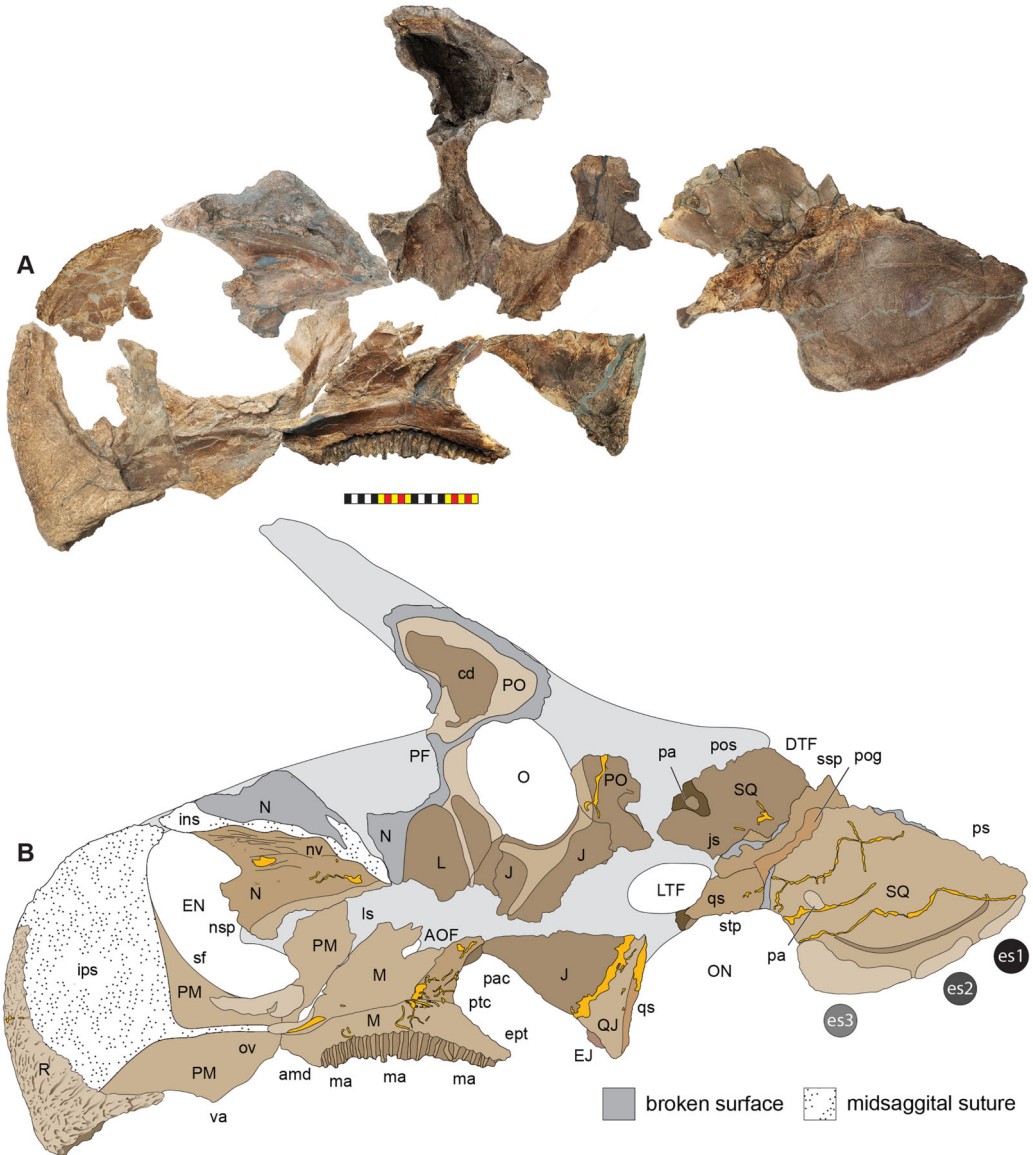

**Figure 11 Medial surface of the skull of *Lokiceratops rangiformis* n. gen et n. sp. (EMK 0012).** (A) Photomosaic of internal surfaces of the facial region of EMK 0012 in medial view. Most elements are from the right side of the skull except for the lacrimal, palpebral, jugal, quadratojugal, and distal wing of the squamosal which are reversed from the left side for clarity. (B) Interpretative line drawings in lateral view. Dark gray indicates broken edges, stipples indicate midsagittal sutures of elements along the midline. Lighter gray indicates the outline of the skull interpreted from the preserved material. Orange denotes putty infilling cracks. Note that while the nasal appears to have a peaked ornament dorsally, there is no evidence of a narial ornament in this specimen. This is an artifact of breakage. *Osteological abbreviations*: amd, anterior maxillary diastema; AOF, antorbital fenestra; EN, external naris; enr, endonarial recess; ept, ectopterygoid contact; ins, internarial suture; ips, interpremaxillary suture; J, jugal; js, jugal suture; L, lacrimal; ls, lacrimal suture; M, maxilla; ma, maxillary alveoli; N, nasal; nnf, nascent narial fossa; nns, nascent narial strut; nse, narial septum; nsp, narial spine; nv, narial vault; O, orbit; ON, otic notch; ov, oral vault; P, parietal; pa, pathology; pac, palatine contact; PM, premaxilla; pmr, premaxillary rugosity; PO, postorbital; pog, paroccipital groove; pos, postorbital suture; ptc, pterygoid contact; qs, quadrate wing suture; qws, quadrate wing suture; R, rostral; stp, squamosal temporal process; va, ventral angle; vsg, ventral squamosal groove. Photos by Marcus Donivan. Scale bar equals 20 cm.

pneumatic chambers in the posterodorsal region of the nasal as seen in *Nasutoceratops* (UMNH VP 16800 and UMNH VP 19466.1).

There is no nasal ornamentation. The convex lateral external surface of the body of the nasal is moderately sculpted with deep, branching neurovascular grooves and a rough, interwoven texture. This texture is similar those present on the dorsal surfaces of the lacrimals, squamosals, and parietals, but unlike the distinctive texture of the postorbital horncores. There is no change in nasal surface texture from the ventral surface of the nasal to the dorsal surface. The anterior process of the nasal lacks the rugosity present on the dorsal surface of the premaxilla or the rostral just anterior to the dorsal surface of the premaxilla. There is also no evidence that the nasal formed a pachyostotic boss as in pachyrhinosaurs that appear in the sediments along the northeastern coast of Laramidia at least three million years later (*Sampson, 1995*; *Sampson & Loewen, 2010*). The overall shape of the preserved dorsal margin and the preserved surface texture on the dorsolateral surface, strongly indicate an absence of nasal ornamentation in *Lokiceratops*, differentiating it from the distinct nasal horns in *Albertaceratops*, *Medusaceratops*, and *Wendiceratops*. *Zuniceratops* also lacks nasal ornamentation and *Diabloceratops* has a very limited nasal ornamentation.

## Circumorbital region

The circumorbital region in *Lokiceratops rangiformis* (EMK 0012) closely resembles those of most basal centrosaurines in having orbits with dorsally elongated postorbital horns and a well-developed antorbital buttress formed by the prefrontal, palpebral, and lacrimal. The suborbital region is comprised of the jugal and quadratojugal as in all other centrosaurines (Figs. 12, 13).

**Lacrimal**—Portions of both lacrimals are preserved in *Lokiceratops* (EMK 0012), the right consisting of most of the circumorbital region and a portion of the anterodorsal region tightly sutured to the right nasal, the left consisting of most of the posterior half of the element including the circumorbital region (Figs. 3–5, 7–9, 11–13). The lacrimal contacts the nasal and premaxilla anteriorly, the maxilla and jugal ventrally, and the prefrontal and palpebral dorsally. The lacrimal forms the anteroventral portion of the orbit below the palpebral and above the jugal. Overall, the lacrimal is a laminar, anteroventrally oriented element that contributes to the lateral surface of the posterior region of the rostrum, contacting the dorsal portion of the posteroventral process of the premaxilla anteriorly, the anterodorsal portion of the dorsal process of the maxilla anteroventrally, the nasal anterodorsally, the prefrontal dorsally, and the jugal posteroventrally. Sutures with the nasal are not visible in external or internal surface view, though sutures with the maxilla, premaxilla, and jugal indicate that the main axis of the lacrimal was oriented anteroventrally. The lacrimal contributes to the ventral half of the anterior margin of the orbital opening, bearing a moderately elevated rim ventrally. Dorsally, the ventral half of the palpebral is sutured to the external surface of the lacrimal to form the pronouncedly robust anterior rim of the orbital opening. Internally, the surface of the lacrimal is

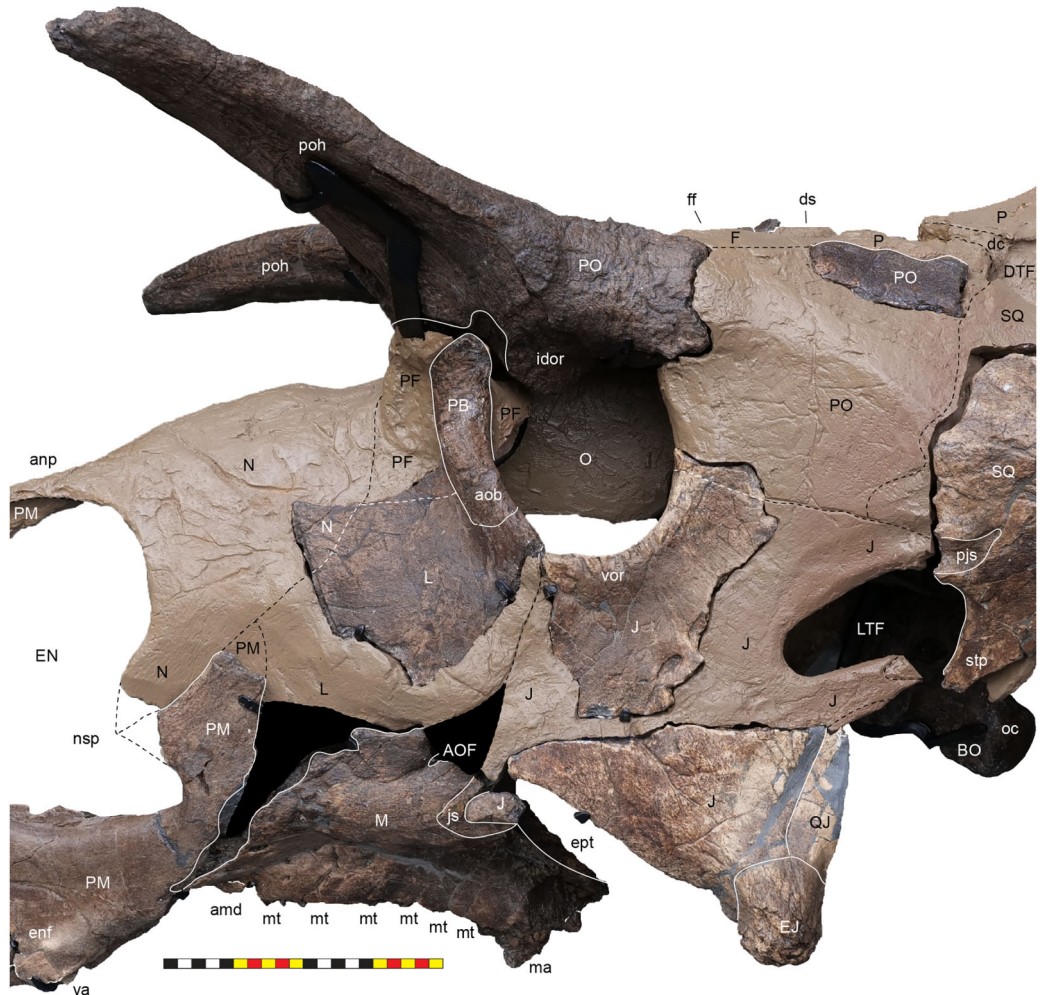

**Figure 12 Left circumorbital region of *Lokiceratops rangiformis* n. gen et n. sp. (EMK 0012).** Photograph of the right circumorbital region of EMK 0012 in lateral view. *Osteological abbreviations*: amd, anterior maxillary diastema; anp, anterior nasal process; aob, antorbital buttress; AOF, antorbital fenestra; BO, basioccipital; dc, dorsotemporal channels; ds, dorsocranial sinus; DTF, dorsotemporal fenestra; enf, external narial fossa; EJ, epijugal; EN, external naris; ept, ectopterygoid contact; f, frontal; ff, frontal fontanelle; idor, internal orbital rim; J, jugal; L, lacrimal; LTF, laterotemporal fenestra; M, maxilla; ma, maxillary alveoli; N, nasal; nsp, narial spine; O, orbit; oc, occipital condyle; P, parietal; PB, palpebral; PF, prefrontal; pjs, posterior jugal suture; PM, premaxilla; pmr, premaxillary rugosity; poh, postorbital horncore; QJ, quadratojugal; SQ, squamosal; rostral; ssp, squamosal step; stp, squamosal temporal process; va, ventral angle; vor, ventral orbital rim. Photo by Marcus Donivan. Scale bar equals 20 cm.

excavated into the anterior and posterior lacrimal fossae (Fig. 7) bounded by sharply pinched crests.

**Jugal**—Sections of both jugals are preserved in *Lokiceratops* (EMK 0012), each consisting of portions of the orbital margin and the ventral complex with the epijugal and quadratojugal (Figs. 3–5, 7–9, 11–13). As in other ceratopsids, the jugal of *Lokiceratops* is a triradiate, laminar element, consisting of an anterior process that would have contacted the lacrimal and maxilla, a posterior process contacting the postorbital and squamosal, and a

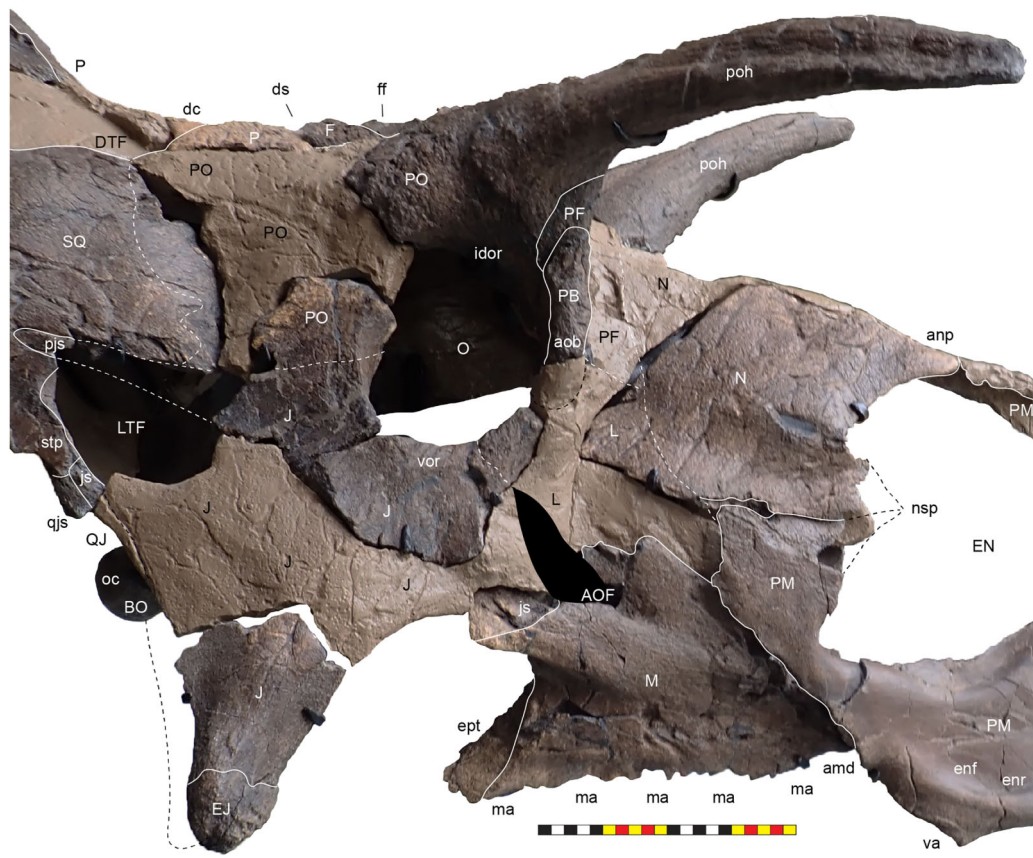

**Figure 13 Right circumorbital region of *Lokiceratops rangiformis* n. gen et n. sp. (EMK 0012).** Photograph of the right circumorbital region of EMK 0012 in lateral view. *Osteological abbreviations*: amd, anterior maxillary diastema; anp, anterior nasal process; aob, antorbital buttress; AOF, antorbital fenestra; BO, basioccipital; dc, dorsotemporal channels; ds, dorsocranial sinus; DTF, dorsotemporal fenestra; enf, external narial fossa; EJ, epijugal; EN, external naris; enr, endonarial recess; ept, ectopterygoid contact; f, frontal; ff, frontal fontanelle; idor, internal orbital rim; J, jugal; L, lacrimal; LTF, laterotemporal fenestra; M, maxilla; ma, maxillary alveoli; N, nasal; nsp, narial spine; O, orbit; oc, occipital condyle; P, parietal; PB, palpebral; PF, prefrontal; pjs, posterior jugal suture; PM, premaxilla; pmr, premaxillary rugosity; poh, postorbital horncore; QJ, quadratojugal; SQ, squamosal; rostral; stp, squamosal temporal process; va, ventral angle; vor, ventral orbital rim. Photo by Mark Loewen. Scale bar equals 20 cm.               

ventral process contacting the quadratojugal and hornlike epijugal. The jugal forms the ventral margin of the orbit and contributes to the dorsal, anterior, and ventral margins of the laterotemporal fenestra, though the latter two portions are not preserved in EMK 0012. Anteriorly, the jugal contact with the maxilla is preserved on the posterodorsal portion of the dorsal process of the maxillae where it transitions into the lateral ridge of the maxilla. The suture with the lacrimal is likely preserved on the left side of EMK 0012, though the surface evidence for the suture is not visible. Posteriorly, the jugal is deeply inset into the anterior lamina of the squamosal along a deeply anastomosing suture, extending posterodorsaly from the posterolateral margin of the laterotemporal fenestra to a sharp point, then along an anterodorally directed suture to the posterior margin of the orbit. The external surface of the jugal is moderately rugose and ornamented with shallow

neurovascular grooves. The dorsal contribution to the orbital margin is elevated as a subtle rim, ornamented with deeper grooves radiating from the orbit. Ventrally, the jugal terminates as a triangular process, its anteroventral margin forming the lateral margin of the entrance to the coronoid fossa for reception of the coronoid process of the occluded mandible and associated musculature. At the ventral and posterior portions of the ventral process, the jugal is tightly sutured to the quadratojugal in an overlapping scarf joint, the jugal laterally overlapping the posteriorly exposed quadratojugal. Both the jugal and quadratojugal contribute equally to the suture for the laterally directed epijugal horn. Internally, a rounded crest extends anteroventrally from the posteroventral margin of the orbit to the posterodorsal portion of the dorsal process of the maxilla, creating a continuous boundary between the posterior lacrimal fossa and the posterior, internal smooth surface of the coronoid fossa (Fig. 7).

**Epijugal**—Both epijugals are preserved in *Lokiceratops* (EMK 0012) and form short, laterally directed epijugal horns (Figs. 3–5, 7–13). The epijugal articulates to the ventrolateral surfaces of the jugal and quadratojugal. Overall, the epijugal horns extend 7 cm laterally from the jugal surface and are triangular in cross-section, with a slight anteriorly directed, flat facet. The external surface of the epijugal horn is ornamented with deep, laterally directed grooves and pits with an overall rugose texture indicative of a keratinous covering.

**Quadratojugal**—The left quadratojugal is preserved and visible in *Lokiceratops* (EMK 0012) articulated with the jugal (Figs. 10–13). In general form, the quadratojugal is a laminar, triangular element with a thickened ventral process and thin anterior lamina. As in other ceratopsids, the quadratojugal is mediolaterally compressed between the ventral process of the jugal and the lateral articulation with the quadrate. The anterior lamina medially underlaps the jugal in a broad scarf joint, thinning to an anteriorly convex termination on the posteromedial half of the jugal. Ventrally, the quadratojugal thickens to form a robust contribution to the suture for the epijugal horn laterally and a broad butt suture with the distal quadrate medially, a small ventral projection of the quadratojugal extending beyond the epijugal to cover the lateral quadrate. The posteroventral margin of the quadratojugal forms the free border of the skull in lateral view, overlapping the lateral margin of the quadrate. Dorsally, it would have extended to meet the squamosal, contributing to the strut posterior to the laterotemporal fenestra.

**Palpebral**—Both palpebral elements are preserved in *Lokiceratops* (EMK 0012) as ovoid protrusions tightly sutured to the anterior margins of the orbits (Figs. 12, 13). In overall shape, the palpebral is blocky, steep sided, and forms the main portion of the antorbital buttress ventral to the postorbital horn. Even though we consider this individual to be of adult maturity, based on many other osteological criteria, the palpebral sutural lines are visible where it contacts the prefrontal and lacrimal, over which it sutures.

**Prefrontal**—Both prefrontals are preserved in *Lokiceratops* (EMK 0012), the left as a fragmentary portion firmly sutured to the medial surface of the palpebral, the right as a

moderately larger portion sutured to the postorbital and medially to the palpebral (Figs. 3–5, 11–13). The prefrontal forms the dorsal half of the anterior orbital margin where it is entirely covered by the palpebral. Dorsally, the prefrontal is firmly sutured to the postorbital ventral to the postorbital horn. The prefrontal meets the nasal anteriorly and the frontal medially. The prefrontals are medial to the palpebrals and form the anterodorsal part of the internal orbit between the postorbital dorsally and the lacrimal anteroventrally. The prefrontals also form part of the antorbital buttress and contacts the nasals anteriorly, and the lacrimals ventrally.

**Frontal**—Only small fragments of the frontals are preserved in *Lokiceratops* (EMK 0012), fused indistinguishably to the medial portions of the postorbitals on the dorsal skull roof and on the dorsal surface of the braincase. One large portion, fused to the posterior lamina of the right postorbital, preserves a portion of the lateral margin of the frontal fontanelle. The frontal would have contacted the nasal and prefrontal anteriorly, the postorbital laterally, and the parietal posteriorly. Internally, the frontals are deeply excavated by pneumatic recesses associated with internal pneumatic tissues continuous into the postorbital and the frontal fontanelle (described above).

**Postorbital**—Both postorbitals are preserved in *Lokiceratops* (EMK 0012), forming distinctive, elongate horns over the orbits, and contributing to the dorsal and posterior margins of the orbits (Figs. 3–5, 7, 11–13). Each postorbital is moderately distorted by crushing, largely effecting the morphology of the horns. The left is compressed dorsoventrally and the right is compressed mediolaterally. In overall form, the postorbital of *Lokiceratops* is similar to other ceratopsids with elongate postorbital horns, including centrosaurines like *Albertaceratops, Wendiceratops, Nasutoceratops, Avaceratops*, and *Diabloceratops*, with a laterally facing orbital portion, a distinct postorbital horncore, and a broad posterior lamina. The orbital portion of the postorbital meets the prefrontal and frontal anteriorly and is laterally capped by the dorsal extent of the palpebral. Posteroventrally, the postorbital meets the jugal at the midpoint of the orbital wall. Dorsally, the orbital portion of the postorbital is continuous with the lateral surface of the postorbital horn, its surface textured with distinct grooves and rugosity. The postorbital horn projects anterolaterally with a moderate ventral curvature in anterior view. Though both postorbital horns are crushed, the postorbital horn would have been subcircular to elliptical in cross section, tapering gradually distally to a rounded point. The external surface of the postorbital horn is ornamented with deep longitudinal grooves and an overall rugose texture indicative of a keratinous covering. Posteriorly, the dorsal surface of the horn is continuous with the posterior lamina, a rugose, dorsally convex extension of the postorbital that meets the frontal and parietal medially, the squamosal posterolaterally, and the jugal laterally.

## Parietosquamosal frill

The parietosquamosal frill of *Lokiceratops* (EMK 0012) closely resembles those of other basal centrosaurines in having crescentic, fan-shaped squamosals and an elongated asymmetrical frill as in *Diaboloceratops*, *Albertaceratops*, and *Medusaceratops*, and differs

from the rounder parietals in *Avaceratops*, *Nasutoceratops*, *Wendiceratops*, and *Sinoceratops*. The main differences in the frill of *Lokiceratops* and other centrosaurines is in its seven epiparietal ornamentations and their orientations (Figs. 3–5, 6, 11, 14–18).

**Squamosal**—Both squamosals are preserved in *Lokiceratops* (EMK 0012), with the left squamosal nearly complete (Figs. 3–5, 7, 11, 14, 15). As in other ceratopsids, the squamosal can be divided into a distally expanding posterior process contributing to the formation of the anterolateral frill, an anteroventrally directed temporal process, and a broad, sheet-like anterior blade. The squamosal contacts the jugal anteroventral and anterodorsal to the laterotemporal fenestra, the postorbital dorsally, and the parietal posteriorly. Medially, at the point where the three processes of the squamosal converge, the squamosal contacts the quadrate anteroventally and the paroccipital process of the otoccipital along an arcuate suture. From this point anteriorly, the temporal and anterior processes diverge around the laterotemporal fenestra, forming its posteroventral and posterior borders. At the posterodorsal margin of the laterotemporal fenestra, the suture for the posterodorsal ramus of the jugal is present as a shallow, tapering groove. Dorsal to the jugal suture, the anterior process contacts the posterior extent of the postorbital along an anastomosing scarf joint, the squamosal passing deep to the postorbital. The squamosal forms the anterolateral boarder of the dorsotemporal fenestra and anteriorly contacts the parietal lateral to the dorsotemporal channels. Overall, the posterior process of the squamosal forms the "fan-shaped" sub-rectangular blade of the anterolateral portion of the frill. The "fan-shape" of this blade expands from a constricted otic notch ventrally. The anterior blade and temporal processes of the squamosal are subequal in length and are demarked by a distinct stepped-up margin at the lateralmost portion of the laterotemporal fenestra. This pronounced "step" is a continuation of the ventrally positioned, perpendicularly oriented, paroccipital groove on the dorsal surface. The dorsal surface of the anterior squamosal blade lacks a prominent rounded ridge or a sharp peaked ridge. The ventral surface of the squamosal has a medial suture for the quadrate on the tip of the anteroventral temporal process, and a subtriangular facet for the medial wing of the quadrate anterolateral to a "slot-like" facet or groove for the lateral paroccipital process of the exoccipital. Part of this quadrate wing scar is the likely origin site for the jaw opening muscle the M. depressor mandibulae (*Sereno, Xijin & Lin, 2009*). Anteromedial to the paraoccipital groove is the smooth adductor chamber that housed *M. adductor mandibulae externus profundus*, and *M. adductor mandibulae externus medialis* (*Holliday et al., 2019*).

The dorsal surface texture of the squamosal transitions from a more heavily rugose texture at the anterodorsal contact with the postorbital to a rugose texture with a number of other variably inscribed vascular grooves on the dorsal surface of the squamosal, most originating from points adjacent to the preserved anterior and anteromedial margins. Overall, the dorsal surface texture is heavily rugose and covered by numerous fine pits that characterize adult centrosaurines (see below). Ventrally, the squamosal is thickened through the contacts for the quadrate and the exoccipital flange. The ventral surface is generally smooth and gently concave. Ventrally, a pronounced groove traces the

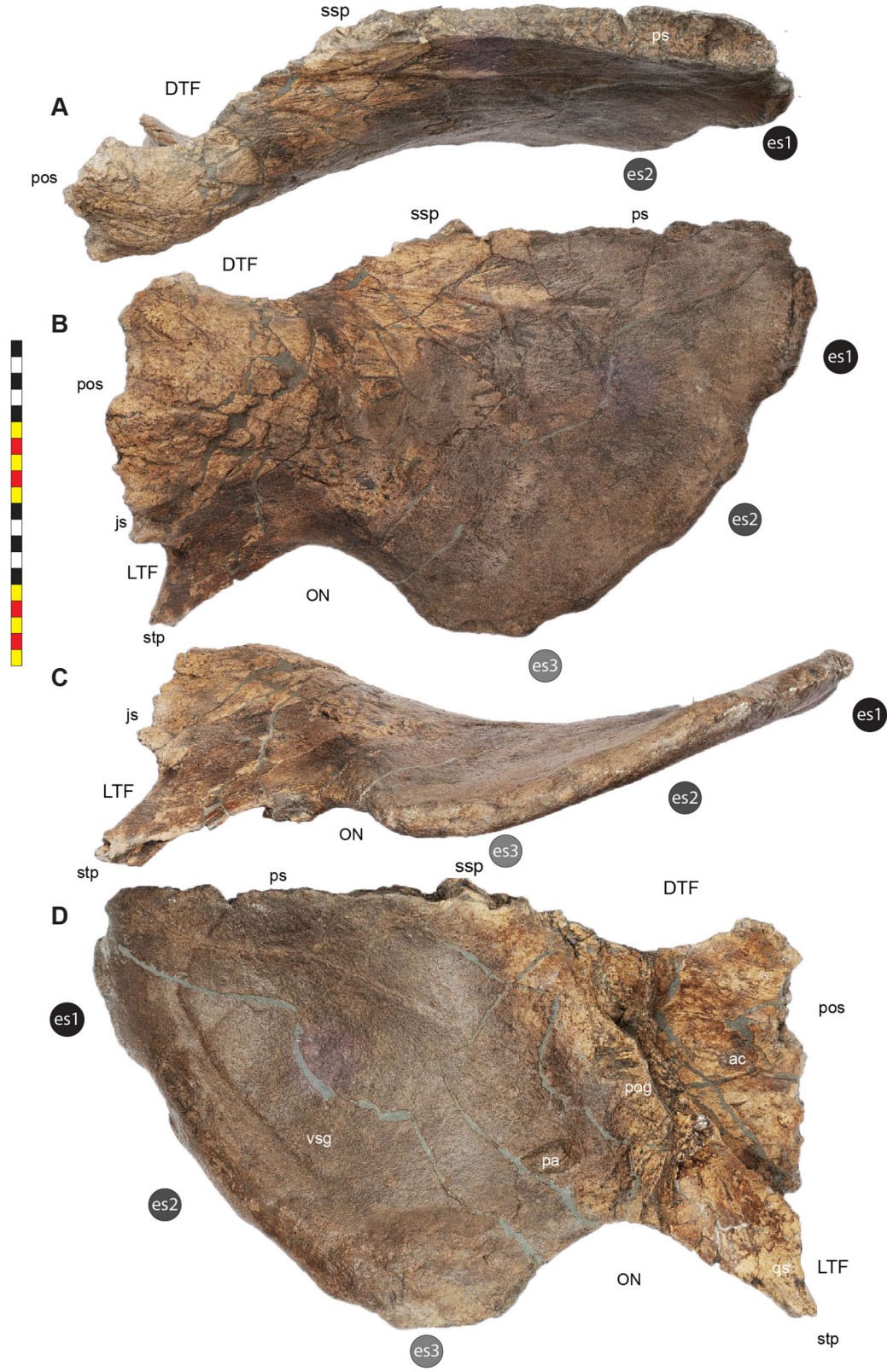

**Figure 14 Left squamosal of *Lokiceratops rangiformis* n. gen et n. sp. (EMK 0012).** Photographs of the left squamosal in dorsal (A), lateral (B), ventral (C) and posteroventral (D) views. Note that all elements were photographed on a flat surface and contain parallax and plastic deformation as found in the quarry.

**Figure 14** (continued)
*Osteological abbreviations*: ac, adductor channel; DTF, dorsotemporal fenestra; es1–es3, episquamosal 1–3; js, jugal suture; LTF, lateratemporal fenestra; ON, otic notch; pa, pathology; pog, paroccipital groove; pos, postorbital suture; ps, parietal suture; qs, quadrate suture; ssp, squamosal step; stp, squamosal temporal process; vsg, ventral squamosal groove. Photos by Marcus Donivan. Scale bar equals 20 cm.

medialmost margins of the episquamosals, but this arcuate line does not follow the crenulations of each episquamosal.

The squamosal differs from the elongated, "sickle-shaped" form of chasmosaurines; the rectangular shape of *Protoceratops andrewsi*, *Diabloceratops eatoni* (UMNH VP 16699), and *Machairoceratops cronusi* (UMNH VP 20550). The "fan-shaped" subtriangular form of the lateral squamosal in *Lokiceratops* conforms to the general morphology of other centrosaurines (*e.g.*, *Yehuecauhceratops mudei*, *Crittendenceratops krzyzanowskii*, *Menefeeceratops sealeyi*, *Nasutoceratops titusi*, *Avaceratops lammersi*, *Albertaceratops nesmoi*, *Medusaceratops lokii*, *Wendiceratops pinhornensis*, *Coronaceratops brinkmani*, *Spinops sternbergorum*, *Centrosaurus apertus*, *Sytracosaurus albertensis*, and *Pachyrhinosaurus lakustai* (see *Maiorino et al., 2013*)). The restricted otic notch is similar to the condition where present in all centrosaurines except *Diabloceratops eatoni*, and *Machairoceratops cronusi*.

We recognize two conditions regarding the nature of the step associated with the dorsotemporal fenestra in centrosaurines. There is both an anteroposterior and dorsoventral nature to this stepped-up suture which has been long recognized (*Dodson, 1986*; *Sampson, 1995*; *Sampson et al., 2013*). We recognize a "step" as being present when the lateral portion of the squamosal forming the lateral border of the dorsotemporal fenestra extends farther posteriorly than the area just posterior to the dorsotemporal fenestra. We quantify the step as being "slightly-stepped" when the posterior extension laterally is less than the dorsoventral thickness of the dorsotemporal fenestra but more than in chasmosaurines. We recognize a "slightly-stepped" parietosquamosal suture in *Protoceratops andrewsi*, *Diabloceratops eatoni*, *Machairoceratops cronusi*, *Menefeeceratops sealeyi*, *Xenoceratops formostensis*, *Nasutoceratops titusi*, and *Avaceratops lammersi*. *Crittendenceratops krzyzanowskii*, *Albertaceratops nesmoi*, *Medusaceratops lokii*, *Wendiceratops pinhornensis*, *Coronaceratops brinkmani*, *Spinops sternbergorum*, *Centrosaurus apertus*, *Sytracosaurus albertensis*, and *Pachyrhinosaurus lakustai* all have a large-stepped parieosquamosal suture with lateral posterior expansion greater than the anteroposterior thickness of the parietosquamosal suture.

The dorsal surface of the lateral squamosal blade *Lokiceratops* lacks the prominent rounded ridge that is typical for basal centrosaurines, such as *Avaceratops*, *Albertaceratops*, and *Wendiceratops*. This is the same ridge that forms a peaked ridge in *Nasutoceratops*. It also lacks the more ventrally placed pronounced dorsal otic ridge present in *Menefeeceratops sealeyi*, *Crittendenceratops krzyzanowskii*, and *Yehuecauhceratops mudei*.

**Parietal**—As in all other ceratopsids, the parietal is an unpaired median element. Ontogenetic evidence suggests that this is the case throughout ontogeny

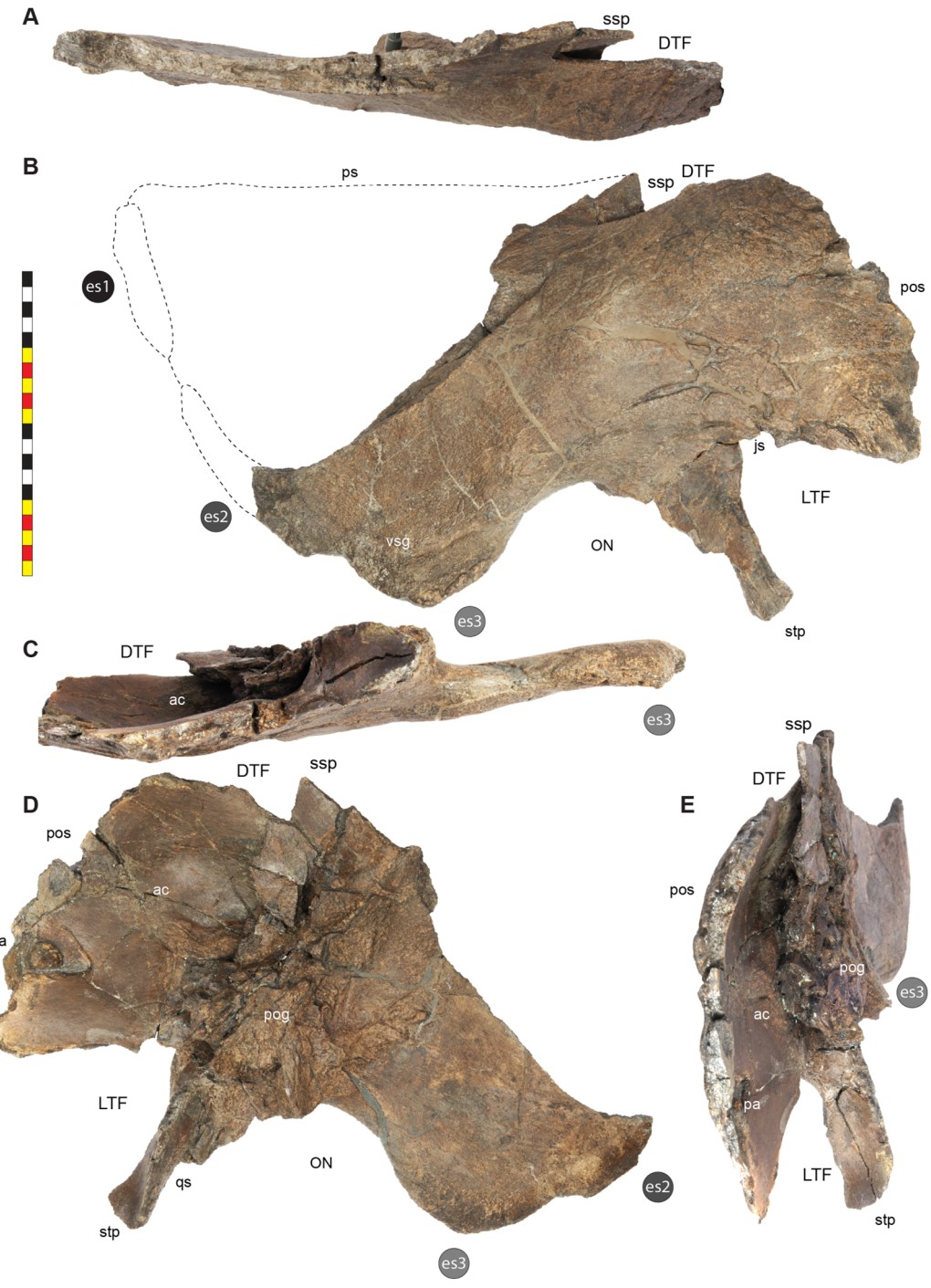

**Figure 15 Right squamosal of *Lokiceratops rangiformis* n. gen et n. sp. (EMK 0012).** Photographs of the right squamosal in dorsal (A), lateral (B), ventral (C), posteroventral (D), and anterior (E) views. Note that all elements were photographed on a flat surface and contain plastic deformation as found in the quarry. Outline of missing portion in B is traced from the complete left squamosal for reference. *Osteological abbreviations*: ac, adductor channel; DTF, dorsotemporal fenestra; es1–es3, episquamosal 1–3; js, jugal suture; LTF, laterotemporal fenestra; ON, otic notch; pa, pathology; pog, paroccipital groove; pos, postorbital suture; ps, parietal suture; qs, quadrate suture; ssp, squamosal step; stp, squamosal temporal process; vsg, ventral squamosal groove. Photos by Marcus Donivan. Scale bar equals 20 cm.

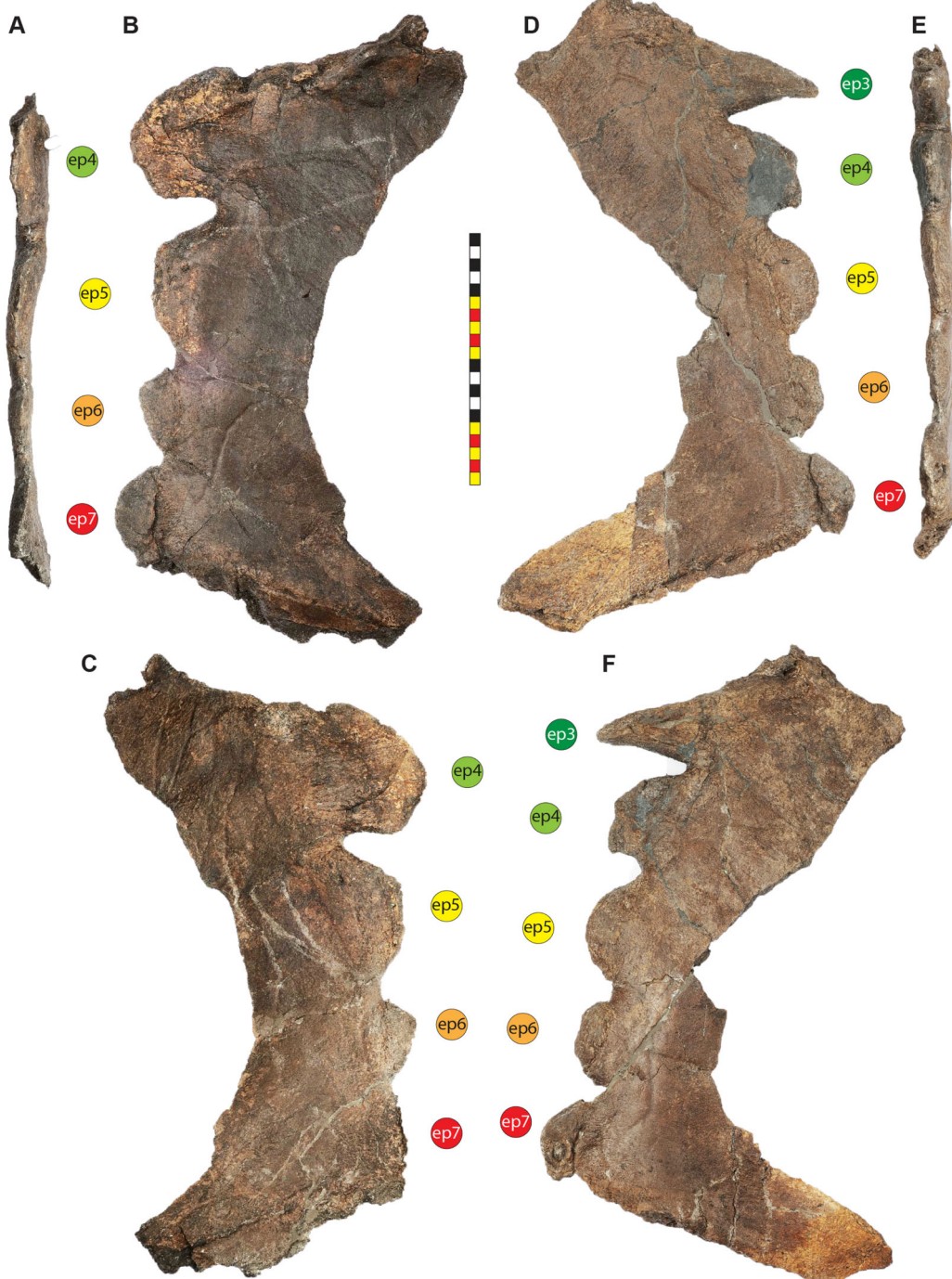

**Figure 16 Lateral parietal bars of *Lokiceratops rangiformis* n. gen et n. sp. (EMK 0012).** Photographs of the right lateral parietal bar of EMK 0012 in lateral (A), dorsal (B), and posteroventral (C) views. Photographs of the left lateral parietal bar of EMK 0012 in dorsal (D), lateral (E), and posteroventral (F) views. Note the imbrication of epiparietals ep3–ep7 in A and E. *Osteological abbreviations*: ep3–ep7, epiparietals 3–7. Photos by Marcus Donivan. Scale bar equals 20 cm.

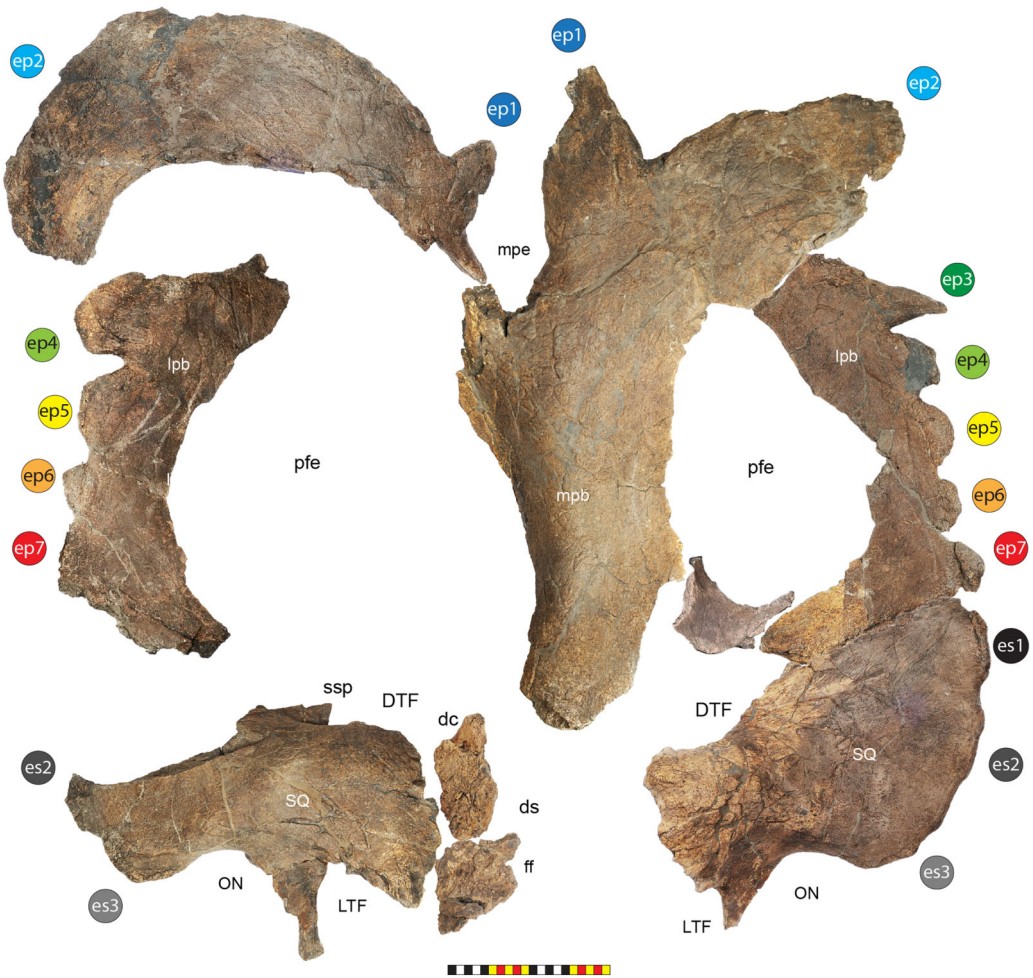

**Figure 17 Elements of the parietosquamosal frill of *Lokiceratops rangiformis* n. gen et n. sp. (EMK 0012) in dorsal view.** Photographs of the parietal, squamosal, and parts of the postorbital and frontal of EMK 0012 in dorsal view. Note that all elements were photographed on a flat surface and contain parallax and plastic deformation as found in the quarry. This is especially true of the midline parietal bar, in which the right side is curled ventrally. The left side of the frill is complete, and each piece clicks into place with the rest of the frill as does the distal right side of the frill. *Osteological abbreviations*: dc, dorsotemporal channel; ds, dorsocranial sinus; DTF, dorsotemporal fenestra; ep1–ep7, epiparietals 1–7; es1–es3, episquamosals 1–3; ff, frontal fontanelle; lpb, lateral parietal bar; LTF, laterotemporal fenestra; mpb, midline parietal bar; mpe, midline parietal embayment; ON, otic notch; pfe, parietal fenestra; SQ, squamosal; ssp, squamosal step. Photos by Marcus Donivan. Scale bar equals 20 cm.

(*Goodwin et al., 2006*; *Horner & Goodwin, 2006*; *Brown, Russell & Ryan, 2009*; *Scannella & Horner, 2010*; *Currie et al., 2016*; *Norell et al., 2020*). Representative parts of the parietal preserved in *Lokiceratops* include a complete left lateral parietal bar, most of the right lateral parietal bar, most of the midline bar, the nearly complete outline of the left parietal fenestra, parts of the parietal between the dorsotemporal fenestra, part of the right dorsotemporal channel, and part of the proximal (anterior) right portion bordering the frontal fontanelle (Figs. 3–5, 7, 16–18). While there is some plastic deformation, especially along the midline parietal bar, the left squamosal articulates perfectly with the complete

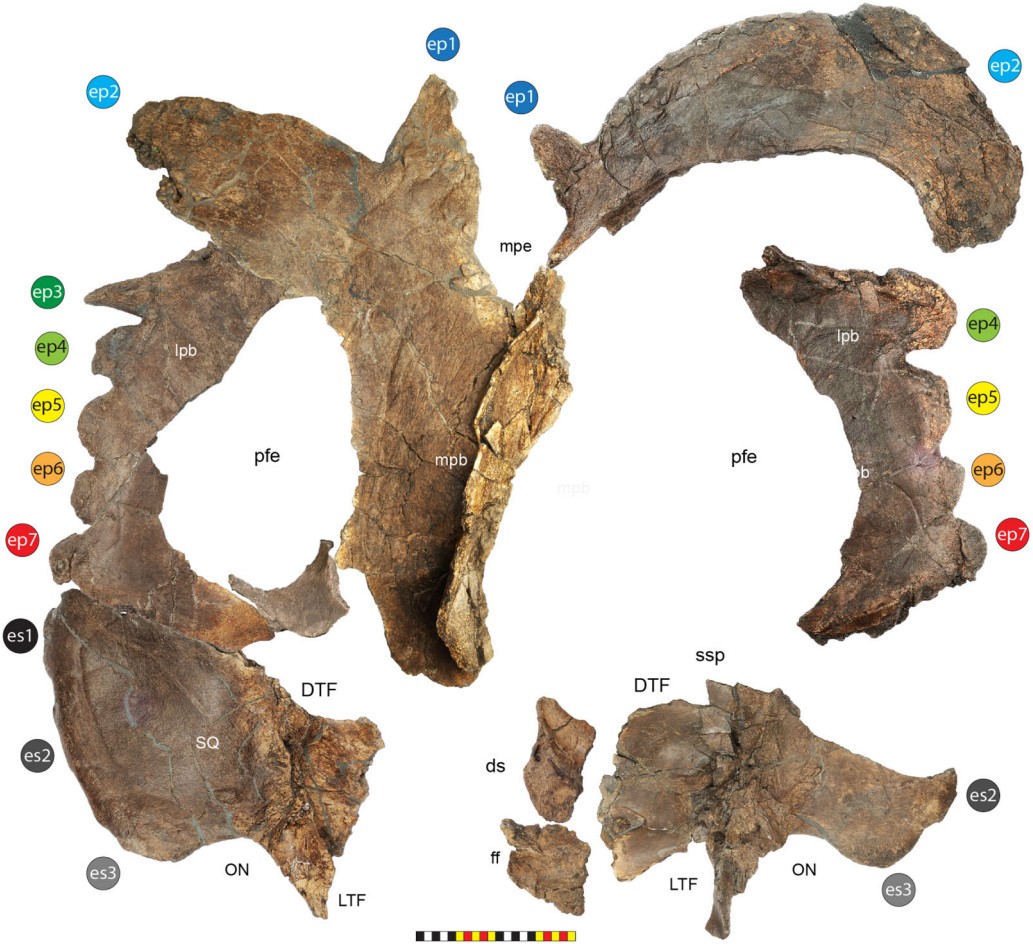

**Figure 18 Elements of the parietosquamosal frill of *Lokiceratops rangiformis* n. gen et n. sp. (EMK 0012) in ventral view.** Photographs of the parietal, squamosal, and parts of the postorbital and frontal of EMK 0012 in ventral view. Note that all elements were photographed on a flat surface and contain parallax and plastic deformation as found in the quarry. This is especially true of the midline parietal bar, in which the right side is curled ventrally. The left side of the frill is complete, and each piece clicks into place with the rest of the frill as does the distal right side of the frill. *Osteological abbreviations*: dc, dorsotemporal channel; ds, dorsocranial sinus; DTF, dorsotemporal fenestra; ep1–ep7, epiparietals 1–7; es1-es3, episquamosals 1–3; ff, frontal fontanelle; lpb, lateral parietal bar; LTF, laterotemporal fenestra; mpb, midline parietal bar; mpe, midline parietal embayment; ON, otic notch; pfe, parietal fenestra; SQ, squamosal; ssp, squamosal step. Photos by Marcus Donivan. Scale bar equals 20 cm.

left parietal, as does the distal right parietal with the midline parietal bar, preserving a complete median embayment. The parietal contacts the frontals anteriorly, the postorbital anterolaterally, a small portion of the squamosal at the anteromedial corner of the dorsotemporal fenestra, and the squamosal lateral to the dorsotemporal fenestra.

The suture between the parietal and squamosal lateral to the dorsotemporal fenestra is visible on both sides but the right side preserves the posterior portion of the squamosal fused to it. The parietal forms the ventral floor of the dorsotemporal fenestra medial to the adductor chamber on the ventral squamosal. The overall shape of the parietal is elongated rather than round with the widest part of the parietal excluding epiparietals is at the

anterolateral contact with the squamosal. The midline bar is broad and rounded dorsally in transverse section with a lenticular transverse cross-section. A distinct medial embayment separates the epiparietal positions ep1 from each other. The posteriormost extent of the parietal is at the base of each ep2. The overall parietal has a slightly concave dorsal surface when viewed laterally and a gently convex one when viewed posteriorly. Much of the borders of the parietal fenestra are preserved indicating a posteriorly elongate shape that is 348 mm long by 160 mm wide on the left side and 340 mm long by 185 mm wide on the right side.

The dorsal surface texture of the parietal is heavily rugose with lightly-to-deeply inscribed vascular grooves (many oriented longitudinally), and numerous fine pits that are characteristic of adult centrosaurines (see below). The broad midline bar exhibits the most rugose surface texture compared to the lateral and transvers bars. The ventral surface is also relatively rugose with many vascular grooves. The texture of the area around the parietal fenestrae is smooth and continuous with the dorsal floor of the dorsotemporal fenestrae anteriorly and the dorsotemporal channels medially.

**Epiossifications of the Frill**—The left side of the frill of *Lokiceratops* (EMK 0012) preserves a full complement of three epiossifications on the squamosal and seven epiossifications on the parietal (Figs. 3–5, 12, 14–18). The right side of the frill preserves two epiossifications (the section where es1 should be is missing, and there is space for three positions in life) on the squamosal and six epiossifications on the parietal (ep3 is interpreted as missing).

The left squamosal has three episquamosals on the margin of the posterior (distal) process. Each is low and crescentic. The proximal most episquamosal, es3, is 123 mm long and 10 mm to its apex on the dorsal surface and 60 mm to its apex on the ventral surface, es2 is 106 mm long by 10 mm wide to its apex on the dorsal surface and 40 mm to its apex on the ventral surface, and the distalmost episquamosal es1 is 95 mm long by 10 mm wide to its apex on the dorsal surface and 30 mm to its apex on the ventral surface. The right squamosal preserves es3 and half of es2. Es3 is 113 mm long by 40 mm wide on the dorsal surface and 50 mm to its apex on the ventral surface, es2 is 102 mm long. All of the episquamosals are completely fused to the margin of the squamosal. Each of the episquamosals is directed in the general plane of the frill, but the ventral edge of the squamosal is slightly curved and the episquamosals are positioned more on the edge of the squamosal, and barely distinguished from the squamosal on the dorsal surface. They are much better demarcated on the ventral surface, suggesting that part of the episquamosal is wrapping ventrally onto the ventral surface of the squamosal. The episquamosals are faintly imbricated with the posterior edge of each episquamosal more dorsally positioned than the anterior edge of the following episquamosals, in the distalmost episquamosals es1 and es2.

There is no evidence of the presence of an epiparietosquamosal on either side of the frill of *Lokiceratops*. The anterior edge of ep7 barely touches the parietosquamosal suture but does not cross the suture. *Machairoceratops* and *Medusaceratops* also lack evidence of an epiparietosquamosal. An epiparietosquamosal occurs in *Diabloceratops*, *Avaceratops*,

*Xenoceratops, Wendiceratops, Centrosaurus, Styracosaurus, Stellasaurus, Einiosaurus,* and *Pachyrhinosaurus lakustai.*

The left parietal preserves seven epiparietals and the right parietal preserves a total of six epiparietals. The lateral edge preserving four epiparietals on the right side is nearly the same length as the portion preserving five epiparietals (ep3 through ep7) on the left side of the frill (Fig. 16), presenting some uncertainty whether the right ep3 is missing or whether there are only six epiparietals on the right side of the frill. The loose fit between the distal right portion of the parietal and the midline bar of the parietal leaves a gap that could accommodate a missing ep3. Or, as an alternate interpretation, the relatively long right epiparietal (interpreted here as ep4) is represented on the left side as the smaller ep3 and ep4 ossifications, a level of bilateral variability not uncommon in ceratopsids (*e.g., Styracosaurus* UALVP 55900 (Holmes et al., 2020)), and suggests a 'normal' count of six epiparietals per side. Since the left parietal is complete from the parietosquamosal suture to the posterior midline embayment, we reconstruct both sides to have the seven epiparietal positions preserved on the left parietal. The following description of individual epiparietals assumes that the right ep3 is missing.

*Lokiceratops* lacks a midline epiparietal (ep0). On the left parietal, ep1 is an uncurved, posteriorly directed epiossification directed in the plane of the parietal along the posterior margin of parietosquamosal frill. The apex of ep1 is broken, and it likely extended farther than the preserved epiossification. Ep1 is 110 mm wide at the base, 132 mm long from the base of the epiossification to its preserved apex on the dorsal surface of the frill, and 164 mm long on the ventral surface of the frill. The surface of ep1 is moderately rugose, ornamented with shallow neurovascular grooves. In contrast, the right ep1 is complete and much smaller measuring 53 mm wide at the base, 67 mm long from the base of the epiossification to its preserved apex on the dorsal surface of the frill and 80 mm long on the ventral surface of the frill.

Each ep2 forms a large blade-like ornament solidly fused into the posterior surface of the parietal. Both ep2 blades are oriented in the plane of the frill and extend posterolaterally. The right ep2 is complete and extends over 518 mm in curvilinear length, with a base that forms 310 mm of the posterior portion of the frill. The right ep2 is 142 mm wide at its narrowest point about midway along blade of ep2. The left ep2 is incomplete distally and what is preserved extends over 284 mm in curvilinear length and 265 mm of the posterior portion of the frill along its proximal base. The maximum thickness of the blade of ep2 is 47 mm.

Epiparietal ep3 is preserved on the left side of the frill and forms a laterally elongated triangular spike oriented in the plane of the frill. The distal tip is broken but the edges of this epiparietal are complete demonstrating that this epiparietal is not part of the base of ep2. Ep3 is 54 mm wide at the base, 92 mm long from the base of the epiossification to its preserved apex on the dorsal surface of the frill and 108 mm long on the ventral surface of the frill.

Epiparietal ep4 is preserved on both sides of the frill and forms a moderately elongated triangular spike. It differs from all subsequent epiparietals which are more typical in shape to those of other general proximal epiparietals in centrosaurines. The distal tip is broken

on the left ep4. The left ep4 is 81 mm wide at the base, is 116 mm long from the base of the epiossification to its preserved apex on the dorsal surface of the frill, is 64 mm long on the ventral surface of the frill and is oriented in the plane of the frill. The right ep4 is complete. The right ep4 is 104 mm wide at the base, 52 mm long from the base of the epiossification to its preserved apex on the dorsal surface of the frill and 101 mm long on the ventral surface of the frill. The right ep4 is distinctly oriented posteriorly and is not in the plane of the frill. The asymmetry between left and right may indicate that the right side only had 6 epiparietals, but given the asymmetry seen in ep1, position ep4 may just be variable.

Epiparietal ep5 is preserved on both sides of the frill and bears a generally crescentic shape common in centrosaurines. Ep5 is oriented laterally in the plane of the frill although it is imbricated. The left ep5 is 84 mm wide at the base, is 46 mm long from the base of the epiossification to its preserved apex on the dorsal surface of the frill, is 59 mm long on the ventral surface of the frill and is oriented in the plane of the frill. The right ep5 is 93 mm wide at the base, 46 mm long from the base of the epiossification to its preserved apex on the dorsal surface of the frill and 59 mm long on the ventral surface of the frill.

Epiparietal ep6 is preserved on both sides of the frill and bears the same generally crescentic shape common in centrosaurines. Ep6 is oriented laterally in the plane of the frill although it is imbricated. The left ep6 is 59 mm wide at the base, is 37 mm long from the base of the epiossification to its preserved apex on the dorsal surface of the frill, is 24 mm long on the ventral surface of the frill and is oriented in the plane of the frill. The right ep6 is 65 mm wide at the base, 43 mm long from the base of the epiossification to its preserved apex on the dorsal surface of the frill, and 43 mm long on the ventral surface of the frill.

Epiparietal ep7 is preserved on both sides of the frill and forms a generally crescentic shape common in centrosaurines. Epiparietal ep7 is oriented laterally in the plane of the frill although it is imbricated. The left ep7 is 62 mm wide at the base, is 31 mm long from the base of the epiossification to its preserved apex on the dorsal surface of the frill, is 37 mm long on the ventral surface of the frill and is oriented in the plane of the frill. The right ep7 is 72 mm wide at the base, 28 mm long from the base of the epiossification to its preserved apex on the dorsal surface of the frill and 35 mm long on the ventral surface of the frill. The epiparietals are faintly imbricated with the posterior edge of each epiparietal more dorsally positioned than the anterior edge of the successive epiparietals in positions ep3 through ep7.

The surfaces of epiparietals ep1 and ep2 exhibit a heavily rugose texture with pits and a number of lightly-to-deeply inscribed grooves. Epiparietal ep3 has a series of grooves that follow the long axis of the epiparietal. Epiparietals ep4–ep7 are textured with rugose pits similar to other centrosaurines. Epiparietal ep7 on the left side has a deep pit in the ventral surface which could be pathologic.

*Lokiceratops* lacks the midline parietal epiossification (ep0) present in *Avaceratops*, *Nasutoceratops*, and *Sinoceratops*. The presence of seven epiossification loci in *Lokiceratops* is only shared with the basal centrosaurines *Diabloceratops* and *Nasutoceratops*. *Machairoceratops* has one epiparietal. *Avaceratops* and *Wendiceratops* have 4 epiparietals. *Albertaceratops* and *Medusaceratops* have 5 epiparietals. *Xenoceratops*,

*Sinoceratops* and *Coronoceratops* have 6 epiparietals. We interpret the more derived centrosaurines *Styracosaurus ovatus*, *Einiosaurus*, *Achelousaurus*, *Pachyrhinosaurus canadensis*, *Pachyrhinosaurus perotorum*, and *Pachyrhinosaurus lakustai* to subsequently have lost position ep1, which independently moved to the dorsum of the frill in *Spinops* and *Centrosaurus*. This would indicate that at least *Einiosaurus*, *Achelousaurus*, *Pachyrhinosaurus canadensis*, and *Pachyrhinosaurus lakustai* had an epiparietal in the equivalent position as ep7 even though these animals only have six total epiparietals (*Clayton et al., 2009*).

The shape of the epiparietals in *Lokiceratops* is distinct from the patterns present in other centrosaurines. Blade-like epiparietals are present in *Xenoceratops*, *Albertaceratops*, *Medusaceratops*, *Wendiceratops*, and *Sinoceratops*. Large, blade-like epiparietals at position ep2 are present in *Xenoceratops*, *Medusaceratops*, *Wendiceratops*, and *Sinoceratops* but not in *Albertaceratops*. Blade-like epiparietals at position ep2 that are oriented in the plane of the frill are present in *Xenoceratops* and *Medusaceratops*, but not in *Wendiceratops*, *Sinoceratops*, and *Albertaceratops*. Elongated spikes at ep1 position distinguish *Lokiceratops* from *Medusaceratops*, and their orientation in the plane of the frill differentiates *Lokiceratops* from *Wendiceratops*.

Many centrosaurines exhibit asymmetry from side to side in epiparietal number and morphology, similar to the differences in cervids (*Ditchkoff & deFreese, 2010*) and specifically caribou (*Miller, 1986*). The extreme asymmetry is ep1 and ep2 in *Lokiceratops* is unusually pronounced when compared to other centrosaurines such as *Coronoceratops brinkmani*, *Centrosaurus*, *Styracosaurus albertensis*, *Einiosaurus procurvicornis*, and *Pachyrhinosaurus lakustai*. Without additional material, it is impossible to determine if this asymmetry is characteristic of *Lokiceratops* or if the degree of asymmetry varies within the species or across ontogeny.

## Additional cranial elements

**Braincase**—Much of the braincase is preserved in *Lokiceratops* (EMK 0012), including the basioccipital, exoccipitals, prootics, and laterosphenoids, though sutures between preserved elements are largely obliterated and distal extremities of most elements are missing (Figs. 10, 19). As in other ceratopsids, the basioccipital contributes to the floor of the braincase anteriorly, and the spherical occipital condyle posteriorly. The occipital condyle is externally smooth, offset from the main braincase by a narrowed neck. Dorsally, the exoccipitals are tightly fused to the neck of the basioccipital on each side of the posterodorsally open foramen magnum, though the exoccipitals do not appear to contribute to the formation of the articular surface of the occipital condyle, terminating at the raised border of the articular surface. Ventrally, the tubera of the basioccipital are not entirely preserved in EMK 0012, though the dorsal portions indicate their presence in *Lokiceratops*. The exoccipitals meet dorsal to the foramen magnum to complete its external borders. Laterally, the paroccipital processes are missing, though their contacts with the squamosals laterally indicate that they would have been similar in morphology to other ceratopsids. The ventral portion of the supraoccipital sits on the midline dorsal to the exoccipitals, preserving the ventral base of a wide, midline crest separating two nuchal

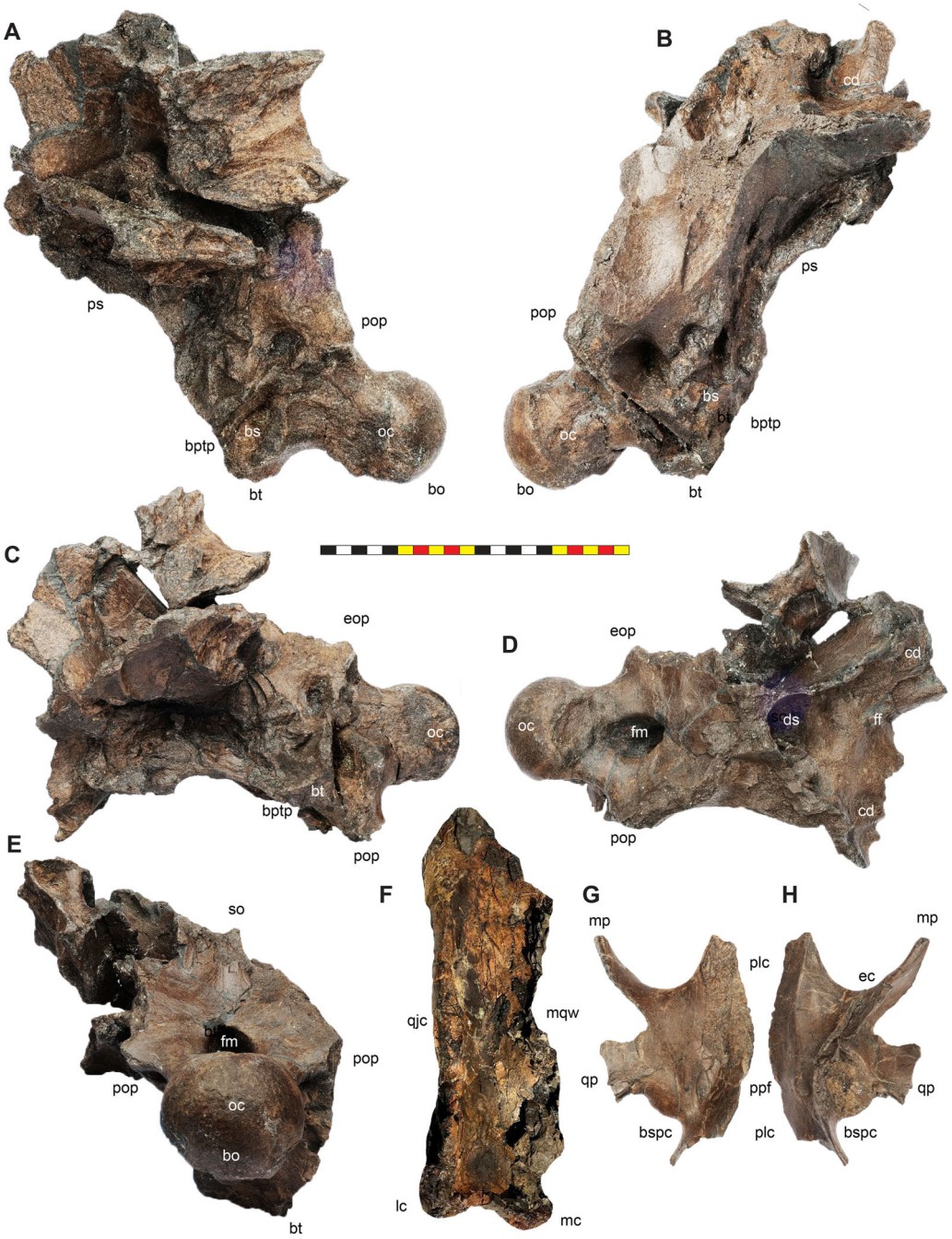

**Figure 19 Other cranial elements of *Lokiceratops rangiformis* n. gen et n. sp. (EMK 0012).** Photographs of the braincase of EMK 0012 in left lateral (A), right lateral (B), ventral (C), dorsal (D), and posterior (E) views. Photographs of the left pterygoid of EMK 0012 in dorsal (F) and ventral (G) views. Photograph of the right quadrate of EMK 0012 in anterior (H) view in the Loki Quarry. The current whereabouts of this element are unknown. *Osteological abbreviations:* bo, basioccipital; bs, basisphenoid; bt, basal tubera; bptp, basipterygoid process; bsspc, pasisphenoid process contact; cd, cornual diverticulum; ds, dorsocranial sinus; ec, eustachian canal; eop, exoccipital process; ff, frontal fontanelle; fm, foramen magnum; lc, lateral condyle; mc, medial condyle; mp, maxillary process; mqw, medial quadrate wing; oc, occipital condyle; plc, palatine contact; pop paroccipital process of the otoccipital; ppf, pterygopalatine foramen; ps, parasphenoid; qjc, quadratojugal contact; qp, quadrate process; so, supraoccipital. Photos A-G by Marcus Donivan. Photo H courtesy of EMK. Scale bar equals 20 cm.

fossae. Though the left side is badly crushed, the right side of the braincase is largely intact, consisting of the prootic posteriorly and the laterosphenoid anteriorly, and preserving the external exits for the cranial nerves. A deep fenestra ovalis is the most conspicuous opening into the braincase on this side. The dorsal contact between the braincase and the parietals is incomplete. Anteriorly, the braincase would have contacted the postorbitals dorsolaterally, and the frontals dorsally.

**Pterygoid**—The left pterygoid of *Lokiceratops* (EMK 0012) is preserved (Figs. 10, 19). The pterygoid contacts the quadrate laterally and the palatines medially. The eustachian canal is present on the medial surface as in *Avaceratops lammersi* (*Penkalski & Dodson, 1999*). The pterygoid is missing part of the quadrate wing that articulates with the medial wing of the quadrate. The eustachian canal is kinked halfway along its length. The ascending process is mostly preserved where the pterygoids meet alongside the palatines medially. At the anterior end of the element is the process that touches the medial posterior surface of the maxilla, and the angle between this and the quadrate process forms the beginning of the eustachian canal. The medial surface of this maxillary process has an articular surface for the palatine that is interrupted by a smoother surface interpreted as the pterygopalatine foramen as in *Triceratops horridus* (YPM 1821, *Hatcher, Marsh & Lull, 1907*). The surfaces for articulation with the basisphenoid process are damaged. Overall, the pterygoid is similar to that of *Diabloceratops eatoni* (UMNH VP 16699), *Avaceratops lammersi* (ANSP 15800 *Penkalski & Dodson, 1999*), *Centrosaurus apertus* (ROM 767, 43219), and *Pachyrhinosaurus lakustai* (TMP 1989.55.249; *Currie, Langston & Tanke, 2008*).

**Quadrate**—The right quadrate of *Lokiceratops* (EMK 0012) was discovered in the quarry, but was not delivered with the specimen when it was acquired by EMK. We describe it based on a photo reposited at the EMK that was taken in the quarry. The quadrate is similar to all known quadrates in centrosaurine ceratopsids. The quadrate has an elongate shaft that is straight in lateral view with a convex anterior surface and a concave posterior surface. Two ventral condyles would have articulated with the articular on the mandible. The medial condyle extends further ventrally than the lateral condyle. The lateral surface has a scar for the articulation of the quadratojugal that spans from just dorsal to the lateral condyle to most of the height of the element. A medial wing preserves part of the contact with the pterygoid.

**Dentition**—The left maxilla of *Lokiceratops* (EMK 0012) preserves teeth (Figs. 8, 10–12, 20). The teeth are similar to those of other ceratopsids with a wear facet on the lingual surface of the most erupted teeth. Unworn teeth are leaf-shaped with enamel present on the labial surface. Each unworn tooth preserves a central ridge on the labial surface that leads to the apex of each tooth with two or three secondary ridges anterior and posterior to this central ridge. Each tooth has 15 to 20 denticles on each side of the central ridge.

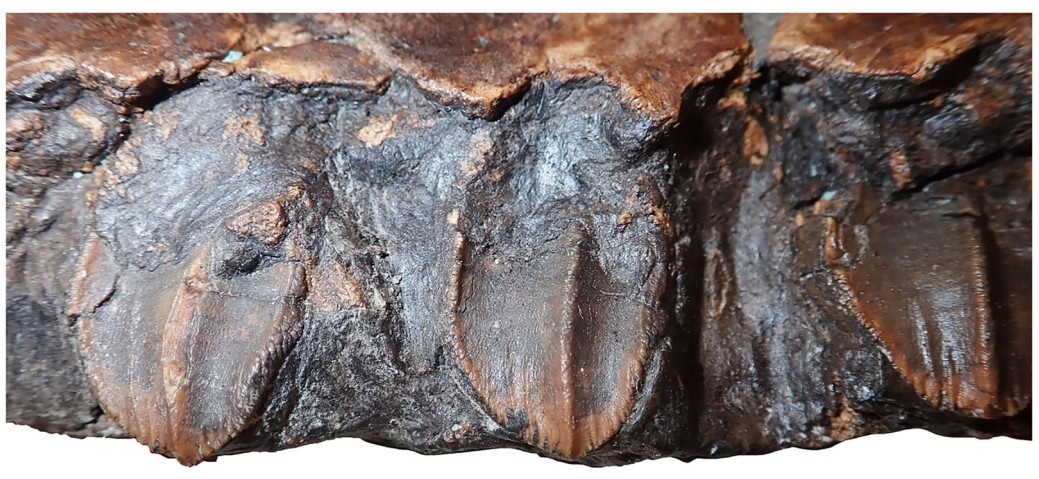

**Figure 20 Maxillary teeth of *Lokiceratops rangiformis* n. gen et n. sp. (EMK 0012).** Photograph of right maxillary teeth of EMK 0012 in semi occlusal view. Photo by Mark Loewen. Scale bar equals 10 mm.

## General description of the axial skeleton

**Posterior Cervical Vertebrae**—A posterior cervical vertebra is preserved in the holotype of *Lokiceratops* (EMK 0012) (Fig. 21), likely Ce8 or Ce9 based on its overall morphology. This vertebra is post-depostionally deformed, but several morphologic characters can be distinguished in this element. The anterior and posterior faces of the centrum are roughly amphiplatyan and the anteroposterior length of the centrum is about one third the height of the centrum. The neural arch with the right transverse process is preserved but the neural spine is missing. The neural canal is about one third of the height of the centrum alone. The prezygapophyses and postzygapophyses are located above the neural canal. The prezygapophyses are directed dorsomedially and the articular facets of the postzygapophyses are directed ventrolaterally. The parapophysis (the articular facet for the capitulum of the rib) is preserved on both sides at the base of the neural arch. The diapophysis is preserved on the left neural arch, near the lateral extent of the transverse process. The transverse process is roughly horizontal. There is some indication that the neural spine would have been posteriorly inclined. Part of the cervical rib is fused to the right side of the centrum, ventral to the parapophysis. The position of the parapophysis is similar to the position on the neural arch in *Styracosaurus albertensis* (CMN 344) for cervical ce9 or possibly ce8 (*Holmes, Ryan & Murray, 2005*). *Lokiceratops* differs in the horizontal angle of the transverse process compared to the 40-degree angle present in the posterior cervicals of *Styracosaurus albertensis*.

**Synsacral Dorsosacral Vertebrae**—The centrum of the penultimate posterior dorsal vertebrae (ds1) is partially preserved and fused to the ultimate posterior dorsal vertebrae (ds2) within the fused synsacrum of *Lokiceratops* (EMK 0012) (Fig. 21). The dorsal extent of the neural spines was lost during excavation. The transverse processes of the ultimate posterior dorsal sd2 extend toward each ilium but do not extend to the ilia. The lateral

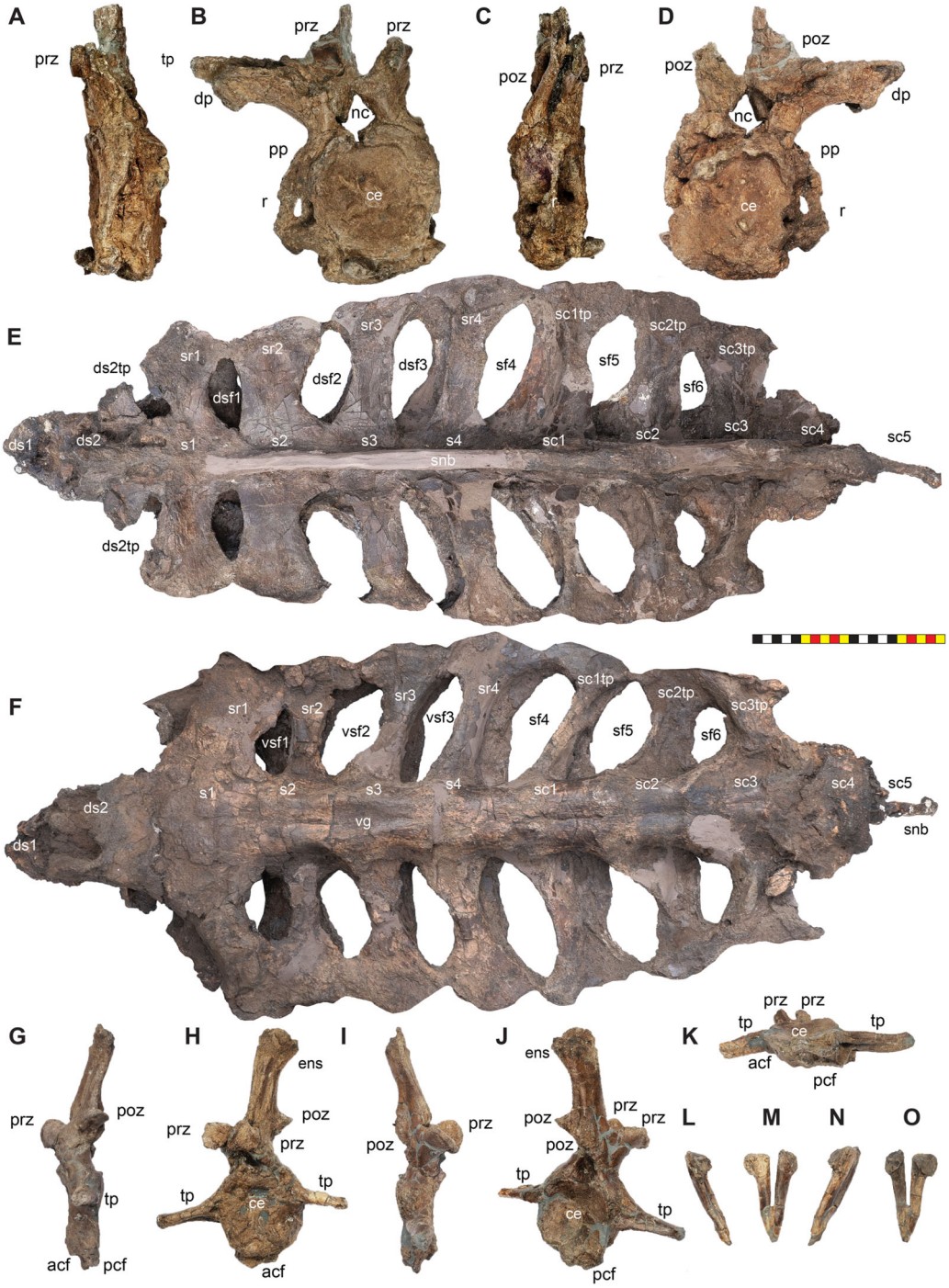

**Figure 21 Axial elements of *Lokiceratops rangiformis* n. gen et n. sp. (EMK 0012).** Photographs of posterior cervical (ce9) of EMK 0012 in (A) left lateral, (B) anterior, (C) right lateral, and (D) posterior views. Part of the right rib shaft is fused to the right side of the vertebra. Photographs of synsacral vertebrae including dorsosacrals, true sacrals, and sacrocaudals in (E) dorsal and (F) ventral views. Photographs of free anterior dorsal vertebrae of EMK 0012 in (G) left lateral, (H) anterior, (I) right lateral, (J) posterior, and (K) ventral with anterior up views. Chevron of EMK 0012 in (L) left lateral, (M) anterior, (N) right lateral, and (O) posterior views. Each of these vertebrae has been plasticly deformed either anteroposteriorly (A–D, G–O) or dorsoventrally (E and F). *Osteological abbreviations*: acf, anterior chevron facet; ce, centrum; ds1–2, dorsosacral vertebrae; ds2tp, transverse process for dorsosacral ds2;

**Figure 21** (continued)
dsf1–dsf3; dorsal sacral fenestrae; nc, neural canal; ens, expanded neural spine; pcf, posterior chevron facet; poz, postzygapophysis; prz, prezygapophysis; r, rib shaft; s1–s4, true sacral vertebrae; sd1–5, sacrodorsal vertebrae (anterior dorsals fused into the synsacrum); sd1tp–sd3tp, transverse process for dorsosacral sd1–3; sf4–sf6, sacral fenestra; sr1–sr4, true sacral ribs, tp, transverse processes; vg, ventral groove; vsf1–vsf3, ventral sacral fenestrae between true sacral ribs. Photos by Marcus Donivan. Scale bar equals 20 cm.

extent of the transverse process is unclear as they are broken, but there is a lack of clear evidence of contact for this on the left ilium. The anteroposterior width of the transverse process is narrower than the s1–s4 sacral ribs and the transverse processes of sc1–3. The centra of both posterior dorsal vertebrae are oval in anterior view, but this is likely due to post depositional deformation. The ventral surface of the penultimate posterior dorsal is poorly preserved. The ventral surface of the ultimate posterior dorsal is flat and lacks the ventral groove that begins just posterior to it on sacral vertebrae s1.

**Synsacrum**—A synsacrum is preserved in *Lokiceratops* (EMK 0012) (Fig. 21) and includes a total of ten vertebral centra fused together along with the neural spine of an eleventh vertebra (an anterior caudal). These eleven vertebrae include co-opted posterior dorsal vertebrae, true sacral vertebrae, and anterior caudal vertebrae. We interpret the presence of four true sacrals as seen in basal ornithischians (*Marsh, 1891a*; *Butler, Upchurch & Norman, 2008*; *Maidment & Barrett, 2011*).

We consider, as do others (*Hatcher, Marsh & Lull, 1907*; *Marsh, 1891a*), the first two vertebrae in the synsacrum that lack sacral ribs to be posterior dorsal vertebrae here referred to as dorsosacrals ds1 and ds2. The subsequent seven centra are coossified with the ilia *via* sacral ribs and transverse processes. The first four of these vertebrae support the acetabulum with both ventral and dorsal connections connected by a subvertical lamina that are interpreted as sacral ribs. Sacrals s1 through s4 show evidence of co-ossified neural spines, most of which were destroyed during excavation.

The sacral ribs are fused between the sacral centra s1–s4, and transverse processes from sacrocaudals sc1–sc3 are fused to both ilia producing a set of six oval sacral foramina on either side of the centra and medial to the ilia. The sacral ribs of the four sacral vertebrae have a dorsal and ventral component forming an "I-beam" shape in sagittal cross-section as the horizontal dorsal and ventral surfaces are connected by a subvertical sheet of bone. The dorsal surfaces of the sacral ribs are oriented posterior to the ventral surfaces, so that the ovals 2–6 between the ribs are positioned more posteriorly on the dorsal surface than on the ventral surface. Sacral ribs sr1 and sr2 are the widest anterorposteriorly on the dorsal surface compared to the subsequent sacral ribs. Ventrally, only sacral rib sr1 is considerably wider anterorposteriorly than the successive sacral ribs.

**Synsacral Sacrocaudal Vertebrae**—Four anterior caudal centra (sacrocaudals sc1–sc4) are fused to the true sacral vertebrae in *Lokiceratops* (EMK 0012) (Fig. 21) and a fused neural spine indicates the presence of a fifth sacrocaudal sc5 that was lost during excavation (see above). The transverse processes on sacrocaudal sc1–sc4 are fused to the posterior blade of

the ilia. These have only a flat ventral surface (lacking the dorsal surface of the true sacral ribs) with a vertical lamina of bone dorsally forming an overall inverted "T-shape" in sagittal cross-section.

**Free Proximal Caudal Vertebrae**—*Lokiceratops* (EMK 0012) includes a free anterior caudal vertebra from the proximal portion of the tail (Fig. 21). It has a round centrum that is 15 cm tall and 10 cm wide. The centrum is 3 cm wide anteroposteriorly, but the vertebrae is anteroposterally compressed by post-depositional plastic deformation. The anterior face of the centrum is slightly higher than the posterior face and both surfaces are amphiplatyan. There are lateral transverse processes on each side at about 60% of the height of the centrum. The transverse processes are 5 cm long on the left side and 7 cm long on the right side. Both transverse processes project laterally with only a slight ventral cant in orientation. The neural canal is round and two centimeters tall.
The prezygapophyses jut forward just above the neural canal and the postzygapophyses tilt backwards on the back of the neural spine above the position of the prezygapophysis. The neural spine is 15 cm tall and two cm wide laterally. The neural spine is slightly posteriorly oriented and originates on the anterior half of the centrum. There is a groove on the anterior surface of the neural spine ostensibly for interosseus ligaments. A corresponding groove on the posterior surface presumably for the same purpose extends only from between the postzygapophyses to halfway up the posterior surface. In anterior view, the neural spine is generally two cm wide with a dorsal expansion to a width of three cm. Posteriorly, a facet for the chevrons occurs both anteriorly and posteriorly, with the anterior facet being twice as pronounced as the posterior facet and corresponding to the shape of the articular surfaces of the chevrons (see below).

**Proximal Chevron**—A single chevron is preserved in *Lokiceratops* (EMK 0012) (Fig. 21). It is "V-shaped" in anterior and posterior views with no lateral curvature. It is ten centimeters long and relatively robust. It is interpreted as pertaining to the proximal third of the tail. The overall shape in lateral view has a slight anterior facet for the preceding caudal centrum and an expanded posterior tab dorsally that forms the articular surface of the subsequent caudal centrum. The ventral shaft is straight and tapers ventrally.
The overall shape of the chevron is similar to the conformation in *Styracosaurus albertensis* (CMN 344) with no curvature or distal expansion.

## Appendicular skeleton

*Lokiceratops* (EMK 0012) preserves some postcranial elements, and all are anatomically similar to most known centrosaurine ceratopsids. Postcranial elements are poorly described for centrosaurines in general, with in-depth comprehensive skeletal descriptions confined to *Centrosaurus apertus* (YPM 002015; *Lull, 1933*) and *Styracosaurus albertensis* (CMN 344; *Holmes & Ryan, 2013*). *Nasutoceratops titusi* (UMNH VP 16800) and *Wendiceratops pinhornensis* have some postcranial elements that have been described (*Lund, Sampson & Loewen, 2016*; *Evans & Ryan, 2015*; *Scott, Ryan & Evans, 2023*). Other taxa like *Medusaceratops lokii*, *Coronosaurus brinkmani*, and *Pachyrhinosaurus lakustai* preserve postcrania, but these presently lack description.

**Coracoid**—The right coracoid is preserved in *Lokiceratops* (EMK 0012) (Fig. 22). The element is fused to the scapula with the suture running perpendicular to the overall trend of the scapular blade, and with the scapula forming more than half of the glenoid.
The suture is thickened mediolaterally producing a change in angle from the scapula to the concave lateral surface of the coracoid. The coracoid has an arcuate overall shape from the anterior surface to the dorsal most point where it sutures to the scapula. The concave lateral surface lacks the anterolateral ridge near the confluence of anterior and ventral margins present in psittacosaurs. There is an anteroventral hook formed anterior to, and ventral to the glenoid fossa. The coracoid contributes to roughly one third of the glenoid along with the scapula. The coracoid foramen is anterior to the scapular suture and dorsal to the anterior end of the glenoid. The coracoid foramen pierces the element from medial to lateral. The entire anterior and dorsal surface of the coracoid preserves a one-centimeter rugose rim with rugosity that is possibly for the insertion of *M. scapulocoracoideus* similar to other ornithischian dinosaurs (*Maidment & Barrett, 2011*; *Fearon & Varricchio, 2016*; *Słowiak, Tereshchenko & Fostowicz-Frelik, 2019*). This rugosity continues onto the acromion on the scapula posteriorly. There is a scar anterior to the coracoid foramen on the concave lateral surface that is the origin of *M. biceps*. There is depression on the ventrolateral surface of the anteroventral hook for the origin of *M. coracobrachialis brevis*. The medial surface of the coracoid is flat with some post depositional crushing and the insertion surface for *M. subcoracoideus*. The anterior surface rugosity is pronounced on the medial surface as well. This may indicate that the whole anterior surface supported the insertion of *M. scapulocoracoideus* or it could simply represent intercoracoid ligaments.

**Scapula**—The right scapula is preserved in the holotype of *Lokiceratops rangiformis* (EMK 0012) and is fused to the coracoid (Fig. 22). Dorsally the scapula and coracoid are medially concave, but the degree of curvature of the element as it conformed to the chest cavity is unknown as the element has been post-depositionally flattened after burial. The overall shaft of the scapula is straight. The anterior portion of the scapula includes the suture with the coracoid that is perpendicular to the overall element above the glenoid fossa.
The anterior portion is waisted just anterior to the midpoint of the element and expands anteriorly to the glenoid ventrally and towards the coracoid dorsally, this point is often referred to as the acromion. The dorsal surface from the acromion forward and continuing along to the anterior surface of the coracoid are insertions for *M. deltoideus clavicularis* and *M. supracoracoideus*, but it is unclear where one ends, and the other begins.
The posterior portion of the scapular blade expands distally but is not "paddle-shaped" and the distal end is "squared off". The scapula contributes around two-thirds to the overall glenoid fossa. This is consistent with all ceratopsid dinosaurs in which the dominant element in the glenoid is the scapula when compared to basal ornithischians. The overall glenoid is ventrolaterally directed as in all marginocephalians. The lateral shaft exhibits a scapular spine, a distinct ridge that runs from the glenoid anteriorly and angles across the shaft to the dorsal surface posteriorly. The oblique orientation of the scapular spine is similar to the condition in all ceratopsids except *Torosaurus* and *Triceratops*. The scapular spine marks the separation between the origins of *M. deltoideus scapularis* and *M. teres*

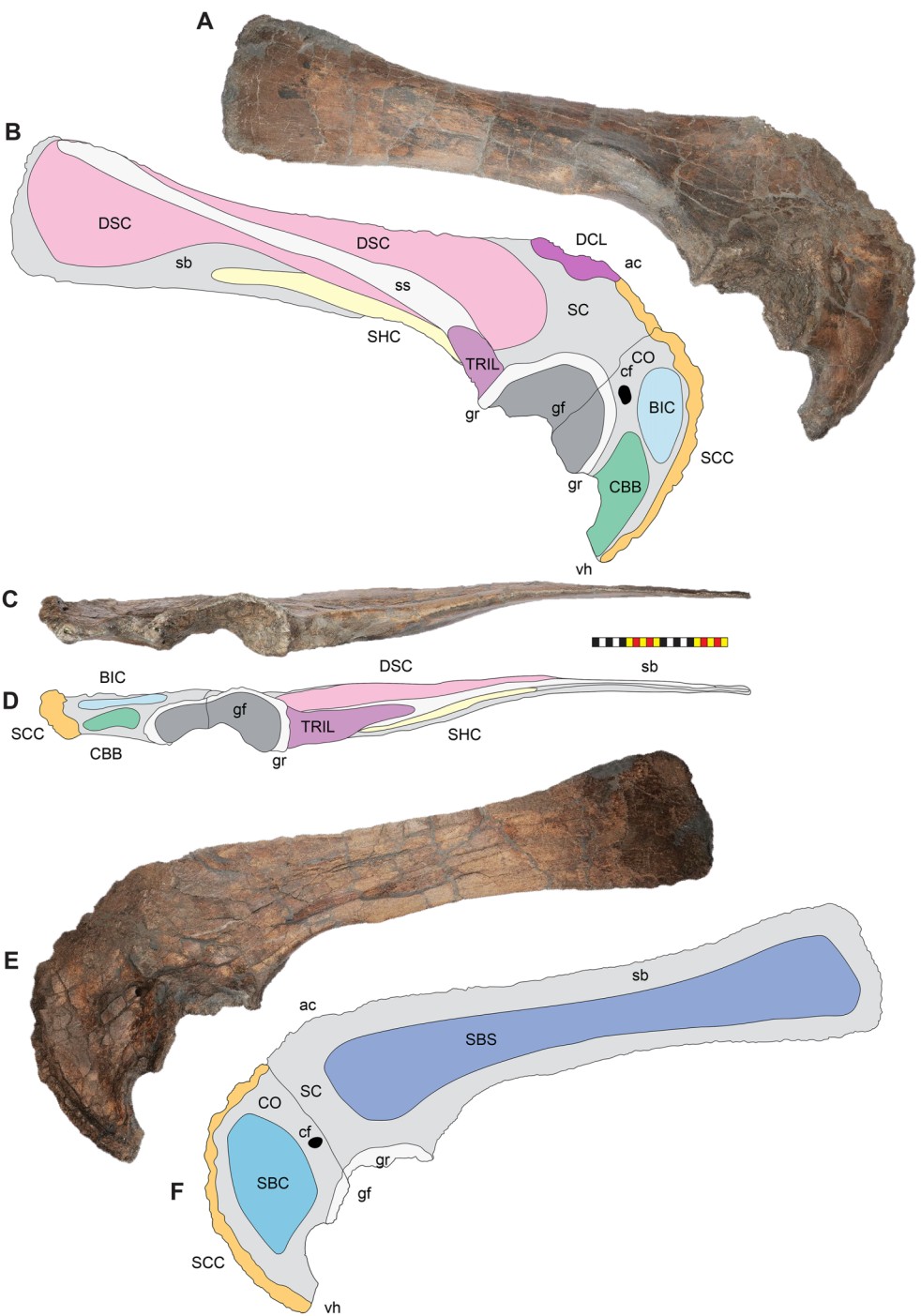

**Figure 22 Appendicular elements of *Lokiceratops rangiformis* n. gen et n. sp. (EMK 0012). (A).** Right scapula and coracoid of EMK 0012 in (A, B) right lateral, (C, D) ventral, and (E, F) medial views. Interpretative line drawings indicate major muscle attachments. *Osteological abbreviations*: ac, acromion; cf, coracoid foramen; CO, coracoid; gf, glenoid fossa; gr, glenoid rim; sb, scapular blade; SC, scapula; ss, scapular spine; scu, scapulocoracoid suture; vh; ventral hook. *Myological abbreviation*: BIC, m. biceps; CBB, m. coracobrachialis brevis; DCL, m. deltoideus clavicularis; DSC, m. deltoideus scapularis, SBC, m. subscapularis; SCC, m. subcoracoideus; SHC, m. scapulohumeralis caudalis; TM, m. teres major; TRIL, m. triceps longus. Photos by Marcus Donivan. Scale bar equals 20 cm.

*major*. The ventral surface of the scapular blade ventral to the scapular spine and posterior to the major origin scar just posterior to the glenoid ridge for *M. triceps longus* is the origin for *M. scapulohumeralis caudalis*. There is a ventrolaterally directed flange on the anteriorlateral surface of this flange that has a distinct muscle scar for the origin of *M. triceps longus*. The medial surface of the scapula exhibits some post depositional crushing on the proximal shaft, but the overall medial surface was flat for the insertion of *M. subscapularis*.

**Ilia**—Both ilia are preserved in *Lokiceratops* (EMK 0012) as part of the fused pelvis (Fig. 23). The pubes and ischia were not fused into the pelvis. The right ilium is missing its anterior blade. The left ilium is complete, with a horizontally positioned anterior blade, a postacetabular lateral expansion, and a vertically oriented posterior blade. The acetabulum is positioned anteriorly on the synsacrum between sacral vertebrae s1–s3. The pubic peduncle is centered on sacral rib sr1 and the ischial peduncle is centered between sacral vertebrae and ribs sr3 and sr4. The ischial peduncle is round and is about the size of the midsacral centra in dimension. The lateral expansion of the ilium is centered on the centrum of sc1 and the posterior blade begins at this position and extends to the midcentrum of sc5.

**Ischia**—Both ischia are preserved in *Lokiceratops* (EMK 0012) (Fig. 23). The right ischium was found in close association to the synsacrum (Fig. 2). The shafts of both ischia are gently curved overall in a ventrally concave manner, and both are distinctly kinked about two-thirds of the length the shaft distally where the two ischia contact each other medially. Proximally the shaft is rounded and transitions to mediolaterally flattened paddle-shape distally. The ischia of *Lokiceratops* shares the rounded, flattened, paddle-shape of its distal end with *Zuniceratops* (MSM P2107 expands to a paddle-shape but is distally incomplete) and definitively with *Wendiceratops* (TMP 2011.051.0037 originally misinterpreted as right ischium with a rectangular distal end (*Evans & Ryan, 2015*; see *Scott, Ryan & Evans, 2023*)). Both *Lokiceratops* and *Wendiceratops* differ in the overall amount of dorsal curvature in the shaft (excluding the distal kink in *Lokiceratops*) when compared to the more strongly ventrally curved ischia in *Zuniceratops* and in the centrosaurines *Medusaceratops*, *Wendiceratops*, *Furcatoceratops* (*Ishikawa, Tsuihiji & Manabe, 2023*), *Centrosaurus*, *Styracosaurus*, the Iddesleigh pachyrhinosaur, and *Pachyrhinosaurus lakustai*. Both *Lokiceratops* and *Wendiceratops* differ in the paddle-shaped distal end of the ischium, compared to the condition in *Medusacertopos lokii* (WDCB-MC-001; FDMJ-V-10), *Centrosaurus apertus* (YPM 002015), and *Styracosaurus albertensis* (CMN 344) which have a pointed distal end that twists and shifts to an anteroposteriorially or dorsoventrally compressed tab. *Lokiceratops* also differs from the rounded, but not mediolaterally compressed distal ends in the Iddesleigh pachyrhinosaur (TMP 2002.76.1) and *Pachyrhinosaurus lakustai*. The distinct kink two-thirds of the way along the shaft on the ischia of *Lokiceratops* differs from the gently curved overall shape of the ischia in *Zuniceratops* (MSM P2107) and all known centrosaurines, including *Medusaceratops*,

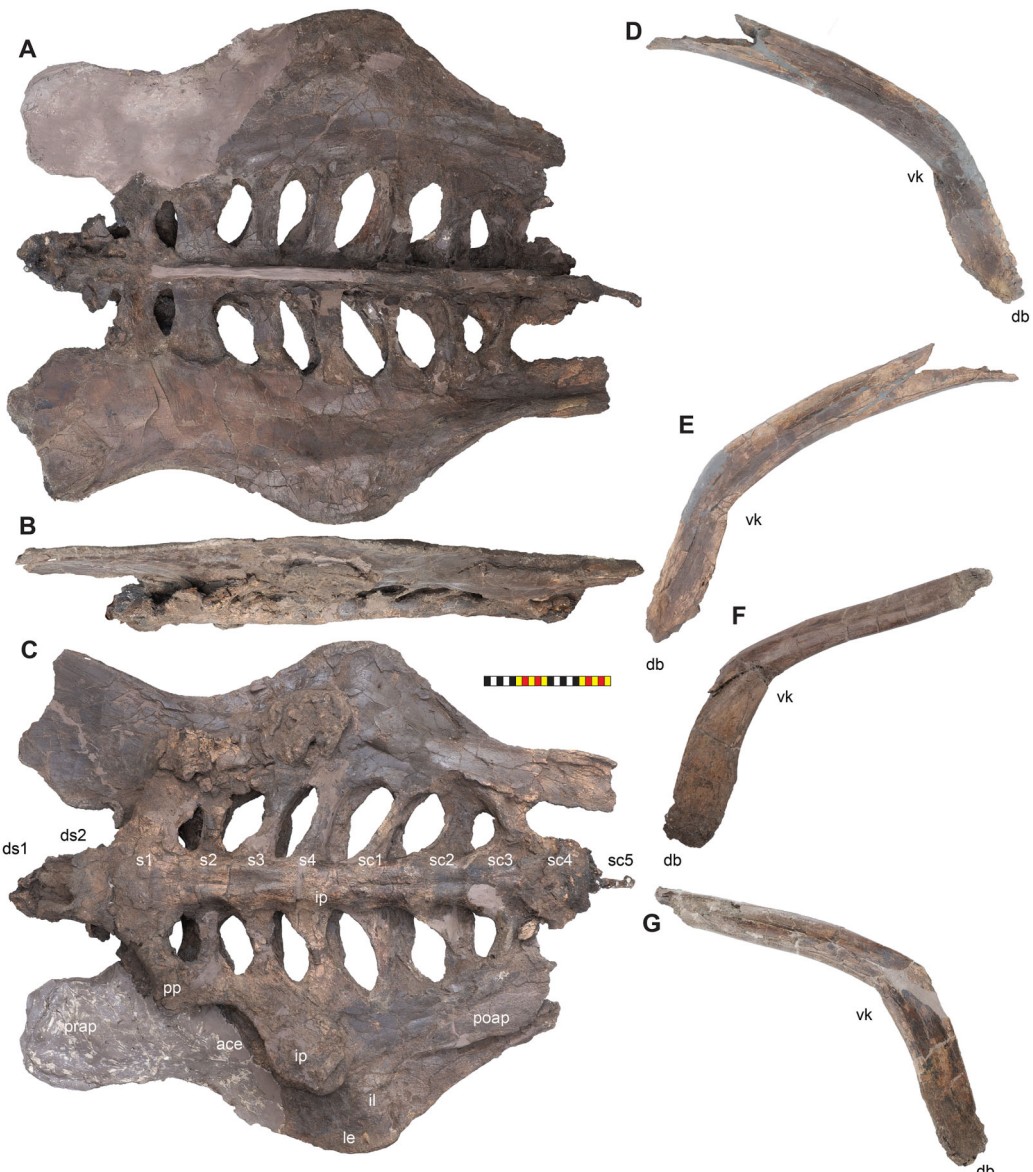

**Figure 23 Pelvic elements of *Lokiceratops rangiformis* n. gen et n. sp. (EMK 0012).** Photographs of posterior dorsal (dorsosacral), sacral, and anterior caudal (sacrocaudal) vertebrae fused into the synsacrum and ilia of EMK 0012 in (A) dorsal, (B) left lateral, and (C) ventral views. Entire pelvic unit is dorsoventrally crushed post-depostionally. Left ischium in left lateral (D) and medial (E) views. Right ischium in right lateral (F) and medial (G) views. *Osteological abbreviations*: ace, acetabulum; db, distal blade; ds1–2, dorsosacral vertebrae; ip, ischial pedulcle; le, lateral expansion; poap, postacetabular process; pp, pubic peduncle; prap, preacetabular process; s1–s4, true sacral vertebrae; ventral kink. Photos by Marcus Donivan. Scale bar equals 20 cm.

*Wendiceratops*, *Centrosaurus*, *Styracosaurus*, the Iddesleigh pachyrhinosaur, and *Pachyrhinosaurus lakustai*.

## Ontogenetic assessment

Aside from the fact that *Lokiceratops* (EMK 0012), at almost two meters in cranial length, is one of the absolutely largest centrosaurine specimens in existence, many features of the

skull also confirm its maturity. The rostral is completely fused to the premaxillae, whereas in juvenile specimens and many subadult ceratopsian specimens, these elements are unfused or connected through a visible suture (*Goodwin & Horner, 2008*; *Kirkland & DeBlieux, 2010*). The sutures between the nasal, prefrontal, and lacrimal are completely obliterated, indicating skeletal maturity. These elements are unfused in juvenile and subadult and specimens of both chasmosaurines and centrosaurines (*Sampson, Ryan & Tanke, 1997*; *Goodwin et al., 2006*; *Goodwin & Horner, 2008*). Similarly, the sutures between the prefrontal, frontal, postorbital, squamosal, and parietal are completely obliterated and covered with deep rugosities along the dorsum of the skull around the dorsocranial complex. Interestingly, while both palpebrals are fused to the lacrimals, the prefrontals, and the postorbitals, they preserve the sutures between each of these elements similar to the condition in juvenile centrosaurines (*Dodson, 1990*) and a juvenile chasmosaurine from the Dinosaur Park Formation (*Currie et al., 2016*).

The nasal exhibits the same surface texture as the rest of the facial skeleton, lacking any evidence of ornamentation or ontogenetic resorption of ornamentation. Although the nasal ornamentation in centrosaurines is variable in samples for which we have multiple individuals and different size classes, no centrosaurine taxon has been demonstrated to lose nasal ornamentation during ontogeny.

Fusion of frill margin epiossifications also suggests maturity in *Lokiceratops* (EMK0012). Epiossification fusion and development can also be used to assess relative maturity (*Sampson, Ryan & Tanke, 1997*). Both epijugals are securely fused onto the jugals compared to the unfused situation in juvenile and many subadult ceratopsid specimens (*Goodwin & Horner, 2008*; *Currie, Langston & Tanke, 2008*; *Kirkland & DeBlieux, 2010*). Epiossifications fuse from the posterior margin of the parietosquamosal frill anteriorly in centrosaurines (*Sampson, Ryan & Tanke, 1997*). Episquamosal fusion occurs only in the most mature individuals (*Frederickson & Tumarkin-Deratzian, 2014*). The episquamosals are completely fused to the squamosals in EMK0012 and are only readily discernable from the blade of the squamosal posteriorly. All of the epiparietals are securely fused to the parietal, with little indication of these being separate ossifications as seen in immature subadult specimens of more derived centrosaurines (*e.g.*, *Centrosaurus*, *Styracosaurus*, *Einiosaurus* (*Sampson, Ryan & Tanke, 1997*; *Wilson & Scannella, 2021*)) and in chasmosaurines (*Goodwin & Horner, 2008*; *Mallon, Ryan & Campbell, 2015*; *Currie et al., 2016*). The epiossifications on the parietals of EMK0012 (Fig. 16) exhibit subtle imbrication between neighboring elements along their lateral margins characteristic of adult centrosaurines that have this feature (*Sampson, Ryan & Tanke, 1997*).

The degree of epiparietal development is variable on specimens with the mottled texture that is typically associated with sub-adult individuals (*Brown, Russell & Ryan, 2009*). More mature specimens exhibit a more pronounced epiparietal ornamentation (*Sampson, Ryan & Tanke, 1997*). The hypertrophied epiparietal ornamentation of *Lokiceratops* is also indicative of skeletal maturity. *Lokiceratops* possesses the most massive, blade-like epiparietals of any ceratopsian. These hypertrophied epiparietal ornaments resemble those of the basal centrosaurines *Medusaceratops* and *Albertacertatops* (considered to be mature), and the pattern seen in adult specimens of more derived centrosaurines (*e.g.*,

*Centrosaurus, Coronosaurus, Styracosaurus, Einiosaurus* (*Sampson, Ryan & Tanke, 1997*; *Frederickson & Tumarkin-Deratzian, 2014*; *Wilson & Scannella, 2021*)).

The available postcranial elements of *Lokiceratops* (EMK0012) also indicate a high degree of skeletal maturity. The two preserved non-synsacral vertebrae of EMK0012 exhibit complete fusion between the centrum and the neural arch that is associated with skeletal maturity (*Brochu, 1996*; *Irmis, 2007*). The scapula and coracoid of EMK0012 are firmly fused together, compared with some juvenile ceratopsians (*Słowiak, Tereshchenko & Fostowicz-Frelik, 2019*).

The surface textures of cranial elements *Lokiceratops* (EMK0012) also exhibit evidence of maturity. Almost without exception, smaller elements of dinosaurs exhibit the characteristic striated periosteal surface bone texture typical of many other juvenile archosaurs (*Bennett, 1993*; *Sampson, Ryan & Tanke, 1997*; *Carr, 1999*). Juvenile bone is characterized by extremely thin, parallel ridges and grooves that are generally aligned parallel to the long axis of the bone or in the direction of greatest growth. These striations are presumably attributable to rapid bone growth in juvenile animals (*Sampson, Ryan & Tanke, 1997*). This pattern is also present on craniofacial elements of theropods such as maxillae, jugals, quadratojugals, and dentaries; forelimb elements such as scapulae, coracoids, humeri, radii, and ulnae; and hind limb elements such as the pubes, ischia, femora, tibiae, fibulae, and metatarsals (*Carr, 1999*; *Claessens & Loewen, 2016*; *Cunningham, Loewen & Sertich, 2019*). Larger elements possess a variety of textures, from rugose to mottled to smooth, all markedly distinct from the striated pattern of juvenile bone. Elements that fall between these sizes exhibit a mosaic between juvenile and adult patterns. These data are consistent with those noted independently in pterosaurs (*Bennett, 1993*) and in ceratopsians (*Sampson, Ryan & Tanke, 1997*). Bone surface texture varies ontogenetically on the parietosquamosal frill of numerous centrosaurines in which there are a range of sizes and has been established as an indicator of the relative maturity in the group (*Sampson, Ryan & Tanke, 1997*; *Brown, Russell & Ryan, 2009*). This is most pronounced on the dorsal surface of the parietal posterior to the parietal fenestrae. Juvenile specimens exhibit a distinctive striated bone surface texture, whereas the largest individuals with fully developed epiparietal ornamentation completely lack this texture, exhibiting a non-porous woven bone surface texture associated with deep rugosities and well-defined vascular channels on the parietal. The parietals of subadults exhibit a "mottled" transitional texture characterized by fine pitting on the bone surface and a lack of the striations that characterize long-grained texture (*Brown, Russell & Ryan, 2009*). EMK 0012 completely lacks any trace of striated or mottled surface texture, and exhibits only adult surface texture, suggesting that the specimen represents a somatically mature individual. This independent method of assessing relative age of specimens is consistent with the fusion of cranial elements (*Sampson, Ryan & Tanke, 1997*) and the degree of closure of the neurocentral sutures (*Brochu, 1996*; *Irmis, 2007*). Together, these features indicate that the holotype of *Lokiceratops rangiformis* represents an adult individual with fully developed ornamentation and can therefore be confidently diagnosed and placed in a phylogenetic analysis.

**Table 2 The temporal distribution of Centrosaurinae.** Formations and possible temporal time windows for centrosaurine taxa. Formations are color coded for biogeographic area or landmass (LM) with green representing southern Laramidia (SL), blue representing northern Laramida (NL), and red representing Asia (A). Possible taxon time spans do not represent absolute temporal position or absolute temporal ranges as many taxa are represented by a single specimen and many have not been precisely dated. The ranges of the taxa are based on the temporal time window in which they could have existed, where possible to Ur/Pb radiometric dates bracketing the specimen (see sources, also further detail in Supplemental Table S1).

| Taxon | Formation | LM | Age range | Sources |
|---|---|---|---|---|
| *Diabloceratops eatoni* | Wahweap Formation | SL | 81.42–81.01 Ma (81.27 + 0.36–0.5) | 36 |
| *Machairoceratops cronusi* | Wahweap Formation | SL | 80.68–79.26 Ma (80.06 + 0.62–0.8) | 36 |
| *Menefeeceratops sealeyi* | Menefee Formation | SL | 81–78.5 Ma | 37,38,39 |
| *Yehuecauhceratops mudei* | Aguja Formation | SL | 80.0–76.28 ± 0.06 Ma | 40,41,42 |
| *Xenoceratops foremostensis* | Foremost Formation | NL | 79.5–78.6 Ma | 43,4,44 |
| *Crittendenceratops krzyzanowskii* | Fort Crittenden Formation | SL | 83–77 Ma | 47,48,49 |
| *Nasutoceratops titusi* | Kaiparowits Formation | SL | 76.394 ± 0.040–75.609 ± 0.015 Ma | 50,4 |
| *Avaceratops lammersi* type | Judith River Formation | NL | 78.594 +− 0.024–76.24 +− 0.18 | 51,46,4 |
| *Avaceratops* sp. MOR 692 | Judith River Formation | NL | 75.8–75.4 Ma | 46,41 |
| *Avaceratops* sp. CMN 8804 | Oldman Formation | NL | 76.470 + 0.14/− 0.084–75.8 Ma | 52,41,4 |
| *Lokiceratops rangiformis* | Judith River Formation | NL | 78.08 + 0.3–0.9 Ma | 4 |
| *Albertaceratops nesmoi* | Oldman Formation | NL | 78.28 + 0.2–0.9 Ma | 4 |
| *Medusaceratops lokii* | Judith River Formation | NL | 78.18 + 0.2–0.9 Ma | 4 |
| *Wendiceratops pinhornensis* | Oldman Formation | NL | 78.08 + 0.3–0.9 Ma | 53,4 |
| *Sinoceratops zhuchengensis* | Hongtuya Formation | A | 77.3 ± 1.54–73.5 ± 0.3 Ma | 31 |
| *Coronosaurus brinkmani* | Oldman Formation | NL | 77.5–76.354 ± 0.057 Ma | 54,50,55,4 |
| *Spinops sternbergorum* | Oldman Formation | NL | 76.717 ± 0.020–76.354 ± 0.057 Ma | 56,4 |
| *Centrosaurus apertus* | Dinosaur Park Formation | NL | 76.470 + 0.14/− 0.084–75.8 Ma | 3,4 |
| *Styracosaurus albertensis* | Dinosaur Park Formation | NL | 75.8–75.3 Ma | 3,57,4 |
| *Styracosaurus ovatus* | Two Medicine Formation | NL | 78.2–75.259 ± 0.027 Ma | 58,59,4 |
| *Stellasaurus ancellae* | Two Medicine Formation | NL | 78.2–75.259 ± 0.027 Ma | 58,59,4 |
| *Einiosaurus procurvicornis* | Two Medicine Formation | NL | 75.259 ± 0.027 Ma | 53,51,4 |
| *Iddesleigh pachyrhinosaur* | Dinosaur Park Formation | NL | 75.25–75.22 Ma | 61,57,4 |
| *Achelousaurus horneri* | Two Medicine Formation | NL | 75.259 ± 0.027–74.6 Ma | 53,51,4 |
| *Pachyrhinosaurus lakustai* | Wapiti Formation | NL | 73.73 ± 0.25–71.89 ± 0.14 Ma | 62,29 |
| *Pachyrhinosaurus perotorum* | Prince Creek Formation | NL | 73.4–70.0 ± 0.2 Ma | 63,64,55 |
| *Pachyrhinosaurus canadensis* | Horseshoe Canyon Formation | NL | 72.7–71.8 Ma | 29 |
| CPC 279 | Cerro Del Pueblo Fm. | SL | 73.63–72.74 Ma | 70,71,55 |

## Phylogenetic analysis

In order to formulate hypotheses of the phylogenetic relationships of *Lokiceratops rangiformis* relative to other centrosaurine ceratopsid dinosaurs, a phylogenetic analysis using cladistic maximum parsimony was employed. The analysis comprised 86 taxa and 377 characters (263 cranial, 61 postcranial, and 53 concerning frill-based ornamentation). Characters 354–377 are scored across Chasmosaurinae and Centrosaurinae and assume modified epiparietal homologies first proposed by *Clayton et al. (2009, 2010)*. Age ranges for each taxon were taken from the literature, and where possible, were updated to the

latest calibrations (Table 2). For the sources of age control for taxa see Supplemental Table SD2. See Supplemental Data File SD3 for characters used in the phylogenetic analysis, and Supplemental Data Files SD4 and SD5 for taxon scorings.

Numerous features indicate that *Lokiceratops rangiformis* was a member of the clade Centrosaurinae; therefore, selection of ingroup taxa and characters focused on this clade. Most described ceratopsid species, and several currently unnamed taxa, were included in the analysis, for a total of 27 centrosaurines, and 19 chasmosaurines.

To ensure proper character polarization and to determine the position of Ceratopsidae within Ceratopsia, we sampled widely across both psittacosaur and non-ceratopsid neoceratopsians. The neornithischian *Hypsilophodon foxi* was constrained as the outgroup because it is a proximate sister group within Ornithopoda and is known from nearly complete remains. Three pachycephalosaurians were included in the analysis to polarize some characters. Characters used in the analysis (SD3) were partially derived from the previous, ever-evolving data matrix initiated by Scott Sampson and Cathy Forster in the 1990's and further fleshed out by Mark Loewen and Andrew Farke during the 2000's and 2010's (*Loewen et al., 2010*; *Sampson et al., 2010*; *Farke et al., 2011*; *Knapp et al., 2018*). In addition to adding several new characters and character states, close attention was paid to existing character definitions; a significant number of characters from previous analyses were deleted, combined, or otherwise revised with an eye toward improving clarity and anatomical precision.

The original character-taxon matrix was assembled in Microsoft Excel (Office Professional Plus, 2019; Microsoft, Redmond, WA, USA; Supplemental Data File SD4) imported into Mesquite v. 3.70 (*Maddison & Maddison, 2011*) and is freely available as a NEXUS text file in Supplemental Data File SD5. The final dataset was analyzed using TNT v. 1.6 (*Goloboff, Farris & Nixon, 2008*; *Goloboff & Catalano, 2016*). Tree searching followed the parsimony criterion implemented under the heuristic search option using tree bisection and reconnection (TBR) with 10,000 random addition sequence replicates, up to 10,000 trees saved per replication, and zero length branches collapsed if they lacked support under any of the most parsimonious reconstructions. All characters were equally weighted. Characters 1, 51, 70, 126, 130, 144, 170, 261, 262, 279, 336, and 339 represent nested sets of homologies and/or entail presence and absence information. These characters were set as additive (also marked as ORDERED in highlighted bold text following character description (see Supplemental Data File SD3)). A total of 288 most-parsimonious trees were found; we report the strict consensus of these trees here (Fig. 24; see also Supplemental Figures S1–S5 for relationships of all Ceratopsia). Tree statistics were calculated using TNT. Bootstrap proportions were calculated using 10,000 bootstrap replicates with 10 random addition sequence replicates for each bootstrap replicate.

With the inclusion of all 86 taxa, the analysis recovered 288 most parsimonious trees with a length of 928 steps, consistency index (CI) of 0.474, and retention index (RI) of 0.898 (Fig. 24; Supplemental Figures S1, S2, S5). With the exclusion of the fragmentary taxon *Crittendenceratops krzyzanowskii*, the analysis recovered 180 most parsimonious trees with a length of 921 steps, CI = 0.471, and RI = 0.896. With the exclusion of the

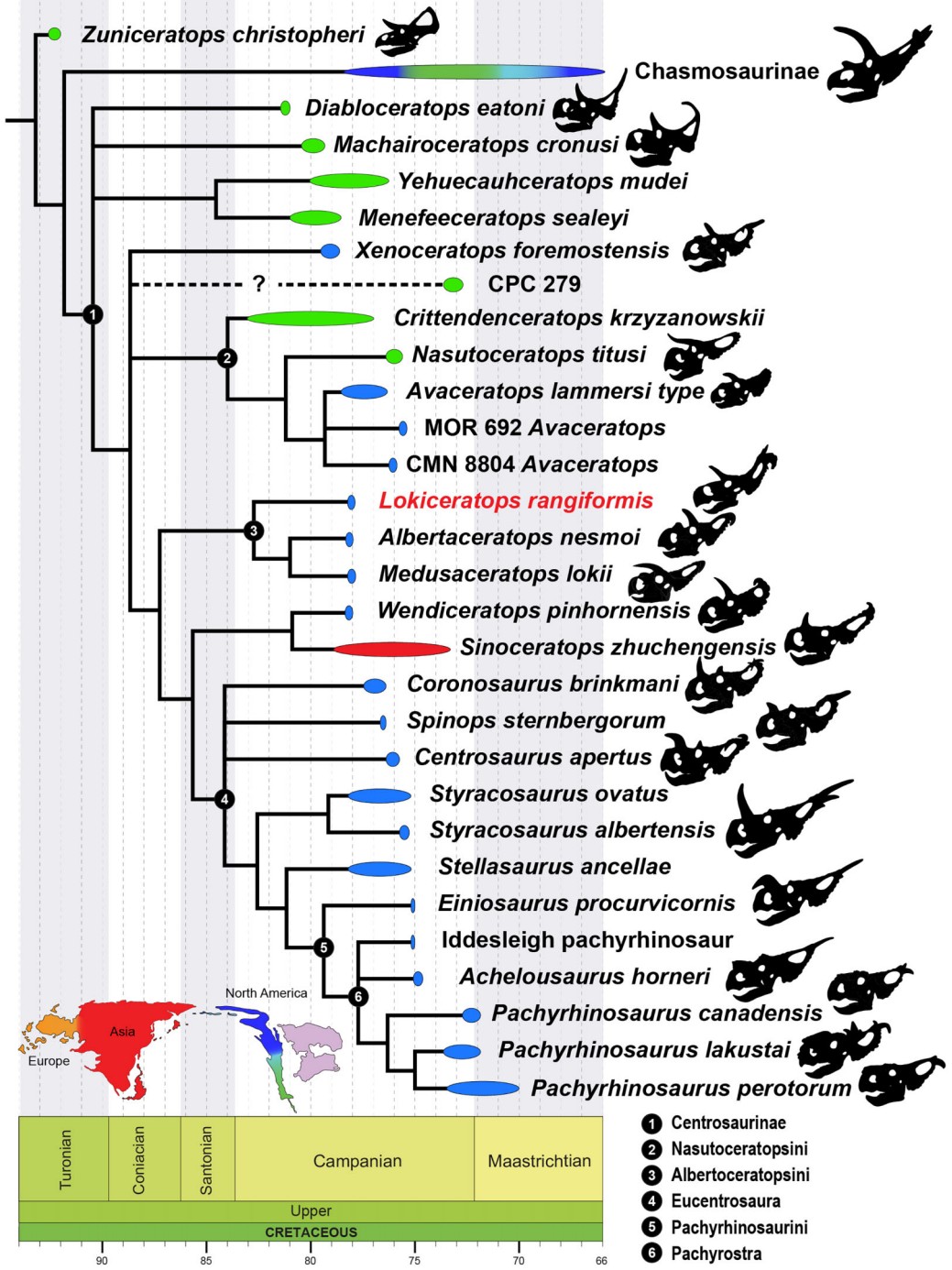

**Figure 24 Phylogenetic relationships of *Lokiceratops rangiformis* n. gen et n. sp. (EMK 0012) within Centrosaurinae.** 50% majority-rules tree of 288 most parsimonious trees (tree length = 928; CI = 0.474; RI = 0.896) recovered in the phylogenetic parsimony analysis using TNT (*Goloboff, Farris & Nixon, 2003*). The only nodes scoring less than 100 are indicated. See text for details. Temporal time windows do not always represent absolute temporal ranges as many taxa are represented by a single specimen and many are poorly dated. The ranges of the taxa are based on the temporal time window in which they could have existed, where possible to Ur/Pb radiometric dates bracketing the specimen (see sources, also further detail in Supplemental Table S1).

fragmentary taxa *Menefeeceratops sealeyi*, *Yehuecauhceratops mudei*, and *Crittendenceratops krzyzanowskii* the analysis recovered 60 most parsimonious trees with a length of 920 steps, CI = 0.475, and RI = 0.898. When analyzed with only the 27 centrosaurines and setting *Hypsilophodon foxi* as the outgroup taxon, the analysis recovered 16 most parsimonious trees with a length of 364 steps, CI = 1.201, and RI = 1.015. When analyzed with only the 27 centrosaurines and setting *Protoceratops andrewsi* as the outgroup taxon the analysis recovered 64 most parsimonious trees with a length of 315 steps, CI = 1.387, and RI = 1.026. The strict consensus tree for all of these analyses is the same with respect to centrosaurine ingroup relationships (Fig. 24; Supplemental Figures S3–S5).

*Lokiceratops rangiformis* is found to be a centrosaurine ceratopsid in all analyses based, in part, on the presence of a round external narial fossa on the premaxilla, a fan-shaped squamosal, large flattened, bladelike epiparietals, as well as the high number (>6) of epiparietals on each side (Characters 57, 64, 66 see SD3). *Lokiceratops rangiformis* is recovered as the sister taxon to *Albertaceratops nesmoi* and *Medusaceratops lokii*, and all three form the clade Albertaceratopsini. This grouping is supported by two synapomorphies: character 127, a circular or oval rather than narrow and slit-like frontoparietal fontanelle; and character 357, epiparietal 1 oriented in the plane of the frill in lateral view (note that this is a local synapomorphy, found in other clades within Centrosaurinae); The decay index (1) and bootstrap support (<50%) for the relationships within Albertaceratopsini are low, although we note that low branch support is unsurprising—in this analysis: the confidently established clade Centrosaurinae itself has a decay index of only 2 and bootstrap support <50 percent.

This phylogeny is broadly consistent with recent phylogenies of Centrosaurinae in several aspects, such as the content of Eucentrosaura (*sensu* Chiba et al., 2017; seen also in Wilson, Ryan & Evans, 2020), a monophyletic Nasutoceratopsini (*sensu* Ryan et al., 2017; Chiba et al., 2017; Wilson, Ryan & Evans, 2020), and the positions of *Diabloceratops* and *Machairoceratops* as sister to most/all other Centrosaurinae (*e.g.*, Chiba et al., 2017; Wilson, Ryan & Evans, 2020; but note variabilities in the latter analysis for parsimony *vs* Bayesian methods). Variation in the positions of taxa such as *Xenoceratops*, Nasutoceratopsini, *Medusaceratops*, *Albertaceratops*, *Sinoceratops*, and *Wendiceratops* across various studies reflect missing data and likely high levels of homoplasy.

# DISCUSSION

## Kennedy coulee ceratopsid richness

The Milk River border region of southern-most Alberta and northern-most Montana, centered on Kennedy Coulee, preserves a distinct dinosaur assemblage in the lower Judith River Formation and lowest Oldman (Unit 1) and uppermost Foremost formations, here informally termed the 'Kennedy Coulee Assemblage' (KCA). The narrow stratigraphic interval of the KCA (Fig. 1) preserves five distinct ceratopsid taxa consisting of four centrosaurines (*Albertaceratops nesmoi*, *Lokiceratops rangiformis*, *Medusaceratops lokii*, and *Wendiceratops pinhornensis*) and one chasmosaurine, *Judiceratops tigris*. Given the rapid turnover of megaherbivorous assemblages documented in other deposits of northern

Laramadia (*e.g.*, *Mallon et al., 2012*; *Mallon, 2019*), this stratigraphic interval, dated to approximately 78 Ma, is currently one of the only windows into this temporal interval in northern Laramidia, possibly synchronous or parasynchronous with some of the fossils from the Two Medicine Formation of Montana (*Varricchio et al., 2010*). In southern Laramidia, synchronous or parasynchronous intervals useful for latitudinal comparisons of assemblages include the Upper Shale Member of the Aguja Formation of West Texas (*Lehman, Wick & Barnes, 2017*) and the upper Pardner Canyon Member of the Wahweap Formation of southern Utah (*Beveridge et al., 2022*).

Given the significance of this interval, the emerging picture of high ceratopsid diversity in the KCA requires scrutiny of each of the established ceratopsian taxa. Of the four established and possibly sympatric centrosaurine taxa from the KCA, *Wendiceratops* and *Medusaceratops* have distinct blade-like nasal horns, with the nasal horn in *Wendiceratops* being exceptionally long amongst taxa possessing blade-like nasal horns. The sole specimen of *Albertaceratops* also has an elongate, blade-like nasal horn in contrast to the absence of nasal ornamentation in *Lokiceratops*. *Wendiceratops* and *Albertaceratops* are also morphologically distinct from *Lokiceratops* in the shape, orientation, and number of frill ornaments. However, superficial similarities between *Lokiceratops* and *Medusaceratops*, including the pair of broad and greatly enlarged second epiparietal ossifications (ep2) and anterolaterally directed postorbital horncores, necessitate clear differentiation between these two taxa.

One compelling difference is overall body size, with *Lokiceratops* at least 20% larger than known adults of *Medusaceratops*. *Chiba et al. (2017)* convincingly demonstrated that the largest individuals of *Medusaceratops* from the Mansfield bonebed had achieved their adult body size, with the adult skull similar in length to other large centrosaurines like *Centrosaurus apertus* and *Styracosaurus albertensis* (~660–790 mm). The skull of *Lokiceratops* is significantly larger, with a basal skull length of 948 mm. While large differences in adult size have been noted in chasmosaurine ceratopsians (*e.g.*, *Triceratops*, *Torosaurus*), centrosaurine taxa for which many adult individuals are known (*e.g.*, *Centrosaurus*, *Coronasaurus*, *Pachyrhinosaurus lakustai*, *P. canadensis*, *Styracosaurus*) have not been shown to vary significantly in adult skull size (*Currie, Langston & Tanke, 2008*; *Frederickson & Tumarkin-Deratzian, 2014*).

While superficially similar, cranial ornamentation is also significantly different between *Medusaceratops* and *Lokiceratops*, with differences falling well outside ontogenetic and intraspecific variation documented in other centrosaurs (*e.g.*, *Brown et al., 2020*; *Holmes et al., 2020*). The presence of a small first epiparietal (ep1) in *Medusaceratops* is clearly established in three large, mature individuals from the Mansfield bonebed locality, including the holotype WDC-DJR-001 and referred specimens WDC-DJR-002 and ROM 73837 (*Chiba et al., 2017*). In contrast, the first epiparietal ossifications of *Lokiceratops* are prominent and asymmetric, with the enlarged left ep1 second in size only to the extremely large ep2 ossifications. These prominent second epiparietals in *Lokiceratops* significantly exceed the maximum size of ep2 in *Medusaceratops*, extending laterally well beyond the margins of the parietosquamosal frill, whereas the enlarged blade-like ep2 of *Medusaceratops* is less than half the relative size of ep2 in *Lokiceratops* (Fig. 25) and does

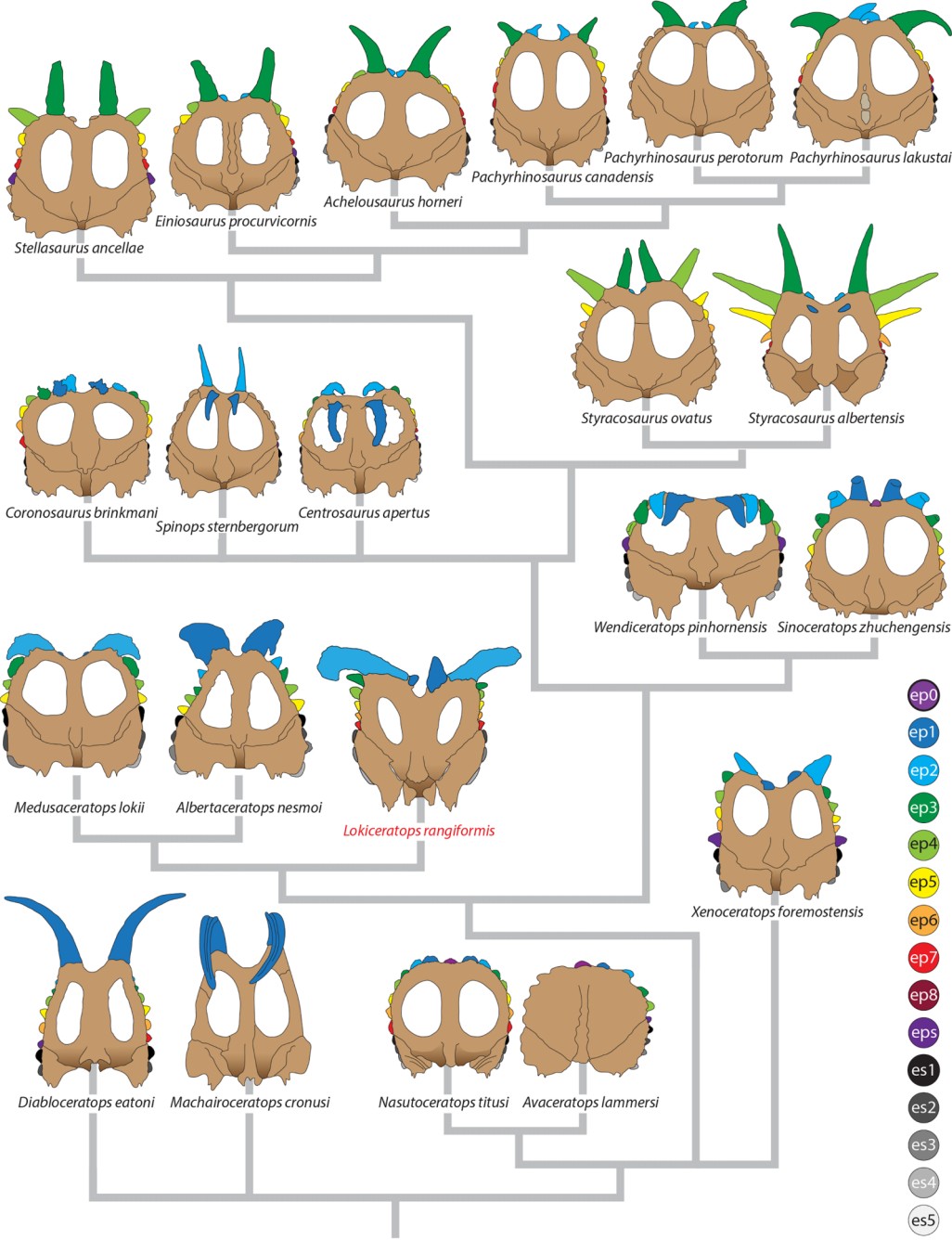

**Figure 25 Comparisons of the known frills of centrosaurines.** Frills of centrosaurine dinosaurs.

not extend laterally beyond the margins of the frill. Proximally, *Medusaceratops* possesses three lateral epiparietals (ep3–ep5), while *Lokiceratops* preserves a minimum of four and up to five lateral epiparietals on the right (ep3–ep6 or ep7) and a total of five on the left (ep3–ep7). This indicates a conservative difference of at least one additional epiparietal position in *Lokiceratops*, and possibly two additional positions. Finally, *Medusaceratops*

exhibits a distinct, elongated blade-like nasal horn (WDC-DJR-012) while *Lokiceratops* shows no definitive evidence for the presence of a nasal horn.

Combined, notable differences in adult body size, multiple significant differences in cranial ornamentation (number of epiossifications, morphology of epiossifications, and the lack of ornamentation on the nasal), are beyond intraspecific variation observed in other centrosaurine taxa, clearly differentiating *Lokiceratops* from other established centrosaurs from the KCA.

Detailed stratigraphic distributions of taxa within the KCA (Fig. 1) show that *Lokiceratops* closely overlaps the holotype of *Wendiceratops* and the chasmosaurine *Judiceratops tigris* (*Longrich, 2013*) at the leve of stratigraphic resolution that is achievable. Erected on the basis of four fragmentary cranial specimens (YPM VPPU 022404, 02341, 023261, and 023262) collected from Kennedy Coulee (*Longrich, 2013*), *Judiceratops* has represented one of the earliest members of Chasmosaurinae for more than a decade. Most of the authors consider the holotype of *Judiceratops tigris*, YPM VPPU 022404 to be nondiagnostic, although co-author Nick Longrich notes that: "two independent studies (*Longrich, 2013*, *Campbell, 2015*) carefully examined the type and referred material and, while differing on details, concur that: (i) the material represents a distinct species based on a unique combination of characters of the frill and autapomorphic postorbital horn morphology; and (ii) exhibits derived features of Chasmosaurinae, a conclusion corroborated by phylogenetic analysis (*Longrich, 2013*). Both of these articles (*Longrich, 2013*; *Campbell, 2015*) were published prior to the description of *Wendiceratops* (*Evans & Ryan, 2015*), and the type of *Judiceratops* should be subject to further, detailed investigation. The referred chasmosaurine squamosal, YPM VPPU 023262, represents unequivocal evidence of a fifth distinct ceratopsid in the 'Kennedy Coulee Assemblage.' This level of ceratopsid richness, with five distinct taxa within a narrow stratigraphic interval, is currently unknown elsewhere in Laramidia.

## High endemism of centrosaurines

Among ceratopsids, Centrosaurinae show an unusual pattern of high endemism, with multiple species occurring differentially along the coastal plain of the Western Interior Seaway. This pattern is striking because, in general, large animals today tend to have large geographic ranges (*Brown, Stevens & Kaufman, 1996*). Yet, based on current evidence and acknowledging uneven temporal and geographic sampling, all known centrosaurine species exhibit relatively small geographic ranges. This pattern is seen not only in genera and species, but also above the species level. That is, centrosaurine subclades—including Albertaceratopsini—also show restricted geographic distributions.

All southern centrosaurine taxa are represented either by a single specimen or limited material geographically restricted by the aerially exposed extent of their stratigraphic intervals and are currently undocumented from equivalent intervals elsewhere. For example, *Nasutoceratops titusi*, currently based on two specimens from the middle unit of the Kaiparowits Formation of southern Utah, may have been wide-ranging across southern Laramidia but fossiliferous non-marine intervals of the same age are not exposed elsewhere. The same is true in northern Montana, where taxa are known from limited

specimens occurring within limited geographic areas (*i.e.*, *Einiosaurus procurvicornis*, *Coronosaurus brinkmani*). However, in taxa with more extensive sampling, ranges appear relatively limited within Laramidia. *Centrosaurus*, by far one of the best represented centrosaurines (*Chiba et al., 2015*), has a geographic range spanning less than 200 km. *Pachyrhinosaurus canadensis* has a known range of 200 km (*Currie, Langston & Tanke, 2008*). All other northern taxa are known from single sites or single localities. *Styracosaurus albertensis*, for example, is known from multiple specimens but all within Dinosaur Provincial Park. Generously, the largest range of any centrosaur (*i.e.*, *Coronosaurus, Centrosaurus, Pachyrhinosaurus canadens*) is around 60,000 km$^2$.

Thus, the preserved ranges of species must underestimate their true ranges. In particular, there is a lack of sampling of centrosaurines to the west of the coastal plain in upland sediments, with the exception of *Crittendenceratops* in intermountain regions of Arizona (*Dalman et al., 2021*). It is possible that lineages ranged farther to the west, and that the geologically better represented coastal plain represents the eastern edge of many species' distributions. However, the restricted ranges seen along the coastal plain suggest endemism is not solely the result of sampling; even assuming they ranged westward across Laramidia, the restricted known latitudinal ranges argue for small true geographic ranges.

Furthermore, if small species ranges were simply a result of poor sampling, we would predict that known species ranges would have tended to increase over time as sampling has improved since initial exploration in the late 19th century. Instead, better sampling has tended to reveal new species, and only modest range extensions of known species. Trends in the data therefore corroborate the presence of small geographic ranges for centrosaurines (Fig. 26) and imply that the small geographic ranges documented in centrosaurines are a real biological phenomenon.

The extant hartebeest species complex is a possibly analogous group, with variable ranges that have responded to climate shifts in the last 500K years, mainly through diversification of small populations (likely due to habitat fragmentation). This clade, which can reliably be divided into several molecularly defined lineages (*Flagstad et al., 2000*), are also morphologically distinguishable by their horn morphology (*Capellini & Gosling, 2006*). While we do not have range sizes for these (IUCN red list lumps all the taxonomic units under the parent species), distribution maps suggest that some subspecies or diagnosable units occupy very limited areas.

The pattern of high endemism in centrosaurines is not only evident at the species level, but at the clade level as well, with centrosaurine clades also exhibiting highly restricted geographic ranges. The Albertaceratopsinae (*Lokiceratops*, *Albertaceratops*, *Medusaceratops*) is currently known only from northern Montana and southern Alberta across a geographic range of 25 km and ~490 km$^2$ of area. *Diabloceratops*-like animals with a single hypertrophied, elongated epiparietal ep1 have a known range distance of 30 km and a range area of ~700 km$^2$. Animals with a dorsal ridge just dorsal to the otic notch on the squamosal (*Menefeeceratops, Yehuecauhceratops, Crittendenceratops*) have a geographic range distance of 1,100 km and a range area of ~220,000 km$^2$. Nasutoceratopsini (*Nasutoceratops, Avaceratops*) have a geographic range distance of 2,000 km and a range

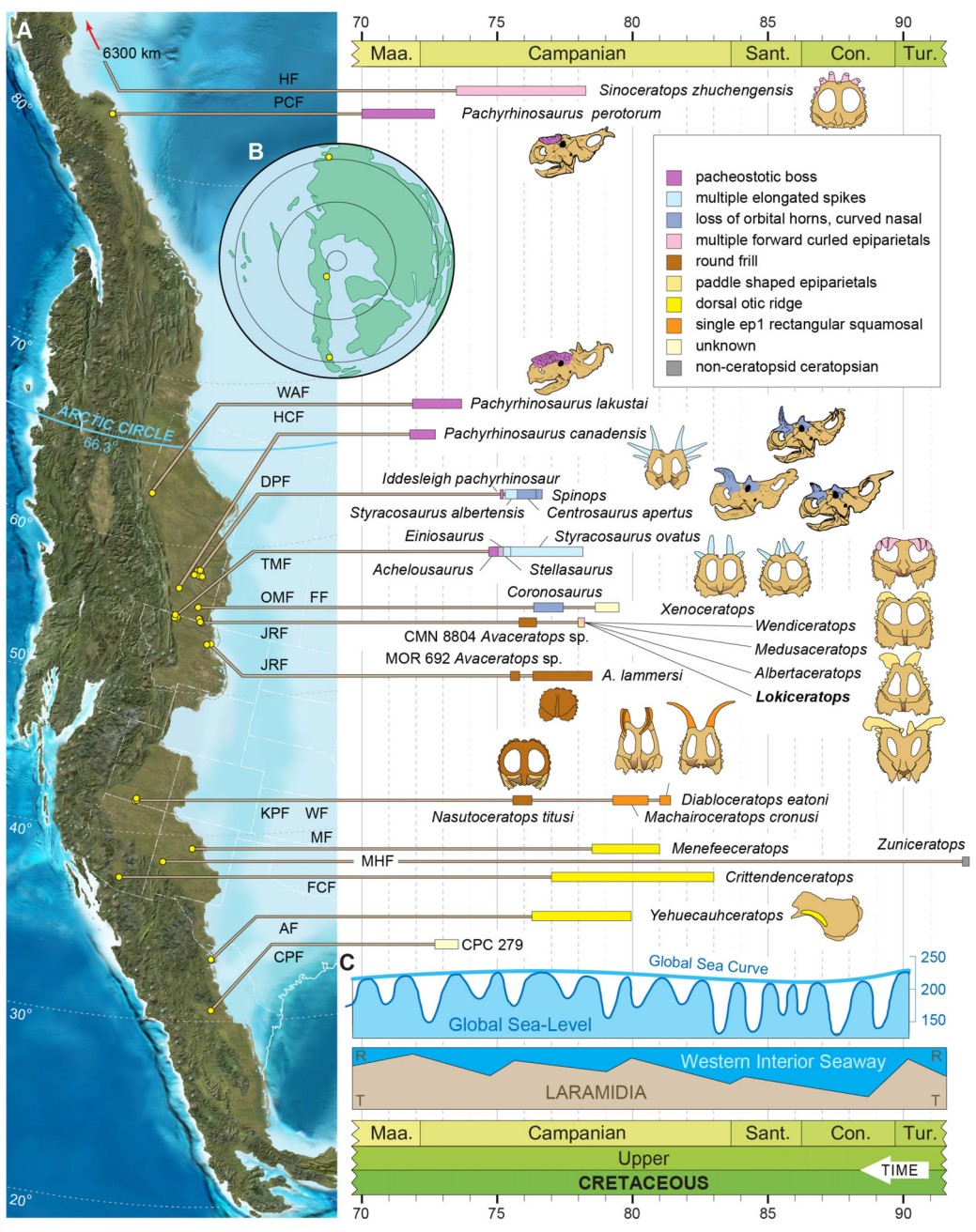

**Figure 26 Paleogeographic and stratigraphic distribution of centrosaurine dinosaurs.** (A) Paleo-geographic and stratigraphic distribution of centrosauruine dinosaurs. (B) Polar projection showing the location of *Sinoceratops* (top), *Pachyrhinosaurus perotorum* (middle), and CPC 279 (bottom) illustrating the wide-ranging distribution of centrosaurines in the northern hemisphere. (C). Global Sea-Levels from *Haq, Hardenbol & Vail (1987)* and Western Interior Seaway transgression and regression cycles from *Li & Aschoff (2022)*. Paleogeographic reconstruction modified after *Blakey & Ranney (2018)*.

area of ~200,000 km². Basal eucentrosaurans (*i.e.*, *Coronosaurus*, *Centrosaurus*, *Spinops*) have a range extension of ~200 km and a range area of ~10,000 km². *Styracosaurus* (*S. albertensis* + *S. ovatus*) ranges over 225 km and a range area of ~12,600 km².

Pachyrhinosaurini (*Einiosaurus, Achelousaurus*, the Iddesleigh pachyrhinosaur, and the three species of *Pachyrhinosaurus*) have the largest range at ~3,300 km and a range area of ~660,000 km$^2$ generously assuming the coastal plain averaged 200 km wide from the uplands to the coast and was habitable along its full extent. These patterns require that not only were lineages isolated long enough to evolve into separate species; these lineages then underwent regional diversifications producing multiple species in the same area.

It has previously been proposed that dinosaurs in the Late Cretaceous of North America showed relatively high levels of endemism (*Lehman, 1997*; *Lehman, 2001*), a pattern now observed in centrosaurines (*Sampson et al., 2013*), chasmosaurines (*Sampson et al., 2010*; *Longrich, 2011*), hadrosaurids (*Gates & Sampson, 2007*), pachycephalosaurids (*Williamson & Carr, 2002*; *Longrich, Sankey & Tanke, 2010*), and tyrannosaurids (*Loewen et al., 2013*). Whether the assemblages can be divided into discrete biogeographic provinces, or complicated, overlapping patterns remains debated, but decades of fieldwork and improved stratigraphic constraint (*Ryan & Evans, 2005*; *Beveridge et al., 2022*; *Ramezani et al., 2022*; *Rogers, Eberth & Ramezani, 2023*) have shown that distinct species inhabited different and relatively circumscribed regions of the coastal plain. Yet while many dinosaurs show endemism, especially at the species level, few clades are known to display endemism to the same degree as documented in centrosaurines.

The hadrosaur *Parasaurolophus*, for example, is represented by different taxa in Alberta, Utah, and New Mexico (*Evans, Bavington & Campione, 2009*; *Gates, Evans & Sertich, 2021*), but the genus itself has a large geographic range. Other genera with large geographic ranges include the saurolophine *Gryposaurus* (*Gates & Sampson, 2007*), the pachycephalosaurid *Stegoceras* (*Sullivan & Lucas, 2006*), and the chasmosaurines *Triceratops* and *Torosaurus* (*Longrich & Field, 2012*). However, this does not exclude the presence of other endemic clades. The chasmosaur *Chasmosaurus*, for example, is known exclusively from many specimens all from a small region in southern Canada (*Godfrey & Holmes, 1995*). Nevertheless, centrosaurines are unusual in that, thus far, none of the subclades are known to be widely distributed.

The discovery of *Lokiceratops rangiformis* also suggests that, in addition to showing high endemism, the local diversity of centrosaurines was high in the Judith River region. Four distinct and coeval centrosaurine ceratopsians (*Lokiceratops rangiformis*, *Albertaceratops nesmoi*, *Medusaceratops lokii*, and *Wendiceratops pinhornensis*) occur within a small geographic area where the Milk River crosses the United States of America/Canadian border. These centrosaurines occur within a tight stratigraphic interval of four meters and range in age from a maximum of 78.28 to 78.1 Ma (Bayesian model median age). This pattern is distinct from other formations, where typically only a single species existed, and is analogous to the pattern of diversity observed in sympatric lambeosaurine hadrosaurids in the Dinosaur Park Formation (*Mallon & Anderson, 2013*).

## Endemism and turnover

The high regional endemism documented in Laramidia is likely connected to a pattern of rapid evolution and turnover. All known centrosaurines also have brief temporal ranges (*Sampson & Loewen, 2010*), of 500 Ka or less. *Lokiceratops, Albertaceratops,*

*Medusaceratops*, and *Wendiceratops* all appear and disappear in a ~200 ka window. *Styracosaurus ovatus*, *Stellasaurus*, *Einiosaurus*, and *Achelousaurus* all occur in a ~500 ka window and are not known to overlap, suggesting an average temporal range of ~125 ka. These are almost certainly underestimates of their true temporal ranges, given the Signor-Lipps Effect (*Signor & Lipps, 1982*; because the first occurrence and last observed/ sampled occurrence of a fossil do not correspond to the true first and last occurrence of the species).

Moreover, the appearance of *Styracosaurus* in the Dinosaur Park Formation (*Ryan & Evans, 2005*), without any recognized ancestor lower in section, may mean that its appearance marks the immigration of this species into the area, rather than *in-situ* evolution. Some of the perceived rapid turnover observed in the record may, therefore, result from change in geographic ranges rather than speciation and extinction events. Still, as with geographic range, improved sampling over the course of the past century has largely tended to reveal new species higher and lower in section rather than extending ranges of known species, suggesting that this rapid turnover reflects an actual evolutionary trend.

Rapid evolution may have been a key driver of high endemism. The appearance of distinct lineages in different regions of the continent suggests rapid evolution of lineages following dispersal, with adaptation to local environments, sexual selection, or both, driving lineages in different directions in terms of adaptations and ornament, causing them to evolve into distinct species. However, endemism itself can also be a driver of rapid evolution. That is, if conspecific populations became isolated from one another, evolution would accelerate due to genetic drift, with mutations rapidly becoming fixed in small populations. Endemism may also increase extinction rates. Small, endemic populations have fewer individuals, making them more vulnerable to extinction. Geographic range is tightly connected to extinction rates, with widespread species being more resistant to extinction, and species with small ranges more vulnerable (*Jablonski, 2008*). Since wide-ranging species inhabit many different environments and regions, an environmental change must affect all areas to eliminate the species, making their extinction more difficult.

## Drivers of dinosaurian endemism

During the Late Cretaceous, as increased volcanic seafloor spreading in the Pacific and Atlantic Ocean basins displaced water onto the continents (*Müller et al., 2022*), the Western Interior Seaway connected the Gulf of Mexico to the Arctic Ocean, between approximately 90 and 70 Ma, with the last remnants of the seaway persisting into the Paleocene (*Blakey & Ranney, 2018*). Laramidia, the western portion of the North American continent isolated by the creation of the Western Interior Seaway (*Hay et al., 1993*), hosted rich assemblages of dinosaurs on a vast coastal plain along its eastern margin, extending from northern Coahuila, Mexico in the south to central Alberta, Canada in the north, and forming a long-lived (>10 Ma) coastal lowland environment seemingly lacking persistent geographic barriers. Transgression events documented by marine tongues such as the Drumheller Marine Tongue and the Bearpaw transgression would have narrowed or expanded the coastal plain, but no persistent marine barriers are known capable of

preventing northern dinosaurs from dispersing south, or vice versa. *Fowler & Freedman-Fowler (2020)* did, however, suggest that intermittent flooding of the coastal plain may have occurred and interpreted that as a driver of ceratopsian diversification. Other factors may have been responsible for the observed patterns of dinosaur distribution and evolution. These factors potentially include climatic zones, floral composition and distribution, distributions of disease and parasites, competition with other dinosaurs, or a combination of these (*Linnert et al., 2014*; *Burgener et al., 2021*).

Ceratopsid dinosaurs evolved in isolation on Laramidia and are found predominantly along the eastern coast of this longitudinally restricted island landmass. It is possible that dispersal between southern and northern regions of Laramidia was physiographically restricted periodically by contemporaneous mountain building, topography, basin evolution, and high sea levels in a region between present day Utah and Montana (*Gates, Prieto-Márquez & Zanno, 2012*). The central Laramidia region currently occupied by the state of Wyoming represents a unique datapoint possibly representing one of these physiographic barriers along the eastern coast of Laramidia during the uppermost middle Campanian, a time in which much of the diversification of centrosaurines occurred. *Maidment et al. (2021)* pointed to this gap in sampling as an artifact in their statistical analysis in which they claim the evidence for endemism is spurious, *Gates et al. (2023)* also point out this lack data but argue for gradational latitudinal change.

The Late Cretaceous mountains produced during the Sevier orogeny, and the associated Wyoming Thrust Belt covering roughly the region from present day Salt Lake City and the Uinta Arch through Jackson Wyoming, were likely not conducive to dinosaur dispersal between northern and southern regions of Laramidia. During the Campanian, shorelines fluctuated over hundreds (~300–500) of kilometers between the hogback ridges of the Wyoming Thrust belt and the Western Interior Seaway (*Lilligraven & Ostresh, 1990*). Around ~77.5–75 Ma, the onset of the Laramide Orogeny changed the nature of the basin from a back-tilted foreland basin with abundant accommodation space to a forward-tilted, irregularly-shallowed seascape across Wyoming and extending into northeastern Colorado (*Bird, 1998*; *Steel, Plink-Bjorklund & Aschoff, 2012*). Previous studies have invoked the loss of floodplains due to sea level rise as a physical barrier between north and south (*Horner, Varricchio & Goodwin, 1992*). *Fowler & Freedman-Fowler (2020)* invoke the Wyoming thrust belt as a barrier to ceratopsian movement prior to ~80 Ma. More detailed chronostratigraphic work is needed to pin down the precise timing of this potential barrier.

There is clear evidence or at least a temporary barrier between the Western Interior Seaway and Sevier uplifts along the Utah/Wyoming border during the Campanian. The Little Muddy Creek transverse ramp structure in Fossil Basin of southwestern Wyoming (*Lamerson, 1985*) preserves Adaville Formation deposited on cliffs of the early Absaroka segment of the Sevier Thrust (*Lamerson, 1982*). The marine deposits of the Adaville Formation (*Lawrence, 1992*) represent lower shoreface sediments, including burrows and hummocky cross-stratification that indicate a depth of 8–20 m, into which large boulders originating from the cliffs above were incorporated. This overlies alluvial conglomerates of the Little Muddy Creek Conglomerate and is itself overlain by the Hams Fork Conglomerate (*DeCelles, 1994*, *2004*). At this specific time, there would only have

been a series of perched north-south oriented alluvium-filled basins between the cliffs of the thrust belt to facilitate biotic dispersal parallel to the seaway. Around ~77–73 Ma, a major river system forms a delta represented by the Ericson Sandstone that covers 3/4 of the state of Wyoming, originating on the western border (*Leary et al., 2014*). This topography led to periods of non-deposition over most of western Wyoming, punctuated by rapid incursions and regressions of the seaway, much of which is represented by the Almond Formation (*Merletti et al., 2018*).

Despite this pattern, it is unclear whether mountains, uplifted alluvial valleys, or rivers could explain the endemism documented to the north and south. First, mountains chains such as the Rocky Mountains and the Andes are poor barriers to dispersal for extant large vertebrates, especially over large timescales. Large-bodied, terrestrial mammals readily disperse around, or even across, mountains. If mountains do not drive endemism in extant large terrestrial animals, it seems unlikely they would do so for Cretaceous dinosaurs. Second, in modern ecosystems, extensive endemism evolves in the absence of geographic barriers. For example, different species of mammals inhabit different regions of North America (*Feldhamer, Thompson & Chapman, 2003*; *Qian, Badgley & Fox, 2009*) despite the absence of barriers to dispersal. Birds display high levels of endemism, especially in tropical environments (*McKnight et al., 2007*), despite being able to fly over geographic barriers such as mountains and rivers. The fact that extant endemism is not primarily driven by geographic barriers implies that Cretaceous endemism may not have been either. Third, the fact that, as discussed above, some dinosaur lineages (*e.g.*, *Parasaurolophus*, *Stegoceras*, *Gryposaurus*) did manage to disperse between northern and southern regions provides strong evidence against the existence of persistent barriers to dinosaur dispersal.

Latitudinally driven gradients in climate may have played a role if different lineages were adapted to different climatic regimes. However, the Late Cretaceous greenhouse interval (*O'Connor et al., 2019*) would have meant a weaker temperature gradient from south to north than observed today. Although high latitude environments were relatively cool (*Spicer & Herman, 2010*), the smaller difference in mean annual temperature between north and south (*Zhang et al., 2019*) may have limited the ability of climate to directly drive endemism, although patterns in ectothermic squamates do show some degree of clade-level endemism (*e.g.*, *Nydam, 2013*; *Nydam, Rowe & Cifelli, 2013*; *Woolley, Smith & Sertich, 2020*), suggesting ecosystem level differences between high and low latitude regions. Rather than temperature, higher latitude environments would still have been highly seasonable as a result of differences in photoperiod. Differences in seasonable availability of light paired with moderate temperature gradients indicate that difference in vegetation may also have played a role. Distinct floral communities existed in Laramidia (*Braman & Koppelhus, 2005*), and herbivore diet specialization may have contributed to local endemism. Although diet may well have been a factor, at times dinosaurs did disperse between northern and southern floral provinces, suggesting a degree of adaptability in terms of diets, or dispersals tied to climatic fluctuations resulting in periodic homogenization or expansion of preferred floral communities. Different species are also similar in their feeding adaptations, (*e.g.*, northern and southern *Parasaurolophus* have

similar beak and jaw morphology), arguing that they probably had relatively similar feeding strategies and diets.

Lastly, resource competition between dinosaurs may have driven endemism (*Mallon, 2019*). Strikingly, tyrannosaurs show a pattern in which multiple large-bodied species rarely if ever co-occur. When two tyrannosaur species do co-occur, they include a larger and more robust species and a smaller and more gracile species: for example *Daspletosaurus* and *Gorgosaurus* in Dinosaur Park (*Currie, 2005*), and *Tarbosaurus* and *Alioramus* in the Nemegt Formation (*Brusatte et al., 2009*). That co-existing species differed in size and morphology implies that their coexistence was made possible by niche partitioning as a means to avoid competition. Among modern birds, extant descendants of dinosaurs, the presence of competing species appears to exert a strong effect on geographic range. When competing species are absent, birds are able to occupy a wider range; when present, they are restricted (*Freeman, Strimas-Mackey & Miller, 2022*). Geographic ranges also change when competitors are eliminated; the extirpation of wolves from most of North America, for example, was followed by the rapid expansion of the coyote into areas of North America where they were previously competitively excluded (*Thurber & Peterson, 1991*). If the presence of multiple, closely-related dinosaur species did restrict the geographic ranges of other clade members, this might explain why lineages were sometimes endemic, and other times widely dispersed; that is, increases or decreases in diversity would restrict or permit broader dispersal. It is worth noting that, with the exception of the Judith River Formation, no formation is known to have two or more centrosaurine taxa living at the same time, suggesting that competition between centrosaurine species may have been a major factor dictating geographic ranges.

## Anagenesis *vs* cladogenesis

As an alternative to cladogenetic patterns, ceratopsid richness has previously been attributed to anagenetic evolution (*Horner, Varricchio & Goodwin, 1992*; *Fowler & Freedman-Fowler, 2020*; *Wilson, Ryan & Evans, 2020*). In ceratopsid taxa with large sample sizes and well-established stratigraphic position, a compelling case can be made for anagenetic transition between taxa (*e.g.*, *Triceratops*; *Scannella et al., 2014*). However, many ceratopsid taxa are known from single specimens, and are often poorly constrained stratigraphically. Previous work inferring anagenesis in dinosaurs of the WIB (*e.g.*, *Freedman-Fowler & Horner, 2015*; *Fowler, 2017*; *Carr et al., 2017*; *Fowler & Freedman-Fowler, 2020*; *Wilson, Ryan & Evans, 2020*; *Warshaw & Fowler, 2022*), rely heavily on stratigraphic position of singleton specimens to construct 'connect-the-dots' ancestor-descendant sequences. This approach ignores the fact that few of the employed taxonomic units have any actual stratigraphic ranges (as opposed to a point occurrence). Thus, the stratigraphic order between them, used as *prima facie* evidence for ancestor-descendant relationships, is perhaps nothing more than a sampling artifact. If two lineages are sister taxa with equivalent stratigraphic ranges, but only one specimen of each lineage is preserved as a fossil, it is highly unlikely that they will be found at exactly the same horizon. At the same time, the anagenesis approach would uncritically slot the 'older' occurrence into the sequence as the ancestor of the 'younger' one.

Further complicating the identification of anagenesis is the use of morphologic autapomorphies to diagnose individual species. To accommodate species with identified autapomorphies into an anagenetic lineage, one would have to lose the autapomorphies ascribed to the supposed ancestral species and evolve the autapomorphies of the descendant species. This rapid acquisition of multiple novel morphologic features is less parsimonious than inferring a cladogenetic lineage splitting event. Given this axiom of systematics dating back to *Hennig (1966)*, it would be necessary to demonstrate that these are not autapomorphies and thus can fit into an anagenetic scheme.

The newly recognized clade, Albertoceratopsini, is a prime example of the risk. As demonstrated, the three species in this clade occur within such a narrow stratigraphic interval that they are indistinguishable. Indeed, the *Wendiceratops* and *Lokiceratops* type localities cannot be distinguished stratigraphically at the resolution available to us. Any reasonable assumption of a stratigraphic range for these taxa would mean that most, if not all, overlapped in time. Relying on artifacts of incredibly low sampling that yield a (perhaps random) stratigraphic order of specimens to draw evolutionary ancestor-descendant conclusions involving extremely rapid acquisition of traits over 1–2 m of sedimentation is the less parsimonious hypothesis. While it is not impossible that *Lokiceratops* represents a member of an anagenetic lineage descended from either *Medusaceratops* or *Albertaceratops*, currently available evidence suggests that all three taxa were sympatric and a result of rapid, geographically restricted cladogenesis. The recovery of additional, stratigraphically dispersed and diagnosable individuals of all three taxa is the only true test of these hypotheses.

## Sexual selection and centrosaurine diversity

The enormous variation in centrosaurine horn and frill morphology has long fascinated and puzzled paleontologists. Disparity in these features is almost entirely responsible for the ever-growing species diversity recognized within the Centrosaurinae. Researchers have long recognized that the features that distinguish centrosaurine taxa generally are found above the naris, orbits, and along the edge (and dorsal surface) of the frill as ornamentation. This has long been interpreted as being driven by sexual selection (*e.g.*, *Knell & Sampson, 2011*; *Hone, Wood & Knell, 2016*; *Knapp, Knell & Hone, 2021*; but also see *Padian & Horner, 2014*). In the absence of characters pertaining to the morphology of frill epiossifications, phylogenetic analyses yield almost no resolution within the Centrosaurinae. With the inclusion of all 86 taxa, with no epiossification characters activated (325–377) our phylogenetic analysis recovered 10,000 most parsimonious trees (with overflow) with a length of 753 steps (Supplemental Figure S4). Despite their obvious importance with regards to centrosaurine diversity, little is understood about the mechanisms underlying epiossification variation, or their role in species recognition and or sexual selection.

However, in many ways, the centrosaurine condition resembles that present in some groups of living birds in which unique species can be distinguished only by plumage and song. For example, the diversity of sympatric centrosaurines within a small geographic area strongly resembles the diversity of capuchino seedeaters (*Sporophila*), a group that includes eight species that diverged from a common ancestor less than one million years

ago (within the last 50,000 generations) without the aid of geographic or temporal barriers (*Hejase et al., 2020*).

The southern capuchino seedeaters differ only in song and coloration; in the absence of ecological differentiation, such superficial differences are regarded as evidence of sexual selection. Variation in epiossifications among ceratopsids, especially between coeval taxa, suggests these features were highly plastic and readily respond to evolutionary pressures (*Hejase et al., 2020*). It is likely that the high diversity of centrosaurines in a restricted geographic setting is the result of high amounts of sexual selection (*Lijtmaer et al., 2004*) and that selective sweeps were responsible for the diversification of centrosaurines, acting to fix differences in epiossification morphology related to species recognition. Selection acting on the genes responsible for the epiossifications would produce modifications to cis-regulatory elements that would express differences in the size and shape of individual epiossifications producing the observed variation in ep1–6 (*Campagna et al., 2017*).

In birds, these genomic islands of differentiation are typically within the regions of the genome that code for differences in melanin, whereas it appears in centrosaurines comparable islands would have been present within the genes coding for frill ornamentation. Variations in color are optimal for diversification in volant species that cannot accommodate heavy, bony ornaments, whereas large, herbivorous quadrupeds like ceratopsids were likely unaffected by the mass gained or lost through epiossification variation. With only skeletal material to work with, it is impossible to determine if centrosaurine diversity was also accompanied by differences in coloration and vocalization.

### Implications for dinosaur diversity

The total diversity of a region, or gamma diversity, is a function of alpha diversity, or diversity on a local scale, and beta diversity, or turnover between localities. The dinosaurs of Laramidia's eastern coastal plain suggest high levels of both alpha and beta diversity. The implication is that regional dinosaur diversity was probably very high, despite the limited size of the landmass. Despite the diversity of dinosaurs known from localities such as Judith River Formation of Montana, the Dinosaur Park Formation of Alberta, and the Kaiparowits Formation of Utah, it is likely that we are significantly underestimating the total diversity of dinosaurs in North America. Our poor sampling of the West Coast, the US Southwest, Mexico, and the Arctic, not to mention Appalachia, likely means many species remain to be discovered in these regions, and that our understanding of dinosaur distribution patterns will always be hampered by gaps due to the absence of fossiliferous units preserving specific temporal intervals.

## CONCLUSIONS

A centrosaurine ceratopsid recovered from the Judith River Formation of Montana, EMK 0012, can be diagnosed as a new species, *Lokiceratops rangiformis*, based on autapomorphies of the parietosquamosal frill and associated ornamentation. The new taxon is mostly closely related to *Albertaceratops nesmoi* and *Medusaceratops lokii*, both recovered in the same small geographic area and in sediments that indicate overlapping

temporal distributions. Together, these data imply a rapid regional radiation of five distinct sympatric ceratopsid taxa (*Albertaceratops nesmoi*, *Lokiceratops rangiformis*, *Medusaceratops lokii*, *Wendiceratops pinhornensis*, and *Judiceratops tigris*). The clade containing *Lokiceratops*, Albertaceratopsini, is geographically restricted within northern Laramidia, a pattern documented in other centrosaurine clades during the Campanian and early Maastrichtian. Endemism has been documented among other dinosaurs (*Sampson et al., 2010*, *2013*; *Loewen et al., 2013*), but so far, the pattern of multiple, regional radiations evidenced here appears to be unique among macroherbivorous dinosaurs, especially those with diverse cranial ornamentation putatively related to sexual selection (*Evans & Reisz, 2007*). Centrosaurine endemism was likely driven by a combination of factors including climate-driven floral differences along a latitudinal gradient, dynamic tectonism, intense sexual selection, and interspecific resource competition, and is possibly analogous with regional radiations of sympatric lambeosaurine hadrosaurids, with the rapid speciation of dinosaurs in the Late Cretaceous of North America resulting in intense competition that restricted geographic ranges. High dinosaur endemicity implies that the diversity of dinosaurs in Laramidia was considerably higher than previously thought, because our limited geographic sampling limits our ability to recover species found in other parts of the continent.

## ACKNOWLEDGEMENTS

For access to collections in their care over the years we thank the following individuals: Matthew Carrano, Mike Brett-Surman, and Robert Purdy (USNM); Terry Ciotka; Scott Hartman (WDCB); John Horner, Patrick Leiggi, Robert Harmon, and John Scannella (MOR); Mark Norell and Carl Mehling (AMNH); Randall Irmis, Mike Getty, Eric Lund, Erin Spear, Laurel Casjen, Carrie Levitt-Bussian, Tylor Birthisel, and Monica Castro (UMNH/NHMU); Edward Daeschler (ANSP), Daniel Brinkman and Walter Joyce (YPM); Luis Chiappe (LACM), Phillip Currie (UALVP), Spencer Lucas and Thomas Williamson (NMMNH); Timothy Rowe (TMM), Robert Sullivan (State Museum of Pennsylvania), David and Janet Gillette (Museum of Northern Arizona), Rosario Gómez (Colección Paleontológica de Coahuila, at the Museo del Desierto), Hailu You (CAS, IVPP), Kevin Seymour (ROM), Kieran Shepherd (CMN), James Gardner and Brandon Strilisky (RTMP), Mark Goodwin and Patricia Holroyd (UCMP), Robert McCord (AzMNH), Khishigjav Tsogtbaatar (IGM), Jan Ove R. Ebbestad (Museum of Evolution, Uppsala University), and Xu Xing (IVPP). Łukasz Czepiński shared images and valuable conversations relating to protoceratopsians.

Marcus Donivan was instrumental in photo documenting specimens. Seth Bourgeous and Sebastian Muñoz provided preparation and molding assistance. Montrose Edmonds provided contractual technical expertise for the photogrammetric generation of digital surface files that assisted with accurate reconstructions of the skull. Cranial reconstructions of EMK 0012 were produced by Sergey Krasovskiy. Catherine Forster, John Scannella, and Denver Fowler critically read and reviewed early versions of the manuscript and their suggestions resulted in substantial improvements.

## INSTITUTIONAL ABBREVIATIONS

| | |
|---|---|
| **AMNH** | American Museum of Natural History, New York, New York, USA |
| **ANSP** | Academy of Natural Sciences, Philadelphia, USA |
| **BMNH** | British Museum of Natural History, London, United Kingdom |
| **CCMGE** | Chernyshev's Central Museum of Geological Exploration, Saint Petersburg, Russia |
| **CMN** | Canadian Museum of Nature, Ottawa,Ontario, Canada |
| **CPC** | Colección Paleontológica de Coahuila, Saltillo, Coahuila, Mexico |
| **DMNH** | Perot Museum of Nature and Science, Dallas, USA |
| **EMK** | Evolutionsmuseet (Museum of Evolution), Knuthenborg, Maribo, Denmark |
| **FDMU** | Fukui Prefectural Dinosaur Museum, Katsuyama, Japan |
| **GPDM** | Great Plains Dinosaur Museum, Malta, Montana, USA |
| **IGCAGS** | Institute of Geology Chinese Academy of Geosciences, Beijing, China |
| **IGM** | Institute of Geology Mongolia, Ulanbaatar, Mongolia |
| **IMM** | Inner Mongolia Museum, Hohhot, China |
| **IVPP** | Institute of Vertebrate Paleontology and Paleoanthropology, Beijing, China |
| **JLUM** | Jilin University Geological Museum, Changchun, China |
| **JRF** | Judith River Foundation, Great Plains Dinosaur Museum, Malta, Montana, USA |
| **JZMP** | Jinzhou Museum of Paleontology, Jinzhou City, Liaoning Province, China |
| **KIGAM** | Korea Institute of Geoscience and Mineral Resources, Daejeon, South Korea |
| **LACM** | Natural History Museum of Los Angeles County, Los Angeles, USA |
| **LH** | Long Hao Institute of Geology and Paleontology, Beijing, China |
| **MNHCM** | Mokpo Natural History and Culture Museum, Mokpo, South Korea |
| **MOR** | Museum of the Rockies, Bozeman, Montana, USA |
| **MTM** | Hungarian Natural History Museum, Budapest, Hungary |
| **NHMUK** | The Natural History Museum, London, United Kingdom |
| **NMC** | Canadian Museum of Nature, Ottawa, Ontario, Canada |
| **NMMNH** | New Mexico Museum of Natural History and Science, Albuquerque, New Mexico, USA |
| **NSM** | National Museum of Nature and Science, Tsukuba, Japan |
| **OMNH** | Sam Noble Oklahoma Museum of Natural History, Norman, Oklahoma, USA |
| **PCM** | Phillips County Museum, Malta, Montana, USA |
| **PIN** | Paleontological Institute of the Russian Academy of Sciences, Moscow, Russia |
| **PM TGU** | Paleontological Museum, Tomsk State University, Tomsk, Russia |
| **PMOL** | Paleontological Museum of Liaoning, Shenyang, China |
| **PMU** | Palaeontological Collections, Museum of Evolution, Uppsala University, Uppsala, Sweden |
| **ROM** | Royal Ontario Museum, Toronto, Canada |

| RTMP | Royal Tyrrell Museum of Palaeontology, Alberta, USA |
|---|---|
| SMP | State Museum of Pennsylvania, Pennsylvania, USA |
| TMM | Vertebrate Paleontology and Radiocarbon Laboratory, University of Texas, Austin, USA |
| TMP | Royal Tyrrell Museum of Palaeontology, Alberta, Canada |
| TCM | The Children's Museum of Indianapolis, Indiana, USA |
| UALVP | University of Alberta Laboratory of Vertebrate Paleontology, Edmonton, Alberta, Canada |
| UMNH VP | Natural History Museum of Utah, Salt Lake City, Utah, USA |
| USNM | Smithsonian Institution, National Museum of Natural History, Washington, D.C., USA |
| UTEP | University of Texas at Austin, Texas, USA |
| WDCB | Wyoming Dinosaur Center, Thermopolis, Wyoming, USA |
| YPM | Yale Peabody Museum, New Haven, Connecticut, USA |
| YPM VPPU | Princeton University Collection, Yale Peabody Museum, New Haven, Connecticut, USA |
| ZCDM | Zhucheng Dinosaur Museum, Zhucheng, China |
| ZMNH | Zhejiang Museum of Natural History, Hangzhou, China |
| ZPAL | Palaeozoological Institute, Polish Academy of Sciences, Warsaw, Poland |

### Funding

This research was financially supported by a research grant from the Evolutionsmuseet at Knuthenborg to Mark A. Loewen, Anna Øhlenschlæger, and Brock Sisson. Additional funding was provided by research grants from the National Science Foundation (NSF 0819953), Natural History Museum of Utah, and the University of Utah to Mark A. Loewen, Savhannah Carpenter, and Scott Sampson. David C. Evans was supported by a Natural Sciences and Engineering Research Council (NSERC) Discovery Grant and a Royal Ontario Museum Peer Review Grant. The funders had no role in study design, data collection and analysis, decision to publish, or preparation of the manuscript.

### Grant Disclosures

The following grant information was disclosed by the authors:
Evolutionsmuseet at Knuthenborg.
National Science Foundation: 0819953.
Natural History Museum of Utah.
University of Utah.
Natural Sciences and Engineering Research Council (NSERC) Discovery Grant and a Royal Ontario Museum Peer Review Grant.

## Competing Interests

Andrew A. Farke is an Academic Editor for PeerJ. Brock A. Sisson is owner of Fossilogic LLC, which produces cast replicas of *Lokiceratops* elements.

## Author Contributions

- Mark A. Loewen conceived and designed the experiments, performed the experiments, analyzed the data, prepared figures and/or tables, authored or reviewed drafts of the article, and approved the final draft.
- Joseph J. W. Sertich conceived and designed the experiments, performed the experiments, analyzed the data, prepared figures and/or tables, authored or reviewed drafts of the article, and approved the final draft.
- Scott Sampson analyzed the data, authored or reviewed drafts of the article, and approved the final draft.
- Jingmai K. O'Connor analyzed the data, authored or reviewed drafts of the article, and approved the final draft.
- Savhannah Carpenter analyzed the data, prepared figures and/or tables, authored or reviewed drafts of the article, and approved the final draft.
- Brock Sisson analyzed the data, prepared figures and/or tables, authored or reviewed drafts of the article, and approved the final draft.
- Anna Øhlenschlæger analyzed the data, prepared figures and/or tables, authored or reviewed drafts of the article, and approved the final draft.
- Andrew A. Farke performed the experiments, analyzed the data, authored or reviewed drafts of the article, and approved the final draft.
- Peter J. Makovicky analyzed the data, authored or reviewed drafts of the article, and approved the final draft.
- Nick Longrich analyzed the data, authored or reviewed drafts of the article, and approved the final draft.
- David C. Evans performed the experiments, analyzed the data, authored or reviewed drafts of the article, and approved the final draft.

## Data Availability

The raw data are available in the Supplemental Files.

The specimen described here (EMK 0012) is available in the publicly accessible, permanent repository of Evolutionsmuseet, Knuthenborg, Maribo, Denmark: https://knuthenborg.dk/evolutionsmuseet/.

Casts of EMK 0012 are reposited as UMNH VP C-991 at the Natural History Museum of Utah, Salt Lake City, Utah, USA and as ROM 88670 at the Royal Ontario Museum, Toronto, Canada.

## New Species Registration

The following information was supplied regarding the registration of a newly described species:

Lokiceratops LSID: urn:lsid:zoobank.org:act:4640DFB2-63D2-483A-93ED-4EF405285CAC.

Lokiceratops rangiformis LSID: urn:lsid:zoobank.org:act:548AA668-EE62-49DA-8CA2-939A00223B92.

Publication LSID: urn:lsid:zoobank.org:pub:77A46B79-9BA1-4764-9AF6-14C69C2B8C8F.

## Supplemental Information

Supplemental information for this article can be found online at http://dx.doi.org/10.7717/peerj.17224#supplemental-information.

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
