# Peer review of "Lokiceratops rangiformis gen. et sp. nov. (Ceratopsidae: Centrosaurinae) from the Campanian Judith River Formation of Montana reveals rapid regional radiations and extreme endemism within centrosaurine dinosaurs"

_PeerJ, doi:10.7717/peerj.17224_

## Round 0.1 · original submission · Major Revisions

The referees here have highly contrasting reviews, and while one recommended rejection of the manuscript, I think that the vast majority of their comments can be addressed and improved, and the key criticism about the stratigraphic work and dates was echoed by the other referee. As such, I am opting for major revisions with the chance to address these issues in the ages and diagnosis of the taxon in particular.

·

Basic reporting

This study describes a large ceratopsid from the Judith River Formation of northern Montana. The writing is clear and the paper includes many informative figures. The inclusion of the color-coded character matrix in Supplemental Information will be extremely helpful for future ceratopsid researchers. The paper names a new clade: Albertaceratopsini; it would be helpful if this clade was highlighted or labeled on one of the figured trees. Minor grammatical notes/questions are listed below.

Experimental design

No comments.

Validity of the findings

Some of the temporal distributions for taxa can be refined. For example, the temporal distributions noted in Table S1 for Einiosaurus (75.259 +/- .027 Ma) and Achelousaurus (75.259 +/- - 74.6 Ma) both include the date provided for the upper Two Med in Ramezani et al 2022 (75.259 +/- .027 Ma) and this would suggest that these taxa overlap temporally. However, these taxa do not overlap stratigraphically as Achelousaurus occurs 25 m above Einiosaurus in the Two Medicine Formation (e.g. Horner et al., 1992; Wilson et al., 2020). This impacts the apparent temporal distributions in Figure 20, S5, and elsewhere.

Additional comments

Aside from the naming/description of the ceratopsid Lokiceratops, the crux of the paper is the hypothesis that some centrosaurine dinosaurs underwent rapid evolutionary radiations and that high endemism is associated with high speciation and diversity. Several recent studies have presented evidence for anagenesis in the Campanian dinosaurs of Laramidia (e.g. Freedman Fowler and Horner, 2015; Fowler, 2017; Carr et al., 2017; Fowler and Freedman Fowler, 2020; Wilson et al., 2020; Warshaw and Fowler, 2022). If correct, this would indicate that many dinosaur taxa are members of transformational lineages as opposed to co-occurring taxa produced by evolutionary radiations. For example, the proposed anagenetic transformation of Styracosaurus > Stellasaurus >Einosaurus > Achelousaurus >Pachyrhinosaurus (Wilson et al., 2020, building on the work of Horner et al., 1992), suggests that these genera were not the product of high speciation, but instead transformation from one to the next. The data presented in the current study indicate that members of the new clade Albertaceratopsini (Albertaceratops, Medusaceratops, Lokiceratops) are known from a narrow stratigraphic interval, but are stratigraphically separated within it. As there is no record of the survival of lower occurring taxa alongside later forms, the alternative hypothesis that at least some of this apparent diversity is a result of rapid anagenesis remains. The purely cladogenetic scenario presented here could be correct, but I think it would strengthen the paper to directly discuss this by addressing the criteria for anagenesis (noted in Carr et al. 2017) and if/how these ceratopsids do not meet these criteria.

The section on Endemism (1632-1715) gives the range area in km2 for several taxa suggested to have high endemism. It would be helpful if the range areas were similarly quantified for some dinosaurs with what would be considered large geographic ranges so that they can be directly compared. Similarly, quantified range areas for some modern taxa with large geographic ranges vs. restricted ranges would also help for direct comparisons with what is being discussed regarding dinosaur ranges.

(1685-7) “These patterns require that not only were lineages isolated long enough to evolve into separate species; these lineages then underwent regional diversifications producing multiple species in the same area.” Studies indicating that several of these species are not coeval and instead are stratigraphically separated members of evolving lineages (suggesting lower diversity [e.g. Fowler, 2017; Wilson et al., 2020]) could be noted and/or addressed here or in the following section.

I commend the authors on this large-scale collaborative project and the huge number of specimens the team has directly observed and collected data for. I think the paper would be strengthened by further assessing/clarifying whether all centrosaurine taxa are proposed to be the result of speciation events leading to high diversity or if some combination of anagenesis and speciation may have produced the patterns found in Laramidia.

Some additional notes:

Line 625: “and parietals are excavated by the dorsocranial sinus”

Line 641: Maltaceratops lokii ?

Line 797: “check space” should be “cheek space”?

Lines 830-831: Perhaps reword: “No evidence of nasal ornamentation is visible. The convex lateral surface is moderately ornamented . . .”

Line 896: “The quadratojugal sits on both the jugal and quadratojugal laterally.” Should this be epijugal?

Line 1040: This section begins with “As in all other ceratopsids, the parietal is an unpaired median element” but then states “Both parietals are fused at the midline. . .”

Line 1474-5: “The parietals of subadults parietals . . .”

Line 1673: “Lokiceratopsinae” ?

Line 1622-3: “the combination of a wide midline parietal with a crescentic chasmosaurine squamosal would be unique”. Perhaps specify Campanian chasmosaurines.

Line 1820: if noting Nanotyrannus as co-occurring with Tyrannosaurus, it would be appropriate to note/cite some of the many studies indicating that these are not separate taxa (e.g. Carr, 1999; Brusatte et al., 2016; Woodward et al. 2020)

Line 1924: “Scanella” should be “Scannella” (thanks)

Figure 2: Spelling of Brachylophosaurus canadensis

Figure 20: Styracosaurus albertensis skull overlaps name

Figure 22 seems to highlight/suggest some potential anagenetic lineages but if this is the intent, it isn’t clear.

Figure S1: Check spelling: “Centrosauridae” and “Chasmosauridae”; also “Ceratipsia” (also Fig. S2 and S4).

Triceratops horridus occurs stratigraphically lower than T. prorsus (Scannella et al. 2014); the opposite is shown in Table S1 and Figure S5.

References not already in the paper:

Carr, T. D., Varricchio, D. J., Sedlmayr, J. C., Roberts, E. M., & Moore, J. R. (2017). A new tyrannosaur with evidence for anagenesis and crocodile-like facial sensory system. Scientific Reports, 7(1), 1-11.

Horner, J. R., Varricchio, D. J., & Goodwin, M. B. (1992). Marine transgressions and the evolution of Cretaceous dinosaurs. Nature, 358(6381), 59-61.

·

Basic reporting

This manuscript describes a new specimen of a centrosaurine ceratopsid from the Judith River Formation (Kennedy Coulee, northern Montana), as the holotype for a new genus and species, Lokiceratops rangiformis. Alongside closely related specimens from the same geographic area, the specimen is proposed to represent a period of exceptional diversity for ceratopsids, which is used to support hypotheses of biogeography.

I am put in a difficult position on this manuscript. I don’t agree with the interpretation, especially the paleobiological ideas about biogeography etc. I’ve complained in the past about some researchers only allowing one hypothesis to be entertained at a time, so I can hardly complain if I disagree with someone else’s ideas. Nevertheless, I think quite a lot of things need to be fixed up, notably various issues with the stratigraphy – and at the very least some acknowledgement that there are alternative paleobiological interpretations, which in my opinion at least, are more parsimonious with the record. I actually think that we are not that far from each others thinking in terms of the number of lineages, I just doubt that every time we see a new morphology, that it is evidence of vicariance.

The specimen is important and should be described. However, I think this manuscript needs a lot of things fixing before it can be published.

A lot of my specific comments are in the attached DOC of the manuscript. I’ve got some generalized points here:

WRITING
Overall, the writing has problems. Firstly, the manuscript has a lot of basic typos, missing words, and occasional language inaccuracies or contradictions (see DOC). There are many coauthors, indeed the paper has joint first authors, so it is hard for me to see how so many basic errors could have crept into the manuscript if it had been read through by even one or two of these authors.

One thing I struggled with in the descriptions, was that many of the descriptive sections start out describing the structure in general – in all ceratopsids. Then the description addresses the preserved material in EMK 0012, but it doesn’t always make clear that this is so. At the point where the description is starting to address EMK 0012, then it should say something like “in EMK 0012 the squamosal is…”. I flagged this a few times in the attached MS DOC.

Often the writing fails to distinguish between what is preserved in the particular specimen, EMK 0012, and what is present in the taxon. I’ve pointed out a few places where this occurs in the text (see MS DOC), but it is a common grammatical error. If a feature is characteristic of the taxon, then you would start a sentence with “the brow horns are not present in Lokiceratops” – this suggests that all individuals of Lokiceratops lack brow horns. Alternatively, if this is thought to only be a feature of this particular specimen, you would say “the brow horns are not present in EMK 0012” – this then allows for the authors to go on and explain that either they are damaged, lost due to taphonomy, modern preservation issues, pathological etc. i.e. they reflect only the EMK 0012 individual.

It is only a minor annoyance, but the manuscript uses odd and often pretentious language with a propensity for using buzzwords and rarely used jargon.

NAME -LOKICERATOPS
There is already Medusaceratops lokii, isn’t it potentially confusing (and unoriginal) to have Lokiceratops as the genus name? If you had a nice second choice perhaps that would be better.

THE SPECIMEN
The skull is fragmented. Few of the bones are complete. Based on the site map this would appear to have been the case in the ground, but it probably does not help that the specimen was collected with use of only two plaster jackets, and these only for the sacrum and scapula. One can only wonder how many essential little pieces might have been missed or damaged.

Still, the authors do a good job of figuring out what goes where. I would like to see more explanation of what is complete or what fits together for some critical areas.

For example, the left side of the parietal is broken into two main pieces. Do they fit together with a good click, or are they more assumed to fit? The Ep3 looks quite triangular, and it is supposedly incomplete. Similarly the Ep2 has a broken edge (although the inadequacy of the image means this is not possible to tell how broken). Is it possible that “Ep3” is actually the broken edge of Ep2?

I comment later about the images of the fossils. I would like to see good dorsal and ventral views of the skull bones, with each bone piece given a large figure. This could be in supp if necessary, but I think it is essential in order to assess the specimens and the characters they exhibit. For example, imbrication of the epis is mentioned in the descriptions, but the imbricated edge of the frill is not well figured – I can’t observe this. I’d also like to be able to see textural changes more clearly, and the underside of the frill is important to see too, as not all ontogenetic changes occur on the upper surface (or at least, not first).

IS THIS REALLY DIFFERENT FROM MEDUSACERATOPS or ALBERTACERATOPS?

It would be more parsimonious if EMK 0012 was just a specimen of a previously named taxon; either Medusa or (perhaps less likely) Alberta. Although there is a list of autapomorphies (and “differentia” for Medusa and Alberta) many differences with Medusa and Alberta are features that are known to be variable individually, or change through ontogeny. Given that EMK 0012 is probably a very aged individual, the possibility should be taken into more consideration that its unusual features are a result of it being an ontogenetic endmember.

Characters used to diagnose or differentiate the taxon are quite weak, or dubious:

(Diagnoses copied from manuscript).

“Diagnosis— Lokiceratops rangiformis is an albertaceratopsin centrosaurine ceratopsid distinguished from other centrosaurines by the following autapomorphies:
• presence of unadorned nasal;
o The nasal in EMK 0012 is incomplete medially (so the presence of a nose horn is not fully observable anyway, and it has a thickened dorsal border in the preserved section (can be seen in Figure 8), which could be a thickened low nasal horn-like structure. The nasal horns of Medusa and Alberta are not prominent at all, so it is not like EMK 0012 would be expected to have had a huge nose horn anyway. Moreover, EMK 0012 is a very large and seemingly mature individual, more so than either Medusa or Alberta. Reduction in horns is something seen during ontogeny in other ceratopsids, including many centrosaurs. Furthermore, the figure caption (8 and 9) state “Note that while the nasal appears to have a peaked ornament dorsally, there is no evidence of a narial ornament in this specimen.”. What is the “peaked ornament” then, if not some small incipient nose horn?
• elongate, uncurved ep1 epiossification directed in plane of frill along posterior margin of parietosquamosal frill
o This is indeed different from the other taxa, however I could believe that this was a late ontogeny character. It is at least possible.
• and hypertrophied, lateral curving epiparietal ep2 directed in plane of frill. The hypertrophied ep2 is relatively larger than any other parietal epiossification within Centrosaurinae.
o But a hypertrophied EP2 is present in Medusaceratops, and let’s be honest, the big EP1 in Alberta is propbably homologous to this too – the “lack” of ep1 in Alberta could be overgrowth from the large Ep2, and it’s not as if the ep1 in Medusa is very obvious either, it’s a tiny bump.
• Both ischia are distinctly kinked distally about two-thirds of the length the shaft at the point where the two ischia contact medially.
o I’d say this is fairly dubious. I don’t want to sound too skull-obsessed but this seems like a fairly major postcranial character that has no precedent, so a person should be skeptical. Looking at Figure 19, one of the ischia (F, G) is broken and reconstructed in the area of the kink, and it is possible that the other ischium is too (D, E). This is not mentioned in the text. Another reason to have larger figures.
• Postorbital horncore bases are deeply excavated by pneumatic cornual sinuses penetrating distance equivalent to two orbit radii into horncore to an extent unknown in other long horned centrosaurs.
o Excavation of cornual sinuses progresses through growth, at least where documented in chasmosaurines. I don’t see any good reason to think that this was not the case in centrosaurines that had long brow horns. I know that some chasmosaurines are being defined taxonomically on the presence of cornual sinuses (e.g. Titanoceratops, Bisticeratops), and I find that rather alarming since these are unusually large individuals giving indication of great age. The Lokiceratops holotype is also a large, presumably old, individual, so the possession of relatively well developed cornual sinuses might not be unexpected. It’s just another dubious character.

Medusaceratops lokii differs from the stratigraphically similar Lokiceratops rangiformis in a number of key features including:
• presence of nasal ornamentation;
o see above
• a lesser extent of postorbital pneumaticity;
o (see above)
• presence of four episquamosals (three in Lokiceratops);
o Dubious… see comments above
• presence of multiple, raised undulations on midline ramus of parietal between the parietal fenestrae;
o In Chasmosaurines, bumps along the midline of the parietal are more prominent in juveniles, and become resorbed thtrough ontogeny. Again, this kind of theing would therefore not be surprising in an especially large centrosaurine, as in Lokiceratops. Anyway, the bumps on the midline parietal surface are only seen in one (referred) Meduaceratops parietal, and they are not especially prominent (Chiba et al, 2017).
• lack of a narrow, medially restricted embayment on the midline of the posterior edge of the parietal;
• a reduced, rather than elongate, posteriorly directed ep1 epiossifications along the posterior margin of the parietosquamosal frill;
• the length of the largest curving epiparietal ep2;
• and the presence of five bilateral epiparietals (seven in Lokiceratops).
o A variable character in centrosaurines even between left and right sides of the same individual (see EMK 0012).

PHYLOGENY
The resultant phylogeny is odd. When Jordan Mallon had a chasmosaurine phylogeny come out upside down, with late-occurring taxa coming out basal, and older taxa derived, he said that it didn’t make a lot of sense (I forget the exact phrasing but I mention it in Fowler & Freedman Fowler 2020). Anyway, parts of this phylogeny don’t make a lot of sense, including lokiceratopsins. You can say that all you can do is objectively code the morphology, then the resultant phylogeny is basically whatever it turns out to be, but I think that some thought can go into how characters are coded, and what results that might have.

It's also not all that well supported. For example:

>LINE 1548: “Lokiceratops rangiformis is recovered as the sister taxon to Albertaceratops nesmoi and Medusaceratops lokii, and all three form the clade Albertaceratopsini. This grouping is supported by two synapomorphies;

>character 127, a circular or oval rather than narrow and slit-like frontoparietal fontanelle;

Is this even visible in EMK 0012?

>and character 357, epiparietal 1 oriented in the plane of the frill in lateral view (note that this is a local synapomorphy, found in other clades within Centrosaurinae also);

This is pretty weak. And this also means that the Loki-Medusa-Alberta clade is not united by any of the ornamentation features that would, at first, look pretty obvious – i.e. the large laterally oriented epis (either ep2 or ep1). Even if the topology is what the authors are looking for, it’s not supported by characters at nodes that make much sense.


>The decay index (1) and bootstrap support (<50%) for the relationships within Albertaceratopsini are low, although we note that low branch support is unsurprising—in this analysis: the confidently established clade Centrosaurinae itself has a decay index of only 2 and bootstrap support <50 percent.”

And this is not compelling evidence for any kind of Lokiceratops clade. Coupled with the fact that the stratigraphically young nasutoceratopsins come out basal to this in a polytomy with Xenoceratops… Either Nasutoceratopsins have a long ghost lineage, or (as seems more likely) there are problems with the cladogram from bottom up. That’s ok, not all cladograms make sense, but if you have something like this with inconsistencies it probably is not best to make this the basis of some quite serious claims about biogeography in the discussion.



STRATIGRAPHY

The manuscript contains stratigraphic errors, inconsistencies, and assertions that are unacceptable. This is a mixture of problems with radiometric dates and taxon ranges.

The manuscript (Figure 1 and in text) uses two Ar/Ar radiometric dates (78.2 & 78.5 Ma) for Kennedy Coulee from Goodwin et al., 1989 (along with a more recent U-Pb radiometric date from Ramezani et al, 2022). The 1989 dates are no longer usable in this original form as the calibration mineral, the Fish Canyon Sanidine, has been progressively dated as slightly older over the past 30 years. In order to be used, the dates must be recalibrated (greater detail for this given in Fowler, 2017). However, the current authors state that they preferred not to use recalibrated Ar/Ar dates from either Roberts et al (2013) or Fowler (2017) (or indeed, Freedman Fowler and Horner, 2015, although this is not mentioned).

Firstly, 11 out of 18 recalibrations of Roberts et al (2013) were done so incorrectly, including the Goodwin dates which as a result are 0.5my too young (the wrong age for the Fish Canyon Sanidine was entered). I point this out in Fowler (2017). The point being that the current authors are right to ignore 11 out of 18 dates in Roberts et al (2013), however the recalibrations in Fowler (2017) or Freedman Fowler and Horner (2015) were performed correctly (using slightly different standard ages). You can’t just ignore this.

And this concern over recalibrated dates is inconsistently applied as the Fowler (2017) dates ARE used in the current manuscript for other formations (see table 2 in supp). So why not for the Kennedy Coulee dates?

I rather suspect the reason is that the new date of Ramezani et al (2022) (78.594Ma) for the middle of this Kennedy Coulee section would not fit with the recalibrated dates of Fowler (2017) (79.2-79.5 Ma), so the authors just choose to ignore the Fowler recalibrations and use the original Goodwin dates instead, which as they are a million years younger, do not clash with the Ramezani dates. If you’re going to do this you need to provide some explanation – even if your explanation is “we don’t know why Ramezani’s date is so different from the historical data – but it is”.

This actually underlines a significant issue with the Ramezani U-Pb dates – they sometimes align with Ar/Ar dates (near perfectly with the Kaiparowits dates), but for a select few other dates, notably those of the Dinosaur Park Fm and surrounding strata, they are off by a million years. Ramezani et al. never note this discrepancy or any other potential issues resulting from these new dates (some explanations, including zircon residence times, would potentially explain this, although may render the DPFm dates as unreliable). One notable problem is that one new date (the only marine date in Ramezani et al 2022– for the Bearpaw Shale above the Dinosaur Park Fm) is dated at 74.289Ma, 1 my younger than the recalibrated Ar/Ar dates. The problem here is that being a marine shale, it inevitably contains ammonites and other marine fauna, so that it now requires that ammonite biostratigraphy across the WIS be revised for this period (since this is based on recalibrated Ar/Ar dates also). I rather suspect that if this was noted explicitly, that there would be pushback from people working on ammonite biostratigraphy, especially since this biostratigraphic framework is quite well resolved for multiple sections across the US (and this has integrated magnetostratigraphy, other ash dates, it will no doubt be a mess). Then there is the weird statement that (in the results): “Our U–Pb age model is unable to accurately constraint [sic] the age of the lower boundary of the Kaiparowits Fm., as bentonites are notoriously absent from its lower unit”… despite there being an Ar/Ar date published by one of the Ramezani et al 2022 coauthors for the lower Kaiparowits (Ash KDR-5, see Roberts et al 2005; 2013, or Zanno et al., 2011, or other papers). Making a statement that there is no date for the lower Kaiparowits is thus bizarre and surely requires explanation.

Anyway, the point being that the new Ramezani U-Pb dates have problems. Using them mixed with recalibrated (Fowler 2017) and unrecalibrated (Goodwin 1989) Ar/Ar dates is like having measurements in inches and cm and treating them as if they are the same. Ironically, the recalibration of the Fish Canyon Sanidine by Kuiper et al (2008) was supposed to improve alignment of Ar/Ar dates with U-Pb (and this has since been further fine-tuned), so if anything, recalibrated Ar/Ar dates should align better with U-Pb dates than previously (and for the most part they do, which is what makes a few of the Ramezani dates so problematic).

I appreciate that I am one of the only people that pays any real attention to taxon ranges, radiometric dates, and cross references this with magstrat, sequence strat, anything I can get my hands on. Most paleontologists might find all this confusing, especially when there are conflicts between methods. I suggest therefore that this manuscript acquire a coauthor who is a stratigrapher, to go through these dates and make some reasonable statements about the choices made on correlations and ranges. The authors have worked with e.g. Federico Fanti before. He would hopefully be able to fix some of these issues. As it stands it’s a mess.


TAXON RANGES
The authors suggest that (line 1727), because of the Signor Lipps effect, that we should expect Lokiceratops, Medusaceratops, and Albertaceratops to have longer stratigraphic ranges (although the same is not suggested for other taxa, which is inconsistent). Given the closeness in stratigraphic position of the specimens (as little as 4meters), then this would mean that they would inevitably have overlapping ranges. Overlaps presumably form the basis for stating that these taxa were coeval (Line 1712), then from there the logical leap is made that this represents extraordinary diversity in very small geographic area. Signor Lipps is not always appropriate. If you are sampling an anagenetic lineage, then you would not expect a given morphology to have any time range at all. See for example, the “community groups” concept (and other discussion) in Boucot (2006). I’m not saying everything is anagenesis, but I find the idea that everything is cladogenesis to be limited and likely wrong.



TAXON RANGE ERRORS
Brachylophosaurus canadensis is erroneously shown in Figure 1 as occurring ~4m below the 84MG8-3 ash. This ash is in the upper part of the section in Kennedy Coulee which represents the upper part of the lower Oldman equivalent. The leached horizon (see Eberth, 2005; especially his regional cross sections) that sits at the bottom of the upper Oldman (equivalent) is sometimes visible at the very top of Kennedy Coulee, but it has usually been cut out by surface erosion.

B. canadensis does not come from the lower Oldman. The holotype specimen comes from the Comrey sandstone, that is the basal part of the upper Oldman Fm. These sorts of simple errors are not really understandable. Maybe the authors are confused since the thickness of the Oldman Fm in Dinosaur Provincial Park is ~40m (Eberth 2005) whereas in southernmost Alberta it is up to 120m (Ryan 2003). I actually included all this kind of information as notes in Fowler (2017), so people don’t have to go looking up this base data for themselves.

I have not been through every taxon plot for this paper, but it is sloppy to make these kind of mistakes.

Spinops is of unknown range – it could be from the uppermost Oldman, or it could be from the Dinosaur Park Fm. The authors show it as uppermost Oldman. This does not reflect the uncertainty.

Pachyrhinosaurus lakustai is shown as having a range of about 600ky, yet it is known from a single mass death assemblage bonebed. This is contrasted with Einiosaurus which is also from a bonebed, yet it is shown as effectively a single line. Inconsistent. Should both of these taxa have Signor Lipps extensions, as the authors suggest for Loki, Medusa, and Albertaceratops?

Perhaps the authors need to come up with a way of showing the difference between 1. known range (ie. the position of actual fossil specimens, well constrained by stratigraphy); 2. Signor-Lipps range; 3. Possible range based on either unknown stratigraphc position or uncertain age range of host formation. I tried to do this in Fowler (2017), and included detailed notes on each taxon stipulating why it was plotted where it was. The authors should include this kind of data; a reference is not enough given the importance of stratigraphic range to the paleobiological hypotheses proposed and discussed.


Ramezani 2023 is cited as a source of stratigraphic data (supp table 2), but no citation is list in the references. Is this a pers. comm.?


FIGURES
The schematic diagrams are nicely drawn, and useful, but few individual bones are adequately figured in close up or otherwise. A lot of the time it’s impossible to see the morphology described in the text. For example, I found myself squinting at the images of the frill pieces, trying to see if they had broken edges, or trying to see details of the textures. All you can really see in the figures is the shape of the reconstructed skull. The braincase is probably the only adequately imaged bone. All the skull bones could really do with a big image in supp info, in multiple orientations.

I really appreciate that Figure 1 includes both the Rogers et al. (2016) members of the Judith alongside the much more useful Canadian terminology (including subunits). It makes it much easier for people to understand the correlation. Although see comments in MS.

Typos etc in Figures:
Figure 2: “Osetograph” (sic). Presumably you mean “Osteograph”… which is a word that I have never seen used before. Who benefits from such pretentious language?
Figure 2: Site map needs a scalebar.
Figure 3:”D” is missing.
Figure 7 there are two B’s
Figure 11 caption, no J for jugal, no L for lacrimal, and others. I assume that this is intentional since the same problem of missing terms is in figs 12 and 13.

The manuscript is very long. For brevity, it would be possible to move most of the postcranial descriptions to the supp info.

DISCUSSION
Most of the discussion I disagree with strongly. However, I can accept that the authors have a right to put forward their interpretation, but I will insist that there does need to be some acknowledgement that alternative explanations exist, however much the authors might not like them.

I will say that the suggestion that the manuscript’s claim that mountains offer no barrier to dispersal is quite a claim since it runs against long established basic principles of biogeography. Also note that in Fowler and Freedman Fowler (2020) we propose a hypothesis (based on previously unutilized high stratigraphic resolution paleomaps) where high sea levels repeatedly separated north and south(ish) regions of Laramidia during the early Campanian, potentially generating the multiple lineages of ceratopsids that we see in middle to upper Campanian rocks (I think that we’re broadly in agreement about the different lineages, I just wouldn’t split them up as much as the current authors do). Anyway, even if you don’t like this hypothesis it is current, reasonable, supported by data, and should be cited.

Experimental design

no comment

Validity of the findings

no comment

---

## Round 0.2 · Minor Revisions

Despite the number and depth of comments from Referee 2, I am considering this to be 'Minor' corrections as I think the fundamental methods and results are still sound and it is more a question of (extensive) edits and updates to the text rather than anything more major with the basics of the paper.

However, the referee does raise a number of issues that do need address, most notably the dates and ranges of taxon appearances, and some additional comments on the taxonomic characters are clearly needed. I would expect to see these more fully addressed in the next submission.

For what it's worth I don't think you need include the species recognition hypothesis as an explanation here (contra referee 1). If every unsupported hypothesis in palaeontology got a mention in every relevant paper then every discussion section would double in length.

·

Basic reporting

I think it would be helpful to highlight the new clade Albertaceratopsini on one of the figured trees.

There are misspellings in some of the supplemental figures. For example: “Centrosauridae” “Chasmosauridae” [S1]; “Montanaceratops” should be “Montanoceratops” [S1, S5].

Experimental design

No comments.

Validity of the findings

I appreciate the authors’ response to my previous comment regarding the criteria for anagenesis and why these ceratopsids do or do not meet those criteria. I think the criteria noted in the authors’ response should be included and briefly discussed in the paper itself. If that is how cladogenesis is being distinguished from anagenesis, it is an important component of the hypothesis regarding centrosaurine diversity.

Additional comments

- In the section on sexual selection, there is a list of references for sexual selection being a driver of ceratopsid diversity. I think there should also be a reference to papers proposing other hypotheses, including species recognition (e.g. Padian and Horner, 2014). Perhaps: - Line 1980: “This has long been interpreted as being driven by sexual selection (e.g., Knell and Sampson, 2011; Hone et al., 2016; Knapp et al., 2021; but see also Padian and Horner, 2014).

Padian, K. and Horner, J.R., 2014. The species recognition hypothesis explains exaggerated structures in non-avialan dinosaurs better than sexual selection does. Comptes Rendus Palevol, 13(2), pp.97-107.

- Standardize use of first names or use last names when referring to researchers throughout the manuscript. For example, lines 293-5: "To assess the systematic position of EMK 0012, the specimen was coded in a matrix initiated by Scott Sampson and Cathy Forster in the 1990’s and expanded by Mark Loewen and Andrew Farke during the 2000’s and 2010’s . . "

-Lines 1737-1741: "Combined, notable differences in adult body size, combined with multiple significant differences in cranial ornamentation (number of epiossifications, morphology of epissfications, and presence/absence of a nasal horn), are beyond intraspecific variation observed in other centrosaurine taxa, clearly differentiating Lokiceratops from other established centrosaurs from the KCA. "

I found this a little difficult to follow with "combined" used twice. Also, check spelling throughout ("epissifications").

·

Basic reporting

I'm providing my review as a word DOC

Experimental design

I'm providing my review as a word DOC

Validity of the findings

I'm providing my review as a word DOC

Additional comments

I'm providing my review as a word DOC

---

## Round 0.3 · accepted · Accept

Thank you for the work you have put into this with the changes and improvements made and the detailed responses. I think this is much improved and deals with the issues raised by the referees as far as possible (given that some disagreements over interpretation will inevitably remain).